# Single-cell genomic variation induced by mutational processes in cancer

Tyler Funnell[1,2,25], Ciara H. O'Flanagan[3,25], Marc J. Williams[2,25✉], Andrew McPherson[2], Steven McKinney[3], Farhia Kabeer[3,4], Hakwoo Lee[3,4], Sohrab Salehi[2], Ignacio Vázquez-García[2], Hongyu Shi[2], Emily Leventhal[2], Tehmina Masud[3], Peter Eirew[3], Damian Yap[3], Allen W. Zhang[3], Jamie L. P. Lim[2], Beixi Wang[3], Jazmine Brimhall[3], Justina Biele[3], Jerome Ting[3], Vinci Au[3], Michael Van Vliet[3], Yi Fei Liu[3], Sean Beatty[3], Daniel Lai[3,4], Jenifer Pham[3], Diljot Grewal[2], Douglas Abrams[2], Eliyahu Havasov[2], Samantha Leung[2], Viktoria Bojilova[2], Richard A. Moore[5], Nicole Rusk[2], Florian Uhlitz[2], Nicholas Ceglia[2], Adam C. Weiner[1,2], Elena Zaikova[3], J. Maxwell Douglas[3], Dmitriy Zamarin[6], Britta Weigelt[7], Sarah H. Kim[8], Arnaud Da Cruz Paula[8], Jorge S. Reis-Filho[7], Spencer D. Martin[4], Yangguang Li[3], Hong Xu[3], Teresa Ruiz de Algara[3], So Ra Lee[3], Viviana Cerda Llanos[3], David G. Huntsman[3,4], Jessica N. McAlpine[9], IMAXT Consortium*, Sohrab P. Shah[2✉] & Samuel Aparicio[3,4✉]

How cell-to-cell copy number alterations that underpin genomic instability[1] in human cancers drive genomic and phenotypic variation, and consequently the evolution of cancer[2], remains understudied. Here, by applying scaled single-cell whole-genome sequencing[3] to wild-type, *TP53*-deficient and *TP53*-deficient;*BRCA1*-deficient or *TP53*-deficient;*BRCA2*-deficient mammary epithelial cells (13,818 genomes), and to primary triple-negative breast cancer (TNBC) and high-grade serous ovarian cancer (HGSC) cells (22,057 genomes), we identify three distinct 'foreground' mutational patterns that are defined by cell-to-cell structural variation. Cell- and clone-specific high-level amplifications, parallel haplotype-specific copy number alterations and copy number segment length variation (serrate structural variations) had measurable phenotypic and evolutionary consequences. In TNBC and HGSC, clone-specific high-level amplifications in known oncogenes were highly prevalent in tumours bearing fold-back inversions, relative to tumours with homologous recombination deficiency, and were associated with increased clone-to-clone phenotypic variation. Parallel haplotype-specific alterations were also commonly observed, leading to phylogenetic evolutionary diversity and clone-specific mono-allelic expression. Serrate variants were increased in tumours with fold-back inversions and were highly correlated with increased genomic diversity of cellular populations. Together, our findings show that cell-to-cell structural variation contributes to the origins of phenotypic and evolutionary diversity in TNBC and HGSC, and provide insight into the genomic and mutational states of individual cancer cells.

The identification and characterization of endogenous mutational processes[4–6] have transformed our understanding of cancer genomes[6–11], and have led to improved prognostic and therapeutic stratification of cancers with genomic instability[12–14]. However, mutational processes are typically inferred from bulk whole-genome sequencing (WGS), which yields aggregate signals from pools of DNA composed of millions of cells. Thus, contemporaneous post-mitotic cell-to-cell variation due to genomic instability is not detectable in bulk sequencing, and has been understudied. Single-cell WGS can readily decompose clone-specific and cellular genomic events[2,3,15], enabling the calculation of copy number alteration (CNA) and structural variation (SV) accrual rates and mutational patterns over thousands of individual cells. This allows for the separation of evolutionary vestigial events, which are present in initial clonal expansions, from contemporaneous 'foreground' events, which reflect ongoing mechanisms of cell-to-cell genomic diversification. For example, breakage–fusion–bridge cycles (BFBCs) and homologous recombination deficiency (HRD) are endogenous mutational processes that accrue SVs with specific patterns including tandem duplications, interstitial deletions and fold-back inversions (FBIs) that generate high-level copy number amplifications[5,10,12,14]. Because HRD and BFBCs

[1]Tri-Institutional PhD Program in Computational Biology and Medicine, Weill Cornell Medicine, New York, NY, USA. [2]Computational Oncology, Department of Epidemiology and Biostatistics, Memorial Sloan Kettering Cancer Center, New York, NY, USA. [3]Department of Molecular Oncology, British Columbia Cancer Research Centre, Vancouver, British Columbia, Canada. [4]Department of Pathology and Laboratory Medicine, University of British Columbia, Vancouver, British Columbia, Canada. [5]Michael Smith Genome Sciences Centre, Vancouver, British Columbia, Canada. [6]GYN Medical Oncology, Department of Medicine, Memorial Sloan Kettering Cancer Center, New York, NY, USA. [7]Department of Pathology, Memorial Sloan Kettering Cancer Center, New York, NY, USA. [8]Department of Surgery, Memorial Sloan Kettering Cancer Center, New York, NY, USA. [9]Department of Gynecology and Obstetrics, University of British Columbia, Vancouver, British Columbia, Canada. [25]These authors contributed equally: Tyler Funnell, Ciara H. O'Flanagan, Marc J. Williams. *A list of authors and their affiliations appears at the end of the paper. ✉e-mail: william1@mskcc.org; shahs3@mskcc.org; saparicio@bccrc.ca

are predicted to induce cell-specific structural changes on individual maternal or paternal alleles, a haplotype-specific analysis is essential for a comprehensive account of genome-scale structural variation. Here we combine single-cell approaches with haplotype-specific analysis to reveal how different mutational processes diversify the genomes of individual cancer cells and thereby determine phenotypic variation and evolutionary selection in human tumours. We apply scaled single-cell WGS and haplotype-specific analysis to an in vitro cell line system with experimentally induced HRD-associated genomic instability and human breast and ovarian tumours defined by SV-associated mutational processes[6,10,12,16,17]. Our study reveals three sources of cell-to-cell variation in cancer genomes, with implications for interpreting phenotypic diversity and evolutionary selection in cancers with genomic instability.

## Induced single-cell genomic instability

We first developed a combined experimental and computational approach for studying genome-scale cell-to-cell variation in human cells, by establishing an in vitro isogenic system of breast epithelium with induced HRD and defined temporal passaging. We generated *TP53* (ref. [18]), *TP53* and *BRCA1,* and *TP53* and *BRCA2* loss-of-function genotype lineages from diploid non-transformed 184-hTERT mammary epithelial cells[19] using CRISPR–Cas9 editing (Fig. 1a, Extended Data Figs. 1 and 2a,b and Supplementary Table 1). We then subjected these cells to tagmentation whole-genome single-cell sequencing (DLP+), which enables scaled analysis of each population and inference of cell-specific rates of structural alterations[3]. In addition, we developed a computational method called SIGNALS, a hidden Markov model (HMM) which phases copy number events to individual homologues[20] in single-cell genomes to quantify haplotype-specific CNA as a source of cell-to-cell variation. SIGNALS was benchmarked on the ovarian cancer cell line OV2295, and when evaluated across different technologies and tumour types showed increased genomic and cellular resolution (0.5 Mb) compared with previously published methods[21,22], identified cell-to-cell diversity that would be unclear when relying on total copy number, and exhibited the expected distributions of phased somatic point mutation variant allele fractions (VAFs) resulting from haplotype-specific gains and losses (Extended Data Fig. 3 and Supplementary Note).

Single-cell WGS libraries (DLP+) (median 0.04× coverage, interquartile range (IQR) 0.03) from each genotype combination were generated as follows: 184-hTERT wild type ($n = 878$ genomes), 184-hTERT$^{TP53-/-}$ (*TP53*$^{-/-}$, two lines, $n = 1,634$), 184-hTERT$^{TP53-/-,BRCA1+/-}$ (*BRCA1*$^{+/-}$, $n = 377$), 184-hTERT$^{TP53-/-,BRCA1-/-}$ (*BRCA1*$^{-/-}$, $n = 382$), 184-hTERT$^{TP53-/-;BRCA2-/-}$ (*BRCA2*$^{-/-}$, two lines, $n = 887$) and 184-hTERT$^{TP53-/-;BRCA2+/-}$ (*BRCA2*$^{+/-}$, $n = 472$) (Fig. 1a, Extended Data Fig. 4 and Supplementary Tables 2 and 3). Per-cell copy number distributions showed a progressive increase in the rates of CNA as a function of *TP53* and *BRCA1* or *BRCA2* loss (Fig. 1b–e). In addition, we observed increasing whole-genome polyploidy (Fig. 1f), chromosomal missegregation (Fig. 1g and Methods) and per-cell alteration counts in *TP53*$^{-/-}$; *BRCA2*$^{-/-}$ and *BRCA1*$^{-/-}$ cells, respectively, relative to wild-type cells. *BRCA1*$^{-/-}$ genomes (median 53 events per cell) also contained higher rates of per-cell segmental alteration counts (Fig. 1h) relative to *BRCA2*$^{-/-}$ (30 and 10), *BRCA2*$^{+/-}$ (6), *BRCA1*$^{+/-}$ (6), *TP53*$^{-/-}$ (5) or wild-type (1) cell lines (all $P < 10^{-10}$). In *BRCA1*$^{-/-}$ cell lines, most cells had also undergone whole-genome duplication, consistent with *BRCA1*- and *BRCA2*-deficient cancers[23,24] (Fig. 1e,f). We then compared distributions of the ratio of gains to losses over cells, assuming that unbalanced ratios would indicate tolerance away from neutrality. The ratio was balanced in the wild-type cells and in *BRCA2*$^{+/-}$ cells; however, *BRCA1*$^{-/-}$, *BRCA1*$^{+/-}$, and *BRCA2*$^{-/-}$ cells exhibited skewed ratios towards losses relative to wild-type cells ($P < 0.05$) (Fig. 1i). SIGNALS analysis revealed extensive loss of heterozygosity (LOH) and haplotype-specific events across cells (Fig. 1b–e), with higher rates of segmental homozygosity in *BRCA1*$^{-/-}$ (6.3×, $P < 10^{-10}$) and *BRCA2*$^{-/-}$ (13.5×

and 2.5×, $P < 10^{-10}$) relative to *TP53*$^{-/-}$ (Fig. 1j). Analysis of cell-to-cell pairwise haplotype-specific copy number (HSCN) distances[20,25] found that *TP53*$^{-/-}$ induced a 3.9-fold (SA906a) and 1.9-fold (SA906b) increase in cell-to-cell divergence, *BRCA2*$^{-/-}$ induced a 4.5-fold (SA1055) and 2.6-fold (SA1056) increase and *BRCA1*$^{-/-}$ induced a 13.7-fold increase (Fig. 1k; $P < 10^{-10}$) relative to pairwise distances in wild-type cells.

We next tested whether haplotype-specific analysis at single-cell resolution could identify properties of mutational processes (Fig. 2). BFBC processes induce segmental amplifications adjacent to terminal losses on the same homologue, staircase-like copy number patterns and clustered FBI breakpoints[26–28] (Fig. 2b–d and Extended Data Fig. 5a). Using haplotype-specific alterations, we identified subclonal and variable amplitude high-level amplifications (HLAMPs, defined by 10 or more copies). HLAMPs were rare in the wild-type setting but increased with *TP53* loss of function, and further increased with *BRCA1* or *BRCA2* loss of function (Fig. 2a). Notably, some HLAMPs were consistent with BFBCs, and affected known oncogenes including *MYC* (SA1188, Fig. 2b; SA906a, Fig. 2c) and *PIK3CA* (SA1054, Fig. 2c). An early passage of SA1188 *BRCA2*$^{+/-}$ (1,395 cells; Extended Data Fig. 5a,b) exhibited hallmark patterns of BFBCs on chr. 3q through the presence of extant cells mapping to expected stepwise stages of progression with successive cell divisions. This included clusters of cells with reciprocal gains and losses, clusters in which the loss was extended and clusters in which a segmental amplification was adjacent to a terminal loss, including examples of cells with *PIK3CA* amplification (Extended Data Fig. 5c–f). Thus, an in vitro system characterized by population-scale single-cell sequencing revealed specific cell divisions that generated cell-to-cell variation in the amplitude and genomic structure of HLAMPs.

We then quantified the extent of parallel HSCN alterations, whereby cells with an identical total copy number at a given locus were composed of subsets of cells segregated by altered maternal or paternal alleles[29] (Fig. 2e,f). The rates of parallel losses and gains were increased in *TP53*$^{-/-}$ cells relative to wild-type cells, and were further increased in *BRCA1*$^{-/-}$ and *BRCA2*$^{-/-}$ populations (Fig. 2g). Notably, the parallel events affected transcriptional phenotypes that resulted from the loss of either allele A or allele B; chr. 2q in SA906b provides an example (Extended Data Fig. 5g,h). Chr 2q losses in matched single-cell RNA sequencing (scRNA-seq) data were readily identified by SIGNALS (Supplementary Note) and cells with the loss of allele A or B clustered together in gene expression space (Extended Data Fig. 5i). The nearest neighbours of monosomic 2q cells in scRNA-seq were equally enriched for losses of both A and B alleles (Extended Data Fig. 5j), suggesting that maternal and paternal allelic losses converge on a common transcriptional phenotype.

In addition to multi-allelic variation, we observed extensive cell-to-cell variation in the genomic locations of breakpoints of CNA events. The precise boundaries of CNAs from cell to cell yielded a pattern that we term 'serrate structural variation' (SSV) (Fig. 2h), which consists of a modal breakpoint across cells, with 'tails' that reflect either a progressive accumulation or 'erosion' away from the modal breakpoint. The aggregate, consensus copy number profiles over cells across the entire SSV regions (analogous to what would be seen in bulk sequencing libraries), revealed sloping copy number changes between integer values, indicative of an averaged signal with underlying variance (Fig. 2h). In some cases, these events were restricted to a single allele (for example, SA906a chr. 19), whereas in others, both alleles were implicated (for example, SA906b chr. 2). Further DLP+ sequencing from serial passaging[18] of these cells (an additional 7,793 cells), indicated that SSV events distributed across serial passaging, consistent with an ongoing mutational process (Fig. 2h).

In summary, the induction of genomic instability in breast epithelium yielded progressively higher rates of genomic divergence between individual cells, measurable as rate distributions with scaled single-cell WGS and cell-specific CNAs. The resulting 'foreground' cell-to-cell variation could be further characterized as clone- and cell-specific

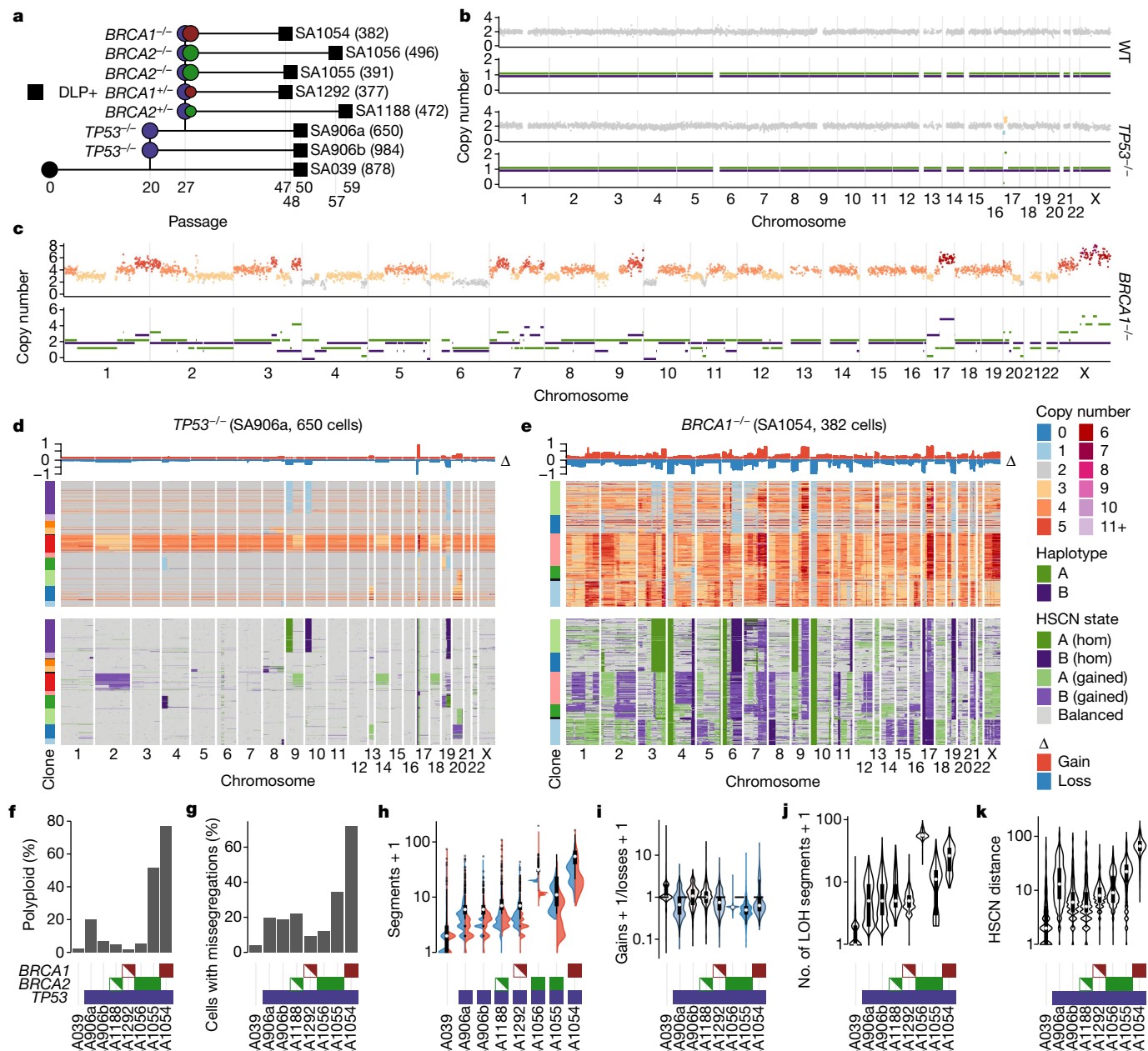

**Fig. 1 | Single-cell genome properties of CRISPR–Cas9-derived isogenic genotypes of 184-hTERT mammary epithelial cell lines. a**, Genotype lineage diagram showing wild-type→*TP53*→*BRCA1/BRCA2* alleles. The horizontal axis shows the relative passage number; the number of cell genomes per lineage is shown in parentheses. **b,c**, Wild-type (WT), *TP53*⁻/⁻ (**b**) and *BRCA1*⁻/⁻ (**c**) 184-hTERT single-cell genomes sequenced with DLP+. Top track, total copy number; bottom track, HSCN states (A haplotype in green; B haplotype in purple). **d,e**, Heat map representations of copy number profiles from cell populations of *TP53*⁻/⁻ (*n* = 650 cells) (**d**) and *BRCA1*⁻/⁻ (*n* = 382 cells) (**e**) lineages. Top, total copy number; bottom, haplotype-specific states. Rows represent cells, and columns the indicated chromosomes. Clone assignment is based on CNA profiles. hom, homozygous.

**f–k**, Comparisons of the rates of polyploidization (**f**), proportions of cells with chromosome missegregation (**g**), distributions over number of segments with gains (red), loss (blue) and either gains or loss (box plots) (**h**), distributions over ratio of gains/losses (**i**), numbers of segments that have lost heterozygosity (**j**) and distributions of pairwise HSCN distances between 250 subsampled cells (*n* = 31,125 cell pairs for all datasets; see Methods) (**k**). **f–j**, One data point per cell; number of cells as shown in **a**. **f–k**, Horizontal axes: cell line genotypes; *BRCA1* red, *BRCA2* green, *TP53* blue. Half-filled red and green boxes indicate *BRCA1*⁺/⁻ and *BRCA2*⁺/⁻, respectively. All box plots indicate the median, first and third quartiles (hinges), and the most extreme data points no farther than 1.5× the IQR from the hinge (whiskers).

HLAMP, parallel allele alteration and serriform patterns of copy number breakpoints in the cellular population.

## Cell-level CNA variation in HGSC and TNBC

On the basis of observing foreground mutational patterns defined by cell-to-cell variation, we next asked how the foreground event types

distributed as a function of HRD and non-HRD mutational processes in TNBC and HGSC cancers. To identify appropriate patient tumour samples for this comparison, we first constructed a 'meta-cohort' of 309 patients comprising 170 patients with HGSC and 139 patients with TNBC with bulk tumour–normal paired WGS to infer the distribution of established mutational processes (106 TNBC and 22 HGSC genomes were newly sequenced for this study and combined with published

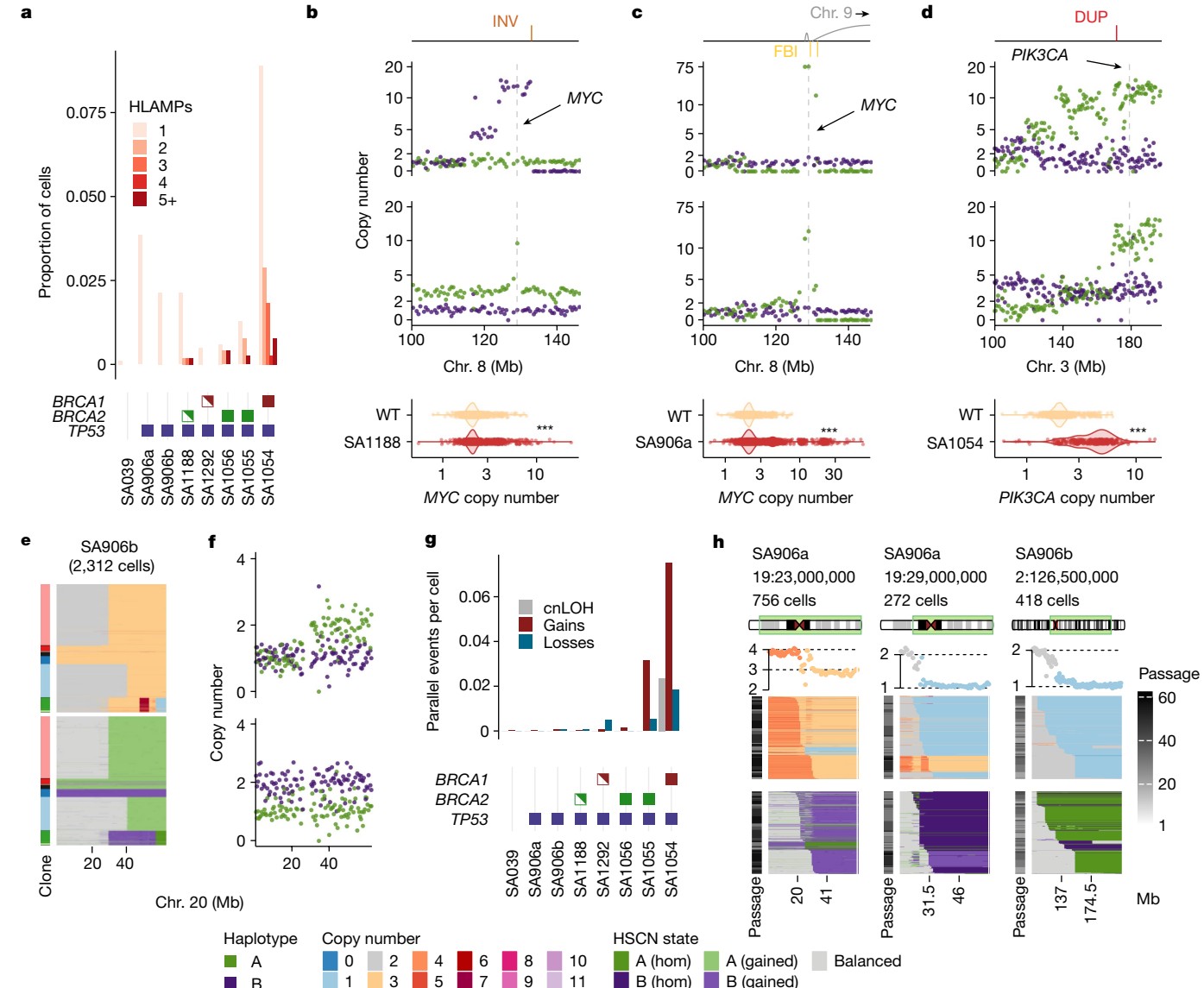

**Fig. 2 | Processes that generate cell-to-cell variation in single-cell genomes. a**, Number of HLAMPs per cell as a total proportion of cells. **b–d**, Oncogenic amplifications found in the cell lines. For each panel: annotated SVs: INV, inversion; FBI, fold-back inversion; DUP, duplications (top); HSCN in two cells with the locations of oncogenes shown with dashed lines and arrows (middle); distributions of copy number in single cells relative to wild-type cells (bottom). ***$P < 0.001$ (bottom); all $P$ values $< 10^{-10}$. **e**, Parallel copy number gains in SA906b: heat maps showing total copy number (top) and HSCN (bottom) for chr. 20 in SA906b ($n = 2,312$ cells). **f**, Two individual cells from **e**. **g**, Number of parallel copy number events per cell in 184-hTERT mammary epithelial cell lines. cnLOH, copy neutral loss of heterozygosity. **h**, Copy number heat maps showing the variation in breakpoint location across cells. Top to bottom: dataset, breakpoint location and number of cells; ideogram indicating the chromosome region shown in the heat map and the number of cells; average copy number across cells in the heat map, with breakpoint-adjacent segment copy number states indicated with dotted black lines; copy number states inferred from HMMCopy; haplotype-specific states inferred from SIGNALS. Heat map $x$ axis, genomic bins; $y$ axis, cells with the indicated breakpoint. The greyscale passage number indicates time points (passage number) of each cell; cells are ordered by breakpoint position (left to right). The left and the middle heat map represent two different clones from the same cell line with distinct breakpoints.

HGSC[12,30–32] and TNBC[3,18,33–35] datasets (Extended Data Fig. 1)). We applied a previously described correlated topic model machine learning approach (MMCTM)[10] and recapitulated previously described groups of tumours. Distinct structural copy number mutational features in both TNBC and HGSC were observed as follows: HRD-Dup (enriched in small tandem duplications and *BRCA1* mutations), HRD-Del (enriched in deletions, *BRCA2* mutations), FBI (enriched in FBIs and *CCNE1* amplification) and TD (enriched in large tandem duplications, *CDK12* mutations) (Extended Data Figs. 6 and 7 and Supplementary Tables 4 and 5). Prognostic association of these groups in the meta-cohort of patients with HGSC was consistent with previous findings[10,12] ($P = 0.0038$; Extended

Data Fig. 7d), with HRD-Del at a higher median survival than HRD-Dup, followed by FBI and TD with the worst median survival (Extended Data Fig. 7e; $P = 0.0022$). We then selected 23 cases (16 HGSC and 7 TNBC) from the meta-cohort across a range of signature types (Extended Data Fig. 1 and Supplementary Table 4), from which we generated patient-derived xenografts (PDXs), passaged over a multi-year period using subcutaneous engraftment[33] (Extended Data Fig. 2c and Methods). DLP+ libraries from HRD-Dup ($n = 8$), TD ($n = 3$), and FBI ($n = 12$) PDXs and patient tissues yielded a total of 22,057 genomes (median 556 per series), and a median of 1.96 million reads per genome (median 0.05× coverage, IQR 0.05; Extended Data Fig. 4 and Supplementary

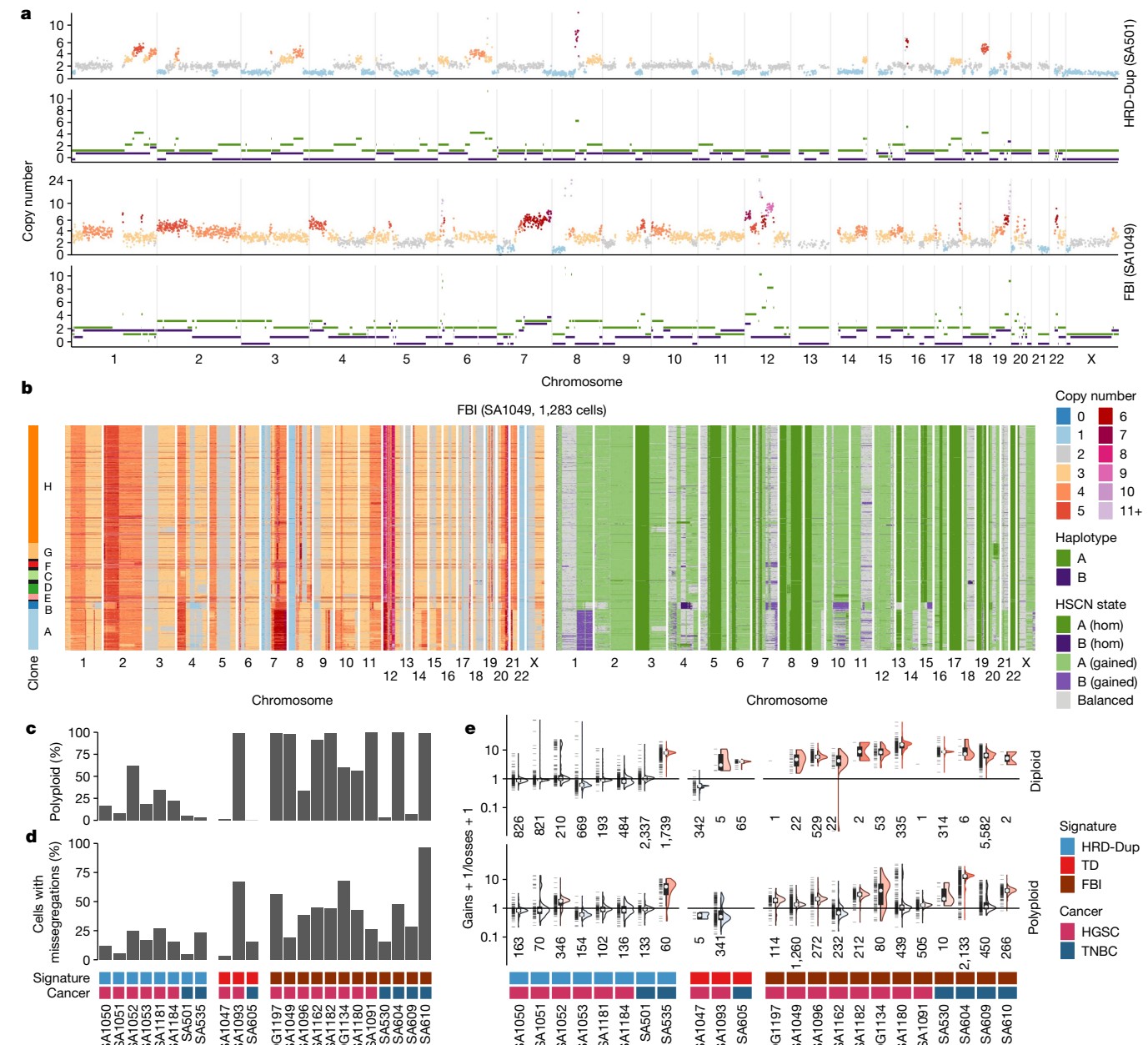

**Fig. 3 | Single-cell genome properties of PDX models and patient tissues.**
**a**, Total copy number (top) and HSCN (bottom) profiles for two single cells from a case of TNBC HRD-Dup (SA501) and a case of HGSC FBI (SA1049). **b**, Total copy number heat map (left) and HSCN heat map (right) for 1,283 cells from SA1049. **c,d**, Proportion of polyploid cells (**c**) and cells with missegregation events (**d**).

**e**, Ratio of chromosomal gains versus losses across different ploidy states and mutational signature groupings (number of cells shown below each violin plot). All box plots indicate the median, first and third quartiles (hinges), and the most extreme data points no farther than 1.5× the IQR from the hinge (whiskers).

Tables 2 and 3). Single-nucleotide variant (SNV) and SV mutational signature profiles that were inferred from DLP+-derived pseudobulk from the PDXs clustered with their bulk WGS counterparts (Extended Data Fig. 8a), indicating consistent mutational signature types without significant distortion of the signals from the original source tumour. In addition, SIGNALS analysis from DLP+ showed that the proportion of the genome identified as homozygous was highly correlated with bulk sequencing ($R = 0.9$, $P < 0.001$; Extended Data Fig. 8b), and that VAFs of somatic mutations were distributed as expected (Extended Data Fig. 8c), indicating accurate single-cell HSCN inference.

Cellular copy number profiles revealed extensive subclonal heterogeneity and cell-to-cell variation in both HRD and FBI tumours (Fig. 3a,b and Extended Data Fig. 8d). However, FBI cells exhibited higher overall

rates of polyploidy (Fig. 3c; $P = 0.02$) and chromosomal missegregation relative to HRD-Dup (Fig. 3d; $P = 0.0015$). In addition, FBI tumours accrued gains at a significantly higher rate than did HRD-Dup tumours, with more skewing of the gain/loss ratio (4.9 versus 2.1, $P = 0.04$; Fig. 3e and Extended Data Fig. 8e,f). This was more pronounced when considering the baseline ploidy of the tumours ($P = 0.0012$; Extended Data Fig. 8g). Indeed, higher rates of polyploidy and segmental copy number gains may provide a greater opportunity for—and greater tolerance of—the large interstitial deletions that are found in some FBI cancers (Extended Data Fig. 7b). Pairwise HSCN distances between cells, reflecting cell-population diversity, yielded highly variable distributions across samples, ranging from a median value of 2 for the diploid TD case SA1047 to more than 123 for the pentaploid FBI case SA604

(Extended Data Fig. 8h). FBI tumours were more diverse than HRD-Dup or TD samples, with average HSCN distances of 71 (FBI), 46 (HRD-Dup) and 26 (TD) ($P = 0.047$ FBI versus HRD-Dup, $P = 0.031$ FBI versus TD; Extended Data Fig. 8i). Thus, considering whole-genome duplication, overall rates of segmental aneuploidy and the gain/loss ratio, FBI and HRD-Dup tumours showed markedly different patterns of CNA accrual at the single-cell level.

## HLAMP amplitude varies within FBI tumours

Next, we determined whether the CNA patterns that gave rise to single-cell variation in the cell lines could also explain cell-to-cell variation in the amplitude of HLAMPs in the tumours. We found extensive heterogeneity in the amplitude of HLAMPs across clonal populations within tumours, that would otherwise be obscured in bulk sequencing. By first focusing on a specific example, we assessed the phenotypic effect of a clone-specific HLAMP in the *KRAS* locus, present with average copy number 16.1 in a clone with 55 cells relative to a sibling clone composed of 230 cells that lacked the amplification (Fig. 4a). *KRAS* was differentially expressed between cell clusters from matched scRNA-seq data (maximum log-transformed fold change (logFC) = 0.346, $q < 0.05$; Fig. 4b), and immunohistochemistry for KRAS both in tissue from the primary patient and in PDX tissue corroborated a punctate pattern of expression across spatially separated regions within tumour sections (Fig. 4c). Thus, in a specific example, clone-specific HLAMP of an oncogene in a minor clone—otherwise not detectable with bulk methods—revealed co-associated clone-specific phenotypic variation. Across the dataset as a whole, FBI tumours had a 1.9-fold higher median HLAMP copy variance than did the other tumours ($P = 0.00096$; Fig. 4d,e), consistent with continual plasticity of HLAMP amplitude as a general property of FBI. Most events were less than 10 Mb in width (56%; Fig. 4f) and exhibited a distribution of maximum observed copy number with median 16.1 and IQR 8.7 (Fig. 4g). Furthermore, we noticed that amplitude variation in HLAMPs affected numerous other known oncogenes, including *ERBB2* (DG1197), *KIT* (DG1197), *KRAS* (SA1049 and SA604), *MYC* (SA1184 and SA1051), *CCNE1* (DG1134, SA1162 and SA604) and *FGFR1* (SA1049 and SA535) (Fig. 4h). Notably, oncogenes with a variable copy number between cells and clones also exhibited greater variability in gene expression than did other genes, as measured by matched scRNA-seq ($P = 0.012$; Fig. 4i).

To determine the structural processes that lead to these events, we found that the rearrangement properties of variable HLAMPs were enriched for FBIs, consistent with BFBCs being a central mechanism of variable HLAMPs in FBI tumours. We also found clusters enriched for simple tandem duplications driving variable HLAMPs in HRD-Dup tumours (Methods and Extended Data Fig. 9a), providing further evidence for the different aetiological origin of these events in FBI and HRD-Dup tumours. This analysis also revealed that in many cases, clone-specific HLAMPs were part of complex genomic structures involving multiple chromosomes. For example, variable amplitude around the *CTNNB1* locus in SA1096 coincided with a translocation between chr. 3 and chr. 6 (Fig. 4j). Long-read single-molecule nanopore sequencing[36] of the same samples validated the presence of this rearrangement (Extended Data Fig. 9d). Other examples of complex inter-chromosomal HLAMPs with orthogonal long-read sequencing included fixed non-variable amplification of *CCNE1* in SA530 (chr. 4 and chr. 19), variable *MYC* amplification in SA1184 (chr. 3 and chr. 8) and amplification on 5q in SA1184 (Extended Data Fig. 9b,c and Supplementary Note).

Thus, cell-to-cell variability in HLAMP—which is not observable with bulk sequencing—is a pervasive mutational pattern that is most pronounced in FBI tumours, and consists of clone-specific complex rearrangements that influence phenotype through variable oncogene expression.

## Haplotype-specific parallel evolution

We next investigated the extent of haplotype-specific parallel copy number evolution in tumours (Fig. 5). Phylogenetic tree analysis using breakpoints inferred from total copy number across the whole genome[18,37] (see Methods) revealed that in some cases, alleles segregated into distinct clades on the tree; for example, gains of 1q in SA1049 (Fig. 5a) and losses at the terminal end of chr. 10 in SA1053 (Fig. 5b). In other cases, gains and losses of different alleles were sporadic and were distributed more randomly across the tree, such as chr. 8 in SA1093 (Fig. 5c). Parallel copy number events were validated using VAFs of mutations found in these regions, in which—as expected—the VAF distribution inverted between two expected values, depending on allelic composition (Fig. 5d). We contend that in bulk sequencing, represented here by pseudobulk with mixtures (see Methods), the computed VAF no longer reflected the underlying copy number state in a heterogeneous mix of cells (Fig. 5e). We therefore suggest that accurate cancer cell fraction (CCF) inference, which depends on accurate VAF values, may be challenging in tumours with parallel copy number evolution.

We confirmed that parallel CNAs influence transcription with matched scRNA-seq. Inactivation of *TP53* is invariably mediated by LOH of chr. 17 in these cancer types, and chr. 17 was indeed mono-allelically expressed across all tumours—in contrast to the hTERT wild-type cell line, which was used here as a control population (Fig. 5i). In addition, genes located at the terminal end of chr. 10 in SA1053 (Fig. 5b), were mono-allelically expressed in 100% of cells, with one cluster of cells expressing the B allele and another group of cells expressing the A allele (Fig. 5f,g). Across all data with matched scRNA-seq, mean BAF values per segment per sample measured in single-cell DNA sequencing (scDNA-seq) were strongly correlated ($R = 0.91$, $P < 10^{-5}$) with those measured in scRNA-seq (Fig. 5h), consistent with allele bias at the DNA level translating to consequent allele bias in expression.

Notably, nearly all tumours exhibited parallel CNA evolution. We classified genomic segments as parallel CNAs if more than 1% of cells had gain or loss of both the A and B alleles and assigned the clonality using total copy number as follows: clonal (CCF > 80%, as in Fig. 5a,b), subclonal (20% < CCF ≤ 80%) or rare (CCF ≤ 20%, as in Fig. 5c). Every tumour had at least one parallel CNA event, with most containing parallel CNAs at different clonalities (Fig. 5j). Across all samples, an average of 6% of clonal segments, 15% of subclonal segments and 7% of rare segments contained parallel CNAs, with a trend for higher event rates in FBI relative to HRD-Dup (Extended Data Fig. 8j,k; $P = 0.02$ for subclonal, $P > 0.05$ for clonal and rare). Motivated by the sporadic pattern of losses of both alleles observed in Fig. 5c, we then tested whether parallel CNAs due to losses were more common on a tetraploid versus a diploid background, using ancestral state reconstruction to estimate the event rate across the phylogenetic tree (see Methods). We found that on a diploid (1|1) background, parallel gains were more common than losses, but that on a tetraploid (2|2) background, parallel losses became more common than gains. This was true for whole chromosome, chromosome arm and segmental aneuploidies (Fig. 5j). The number of parallel CNAs was significantly correlated with both copy number and phylogenetic distances computed using total copy number (Fig. 5k,l). Thus, parallel copy number evolution was a pervasive feature, affecting the interpretation of somatic mutations, haplotype-specific expression and overall levels of genomic diversity in TNBC and HGSC tumours.

## Increased CNA serriformity in FBI tumours

SSVs first identified in cell lines from single-cell WGS represent a structural mutation type that is not identifiable using bulk WGS, as the cell-to-cell variation in copy number breakpoints is obscured. We analysed the tumour DLP+ data for the presence of SSVs (Fig. 6a; heat maps 2–5 from left). SSVs, occurring at a megabase length scale, were distinct from small cell-to-cell variations in copy number breakpoint

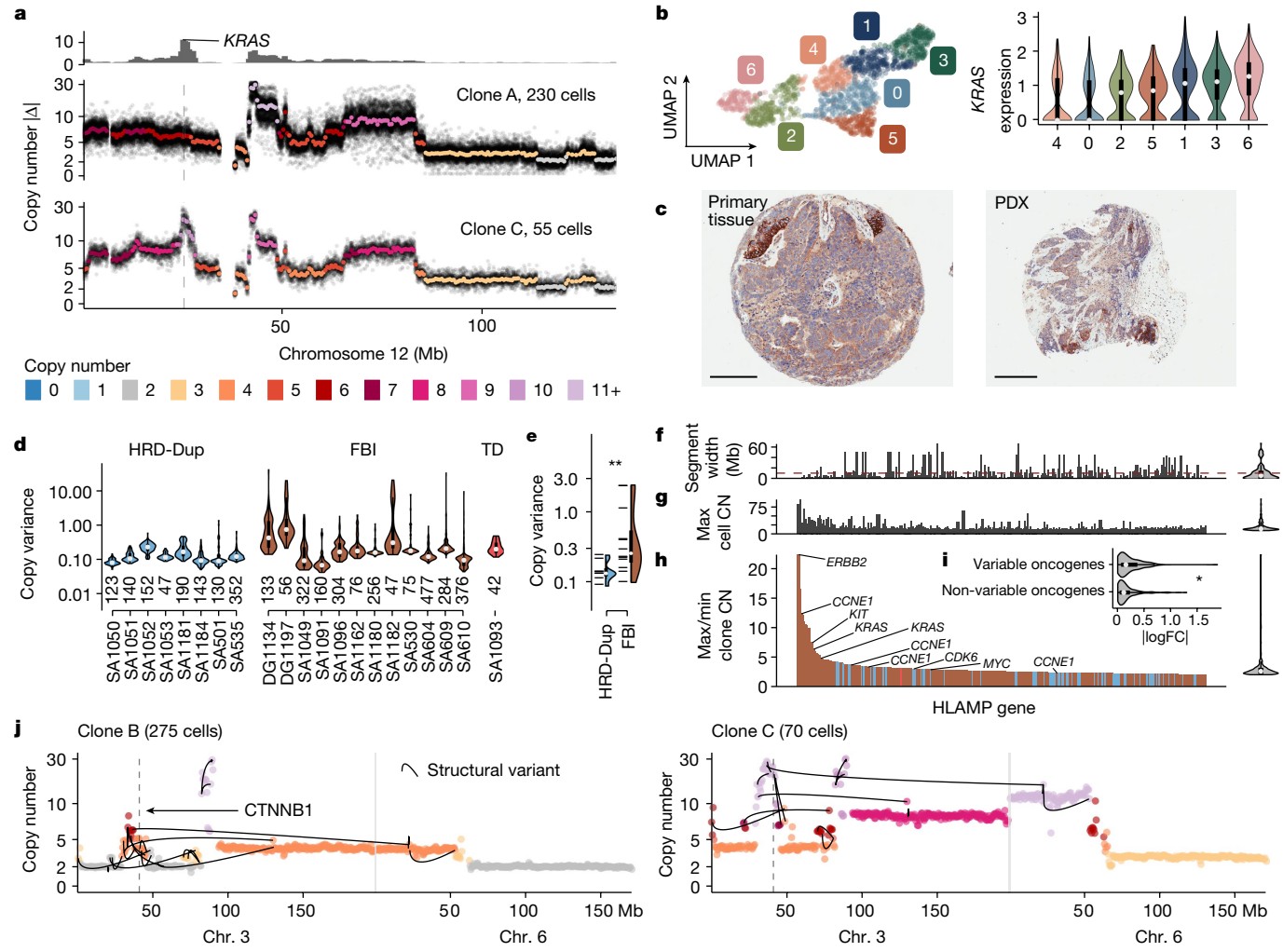

**Fig. 4 | HLAMP copy number variation. a,** Clone and single-cell whole-genome consensus copy number profiles for chromosome 12 in FBI tumour (SA1049) clones A and C. The top track shows the absolute difference between the copy number of the two clones; the bottom two panels show consensus copy number profiles (coloured points) and all single-cell values (small black points). Cell numbers indicate the number of cells used for consensus copy number evaluation. **b,** Uniform manifold approximation and projection (UMAP) dimensionality reduction of scRNA-seq data from SA1049 coloured by gene expression cluster and violin plots of gene expression per cluster of *KRAS*; *n* = 1,697 cells (per cluster: 0 = 287, 1 = 286, 2 = 276, 3 = 270, 4 = 221, 5 = 200, 6 = 157). **c,** Immunohistochemistry staining of KRAS protein in primary human tissue (left) and PDX tissue (right). Scale bars, 200 μm (left); 300 μm (right). Images are representative of two cores stained from each PDX tissue. **d,** Copy number variance across cells for HLAMP bins within each dataset (number of bins shown below each violin). Datasets without HLAMPs not shown.

**e,** Distribution of mean copy variance in eight HRD-Dup cases versus 12 FBI cases. *P* = 0.0096 (two-sided Wilcoxon test); **\*\****P* < 0.01. Notches show individual data points. **f,** Width of genomic segment containing the amplification; dashed red line indicates a width of 10 Mb. **g,** Maximum cell copy number (CN) per gene. **h,** Clone maximum/minimum copy number ratio of cancer genes overlapping HLAMP regions. Genes across all cancer datasets with ratio > 2 are shown (*n* = 296). Colours as in **d.** For **f**–**h,** distributions of values are shown in a violin plot on the right. **i,** Distribution of the maximum logFC between gene expression clusters in matched scRNA-seq for variable oncogenes (*n* = 140) versus non-variable oncogenes (*n* = 159). *P* = 0.019 (two-sided Wilcoxon test). **j,** Consensus copy number profiles in two clones in SA1096 overlaid with lines indicating SVs. All box plots indicate the median, first and third quartiles (hinges), and the most extreme data points no farther than 1.5× the IQR from the hinge (whiskers).

localization, which may occur owing to fluctuations in sequencing coverage rather than true changes in copy number (for example, Fig. 6a; invariant copy number heat map boundaries, heat map 1 from left). SSVs were also visible in single cells comprising the serration pattern (Fig. 6b). Additional confirmation of the SSV scale was obtained from allele phasing, in which the concomitant loss of heterozygosity was observed (Fig. 6b, bottom track). We computed serration scores per breakpoint event to identify the relative degree of variation in breakpoints across cells in each cancer. Scores were calculated as the fraction of event-containing cells with rare (less than 5% of cells) event breakpoint positions (see Methods) and with breakpoint regions that met size and prevalence criteria (≥20 Mbp, that is, 40 genomic bins; ≥100 cells with breakpoint event) to permit the detection of positional and

cell-to-cell variation. Variable cell-to-cell breakpoints were common, with 6.6% of regions having serration scores of 0.15 or higher (that is, 15% of cells or more have a rare event breakpoint position) across all cases, with FBI cases having the highest (12.1%), HRD-Dup cases the lowest (1.2%), and TD cases intermediate (10.5%) rates. Comparison of distributions of serration using a mixed effects linear model accounting for individual variation indicated that FBI cases had higher degrees of breakpoint variance in breakpoint regions (*P* = 0.0081; Fig. 6c,d). We further observed that serration scores increased in cases with more polyploid cells (*R* = 0.68, *P* = 0.001; Fig. 6e), and as a function of cell-to-cell HSCN distance (*R* = 0.62, *P* = 0.0033; Fig. 6f), implicating SSVs as an additional genome-diversifying mechanism in TNBC and HGSC cancers.

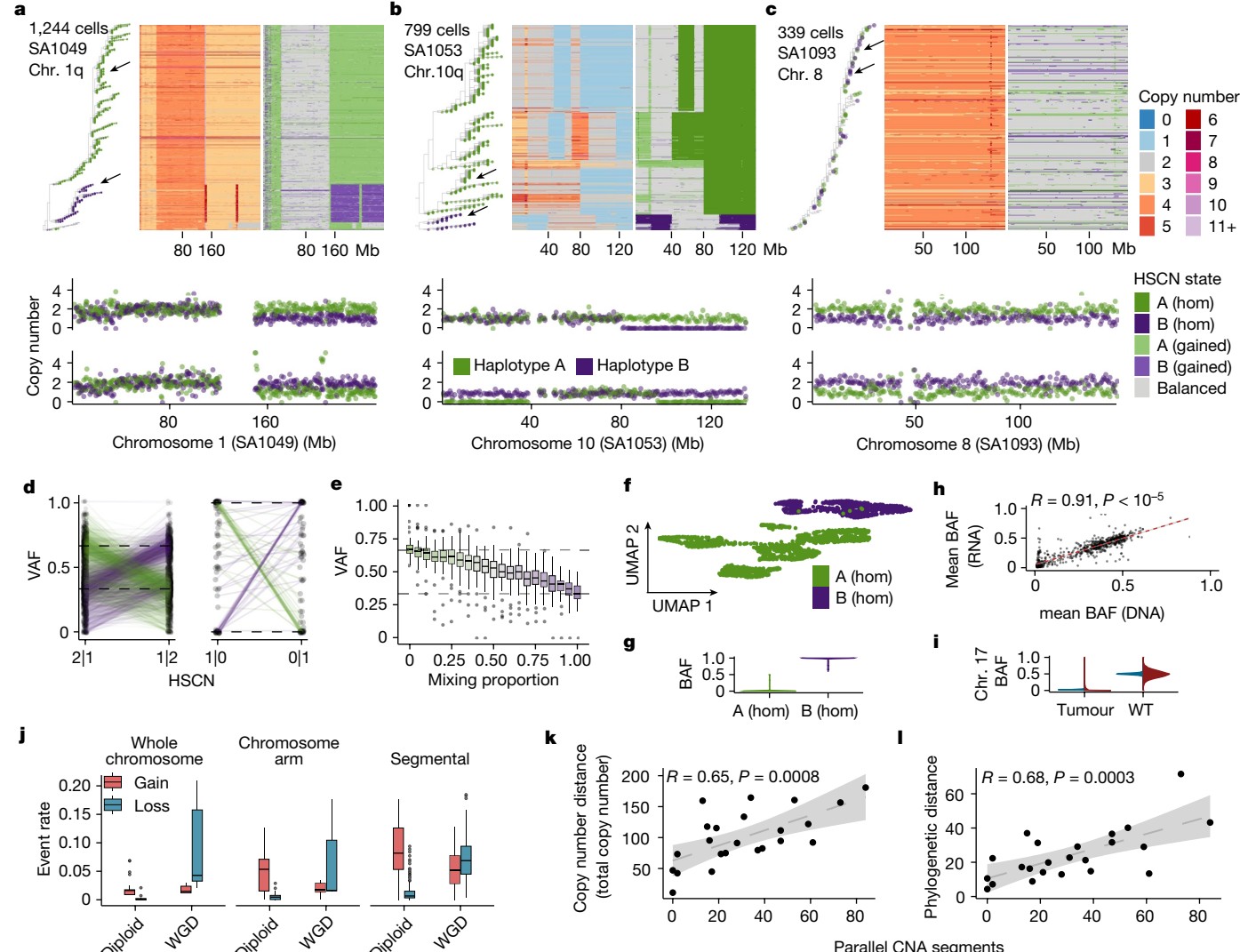

**Fig. 5 | Haplotype-specific parallel copy number evolution. a–c**, Heat maps of chromosomes 1q (**a**), 10q (**b**) and 8 (**c**) ordered by a phylogenetic tree. The tips of the phylogeny are coloured according to the allelic phase of the region of interest. Arrows indicate single cells, the copy number profiles of which are shown below each heat map. **d**, VAFs of SNVs in parallel copy number events in two haplotype-specific states in which the dominant allele switches between the two states. Each point is the VAF of a single SNV; lines connect the same SNV in the two states. Dashed lines indicate the expected VAF on the basis of the states. **e**, VAF of mutations (*n* = 66) present clonally on allele A after computationally mixing data from SA535 chr. 8 in cells with copy number 2|1 and 1|2. Mixing proportion = 0 means that all cells are in state 2|1 and mixing proportion = 1 means that all cells are in state 1|2. **f**, UMAP of scRNA-seq data from SA1053 coloured by allelic state of genes at the terminal end of chromosome 10. A (hom), *n* = 1,614 cells; B (hom), *n* = 890 cells. **g**, BAF (B-allele frequency) distribution of cells in **f**. **h**, Scatter plot of mean BAF per segment across all datasets (*n* = 828) computed in RNA versus DNA. **i**, BAF distribution on chromosome 17 in all tumours and cell lines with matched scRNA (*n* = 21,347 cells DNA; *n* = 70,553 cells RNA) versus wild-type cell line (*n* = 1,963 cells DNA; *n* = 5,752 cells RNA). **j**, Rate of gains and losses within whole chromosomes (*n* = 35 events), chromosome arms (*n* = 31 events) and segments (*n* = 341 events) on diploid (1|1) and tetraploid (2|2) backgrounds. WGD, whole-genome duplication. **k,l**, Correlation of the number of parallel copy number events with copy number distance (*P* = 0.0008) (**k**) and phylogenetic distance (*P* = 0.0003) (**l**). Annotations at the top indicate the correlation coefficient (*R*) and *P* value derived from a linear regression; shaded areas in plots show the 95% confidence interval of the linear regression. All box plots indicate the median, first and third quartiles (hinges), and the most extreme data points no farther than 1.5× the IQR from the hinge (whiskers).

## Discussion

Our findings show that cell-to-cell variation at the level of structural and copy number alteration is a pervasive 'foreground' feature of TNBC and HGSC cancers that is exhibited against distinct endogenous mutational processes of genomic instability. Because CNAs can influence the expression levels of hundreds of genes, each of the foreground mutational patterns provides extensive and distinct genomic diversity upon which selection may act. Oncogenic HLAMPs are understood to be key drivers of tumour progression and are prognostic in HGSC when co-localized with FBIs[12]. Here we reveal an additional layer of complexity, finding that the amplitude of HLAMPs can vary substantially between cells. Although this has been recognized as a defining feature of extrachromosomal DNA amplifications[38], we propose that it is also a general property of other classes of HLAMPs, such as those mediated by BFBCs and by complex inter-chromosomal rearrangement processes[39]. This has important implications for therapeutic strategies to target frequently altered oncogenes, as cancer types of high genomic instability may be predisposed to containing treatment-resistant clones. Multi-allelic variation within the same locus is also a highly prevalent feature of breast and ovarian cancers, consistent with some previous observations in other cancers[21,22,30].

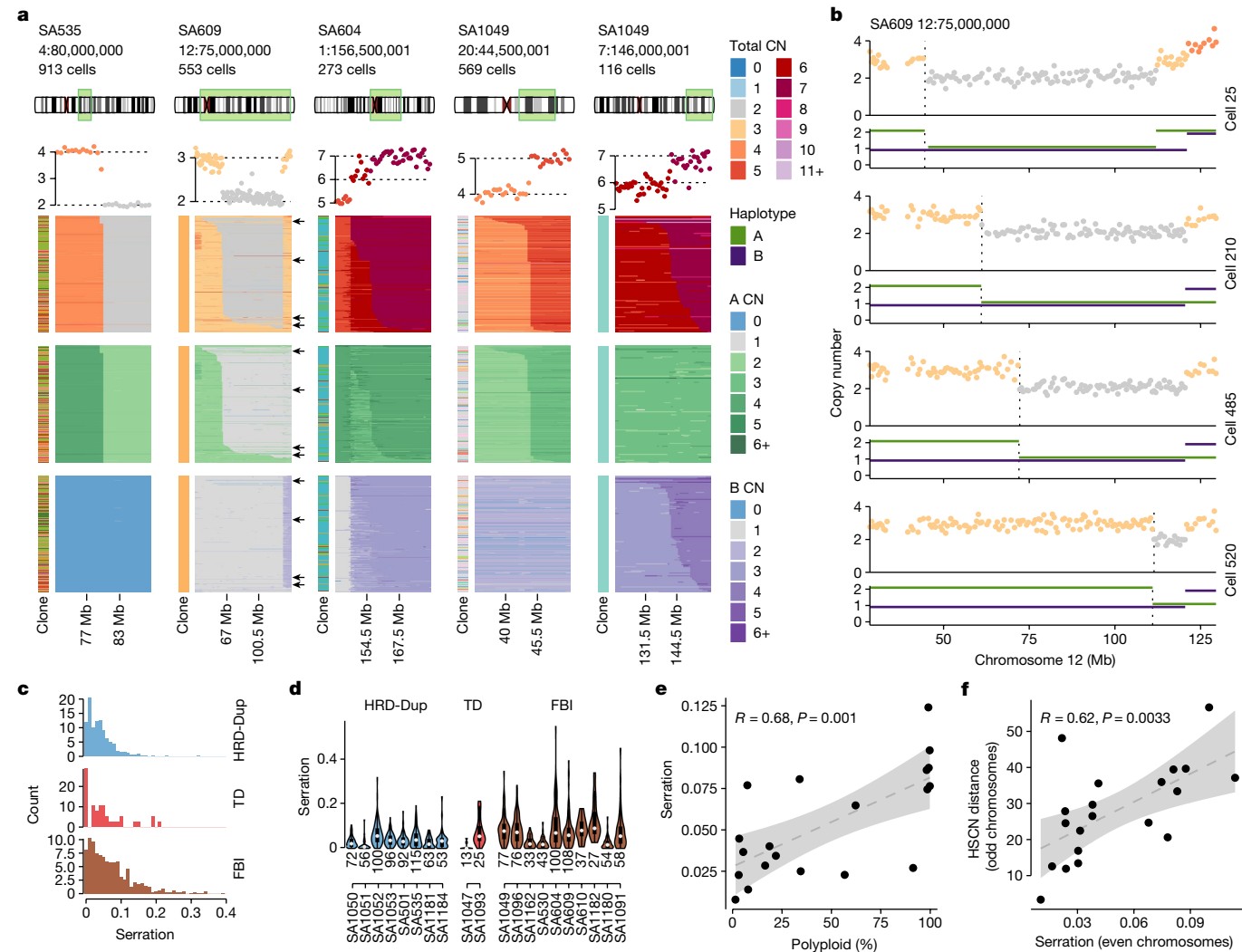

**Fig. 6 | Breakpoint serriform variability. a**, Copy number heat maps showing variation in breakpoint location on the horizontal axis across single cells along the vertical axis. Top to bottom: dataset, breakpoint location and number of cells; ideogram indicating the chromosome region shown in the heat map; average copy number across cells in the heat map, with breakpoint-adjacent segment copy number states indicated with dotted black lines; copy number states inferred by HMMCopy; A allele (green) and B allele (purple) copy number states inferred by SIGNALS, Copy number state shading is shown in the adjacent key. Heat map *x* axis, genomic bins; *y* axis, cells with the indicated breakpoint. Cells are ordered by breakpoint position (left to right). Arrows in heat map 2 (from left) indicate the cells shown in **b**. **b**, Four single-cell copy number profiles from the SA609 SSV event in **a**. The cell number from the top of the heat map is indicated to the right of the profiles. Top, total copy number; bottom, HSCN. A and B alleles are indicated with green and purple, respectively. The dotted vertical line indicates the cell-specific breakpoint location. **c**, Breakpoint serration distribution for all cancer datasets for which scores could be computed (see Methods), segregated as FBI (brown), TD (red) and HRD (blue). **d**, Distribution of serration scores by case; colours as in **c** (number of scores shown below each violin). **e**, Mean per-case serration scores versus polyploid cell percentage. **f**, Mean cell-to-cell HSCN distance (even chromosomes only) per case versus serration (odd chromosomes only). Shaded areas show the 95% confidence interval of the linear regression; the correlation coefficient and *P* values are annotated at the top (*P* = 0.001 for **e** and *P* = 0.00061 for **f**). All box plots indicate the median, first and third quartiles (hinges), and the most extreme data points no farther than 1.5× the IQR from the hinge (whiskers).

Notably, events that appeared clonal at the total copy number level were often composed of distinct clades with different alleles gained or lost; this might reflect evolutionary convergence for favourable karyotypes at the total copy number level, as shown by transcriptional phenotypic convergence. Evolutionary time series modelling[18] is likely to further help to resolve patterns of phenotypic selection from parallel CNAs. We also highlight that sporadic gains and losses happen on both alleles, with rates increased on a whole-genome-doubled background relative to diploid, potentially reflecting increased fitness tolerance owing to genomic redundancy. Finally, megabase-scale copy length variation at a single-cell level (SSV) has been observed in vitro with cell-selected single-cell sequencing[1]. Here we show with single-cell genome sequencing at the cell-population level that SSVs are in fact prevalent in TNBC and HGSC and distribute across clones within tumours. Although the underlying mechanisms that generate SSVs are unknown, they represent a new class of variation that may contribute to the structural copy evolution of tumours enriched in the FBI background and in polyploid genome states. We observed each of the foreground mutational patterns in all mutational processes, but FBI-type tumours showed a significant enrichment in all three foreground patterns. As such, FBI may comprise a distinct phenotypic class in which foreground mutational patterns generate diversity that could underlie poor prognostic significance. We conclude that scaled single-cell sequencing is a useful means to reveal hidden cellular states of structural copy number diversity in genomically unstable tumours. The data that we present here show that foreground

mutational patterns are key determinants of genomically encoded phenotypic diversity and consequent 'evolvability' in cancer.

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

**IMAXT Consortium**

Gregory J. Hannon[10], Georgia Battistoni[10], Dario Bressan[10], Ian G. Cannell[10], Hannah Casbolt[10], Cristina Jauset[10], Tatjana Kovačević[10], Claire M. Mulvey[10], Fiona Nugent[10], Marta Paez Ribes[10], Isabella Pearson[10], Fatime Qosaj[10], Kirsty Sawicka[10], Sophia A. Wild[10], Elena Williams[10], Samuel Aparicio[3,4], Emma Laks[3], Yangguang Li[3], Ciara H. O'Flanagan[3,26], Austin Smith[3], Teresa Ruiz de Algara[3], Daniel Lai[3], Andrew Roth[3,4,11], Shankar Balasubramanian[10,12,13], Maximilian Lee[10,12,13], Bernd Bodenmiller[14], Marcel Burger[14], Laura Kuett[14], Sandra Tietscher[14], Jonas Windhager[14], Edward S. Boyden[15], Shahar Alon[15], Yi Cui[15], Amauche Emenari[15], Daniel R. Goodwin[15], Emmanouil D. Karagiannis[15], Anubhav Sinha[15], Asmamaw T. Wassie[15], Carlos Caldas[16], Alejandra Bruna[16], Maurizio Callari[10], Wendy Greenwood[10], Giulia Lerda[10], Yaniv Eyal-Lubling[16], Oscar M. Rueda[16], Abigail Shea[16], Owen Harris[17], Robby Becker[17], Flaminia Grimaldo[17], Suvi Harris[17], Sara Lisa Vogl[17], Johanna A. Joyce[18], Spencer S. Watson[18], Sohrab P. Shah[2], Andrew McPherson[2], Ignacio Vázquez-García[2], Simon Tavare[10,19,20], Khanh N. Dinh[19,20], Eyal Fisher[10], Russell Kunes[19,20], Nicholas A. Walton[21], Mohammed Al Sa'd[21], Nick Chornay[21], Ali Dariush[21], Eduardo A. González-Solares[21], Carlos González-Fernández[21], Aybüke Küpcü Yoldaş[21], Neil Miller[21], Xiaowei Zhuang[22,23,24], Jean Fan[22,23,24], Hsuan Lee[22,23,24], Leonardo A. Sepúlveda[22,23,24], Chenglong Xia[22,23,24] & Pu Zheng[22,23,24]

[10]Cancer Research UK Cambridge Institute, Li Ka Shing Centre, University of Cambridge, Cambridge, UK. [11]UBC Data Science Institute, University of British Columbia, Vancouver, British Columbia, Canada. [12]Department of Chemistry, University of Cambridge, Cambridge, UK. [13]School of Clinical Medicine, University of Cambridge, Cambridge, UK. [14]Department of Quantitative Biomedicine, University of Zurich, Zurich, Switzerland. [15]McGovern Institute, Departments of Biological Engineering and Brain and Cognitive Sciences, Massachusetts Institute of Technology, Cambridge, MA, USA. [16]Department of Oncology and Cancer Research UK Cambridge Institute, University of Cambridge, Cambridge, UK. [17]Súil Interactive, Dublin, Ireland. [18]Department of Oncology and Ludwig Institute for Cancer Research, University of Lausanne, Lausanne, Switzerland. [19]Herbert and Florence Irving Institute for Cancer Dynamics, Columbia University, New York, NY, USA. [20]New York Genome Center, New York, NY, USA. [21]Institute of Astronomy, University of Cambridge, Cambridge, UK. [22]Howard Hughes Medical Institute, Harvard University, Cambridge, MA, USA. [23]Department of Physics, Harvard University, Cambridge, MA, USA. [24]Department of Chemistry and Chemical Biology, Harvard University, Cambridge, MA, USA.

## Methods

### Generation and culture of human mammary epithelial cell lines

The wild-type human mammary epithelial cell line 184-hTERT L9 (SA039) and isogenic 184-hTERT *TP53* knockout (SA906) cell line, generated from 184-hTERT L9, were cultured as previously described[18,19,40] in Mammary epithelial cell growth basal medium (MEBM) (Lonza) supplemented with the SingleQuots kit (Lonza), 5 µg ml$^{-1}$ transferrin (Sigma-Aldrich) and 10 µM isoproterenol (Sigma-Aldrich). Additional truncation mutations (Supplementary Table 4) of *BRCA1* (SA1054: c.[427_441+36delGAAAATCCTTCCTTGGTAAAACCATTTGTTTTCTTC]; [437_441+8delCCTTGGTAAAACC]; SA1292 c.[71_75delGTCCC];[=]) and *BRCA2* (SA1056: c.[6997delG];[6997_6998delGT]; SA1188: c.[6997_6999delGT];[=]; SA1055: c.[3507_3522delinsGA];[3509_3520delinT]) (hg19) were introduced by CRISPR–Cas9 nuclease (pX330 hSpCas9) with an *RFP* reporter gene using Mirus TransIT LT1 transfection (Mirus Bio). Clonal populations were generated by flow sorting and propagating single RFP-positive cells. Mutations were verified by TOPO cloning and Sanger sequencing of both alleles for genotypes, protein expression by western blotting and absence of off-target effects by sequencing of the top hits. SNV positions from Sanger sequencing data were annotated with information from GENCODE v.19 (ref. [41]). Variant sequence and position was used to annotate variant calls with records from Clinvar 20200206_data_release[42] and COSMIC v. 91 (ref. [43]). Although multiple *BRCA2* homozygous loss-of-function alleles could be derived from 184-hTERT$^{p53-/-;BRCA2+/-}$ intermediates, only a single homozygous *BRCA1* allele was retrieved from the 119 clones of 184-hTERT$^{p53-/-,BRCA1+/-}$ that were screened, emphasizing that even with a *p53* deletion, full loss of *BRCA1* is initially negatively selected. OV2295 cells[44] were maintained in a 1:1 mix of Media 199 (Sigma-Aldrich) and MCDB 105 (Sigma-Aldrich) supplemented with 10% fetal bovine serum (FBS) under normoxic conditions. Cell lines were authenticated by short tandem repeat (STR) profiling and tested for mycoplasma by LabCorp.

### Immunoblotting

184-hTERT cells were lysed directly in 1× Laemmli buffer supplemented with 7.5% β-mercaptoethanol and proteins were denatured at 95 °C for 15 min. Protein from 250,000 cells was resolved on a 4–15% acrylamide gel (Biorad) or 3–8% Tris-acetate acrylamide gel (Novex) and transferred to a nitrocellulose membrane with Towbin transfer buffer overnight at 30 V at 4 °C. Blots were blocked with 5% milk in TBST for 1 h and incubated overnight at 4 °C with mouse anti-p53 (Santa Cruz SC-126, 1:500 in 5% bovine serum albumin (BSA)), mouse anti-BRCA1 (Santa Cruz SC-6954, 1:200 in 5% BSA), mouse anti-BRCA2 (Millipore OP95, 1:200 in 5% BSA) or goat anti-GAPDH (SC-48166, 1:500 in 5% BSA). Blots were washed five times for 5 min in TBST and incubated with anti-goat HRP-conjugated secondary antibody (Abcam ab6721, 1:5,000 in 5% BSA) for 1 h at room temperature, then washed five times for 5 min and the signal was imaged using Immobilon Western Chemiluminescent HRP Substrate (MilliporeSigma, WBKL20500) and the ImageQuant LAS 4000 (GE Healthcare) using the ImageQuant TL software.

### Verification of mutations in the 184-hTERT cell line

Genomic DNA was extracted from 184-hTERT cell lines and regions of interest of *BRCA1* or *BRCA2* were amplified by PCR. Amplicons were inserted into a pCR-TOPO vector and transformed into *Escherichia coli* using the TOPO TA cloning kit (Thermo Fisher Scientific). Colonies were selected, DNA purified by Purelink Quick Plasmid Miniprep kit (Thermo Fisher Scientific) and sequenced by Sanger sequencing to assess CRISPR-induced mutations.

### Acquisition of samples from patients and patient consent

Samples were acquired with informed consent, according to procedures approved by the Ethics Committees at the University of British Columbia. Patients with breast cancer undergoing diagnostic biopsy or surgery were recruited and samples were collected under protocols H06-00289 (BCCA-TTR-BREAST), H11-01887 (Neoadjuvant Xenograft Study), H18-01113 (Large-scale genomic analysis of human tumours) or H20-00170 (Linking clonal genomes to tumour evolution and therapeutics). HGSC samples were obtained from women undergoing debulking surgery under protocols H18-01652 and H18-01090. Banked HGSC and TNBC specimens were obtained at the Memorial Sloan Kettering Cancer Center following Institutional Review Board (IRB) approval and patient informed consent (protocols 15–200 (HGSC) and 18–376 (TNBC)). HGSC and TNBC clinical assignments were performed according to American Society of Clinical Oncology guidelines for ER, PR and HER2 positivity.

### Xenografting

Fragments of tumours from patients were chopped finely with scalpels and mechanically disaggregated for one minute using a Stomacher 80 Biomaster (Seward Limited) in 1 ml cold DMEM/F-12 with glucose, L-glutamine and HEPES (Lonza 12–719F). Two hundred microlitres of medium containing cells or organoids from the resulting suspension was used equally for transplantation in four mice. The remaining tissue fragments were cryopreserved viably in DMEM/F-12 supplemented with 47% FBS and 6% dimethyl sulfoxide (DMSO). Tumours were transplanted in mice as previously described (Eirew) in accordance with SOP BCCRC 009. Female NOD/SCID/IL2rγ$^{-/-}$ (NSG) and NOD/Rag1$^{-/-}$Il2rγ$^{-/-}$ (NRG) mice were bred and housed at the Animal Resource Centre (ARC) at the British Columbia Cancer Research Centre. For subcutaneous transplants, mechanically disaggregated cells and clumps of cells were resuspended in 150–200 µl of a 1:1 v/v mixture of cold DMEM/F-12:Matrigel (BD Biosciences). Female mice (8–12 weeks old) were anaesthetized with isoflurane and the mechanically disaggregated cell clump suspension was transplanted under the skin on the left flank using a 1-ml syringe and 21-gauge needle. Mice were housed at a 18–25 °C temperature range and 20–70% humidity range, with a 12-h daylight cycle (on at 06:00; off at 18:00). All animal experimental work was approved by the animal care committee (ACC) and animal welfare and ethical review committee at the University of British Columbia (UBC) under protocol A19-0298.

### Tissue processing

Xenograft-bearing mice were euthanized when the size of the tumours approached 1,000 mm$^3$ in volume, according to the limits of the experimental protocol. The tumour material was excised aseptically and processed as described for primary tumour. A section of tumour was fixed in 10% buffered formalin for 24 h, dehydrated in 70% ethanol and paraffin-embedded before duplicate 1-mm cores were used to generate tissue microarrays for staining and pathological review. Remaining tumour was finely chopped and gently paddle-blended, and released single cells and fragments were viably frozen in DMEM supplemented with 47% FBS and 6% DMSO.

### Histopathology of PDX tumours

Deparaffinized 4-µm sections of tissue microarrays were stained with haematoxylin and eosin or KRAS (Lifespan Bioscience, LS-B4683, 1:50), performed using the Ventana Discovery XT platform and the UltraMap DAB detection kit. HGSC pathology was confirmed by an anatomical pathology resident at University of British Columbia, under the supervision of a certified staff pathologist.

### WGS

Genomic DNA was extracted from frozen tissue fragments using the DNeasy Blood and Tissue kit (Qiagen) and constructed libraries for whole genomes of 309 tumour–normal pairs were sequenced on the Illumina HiSeqX according to Illumina protocols, generating 100-bp paired-end reads for an estimated coverage of sequencing between 40× (normal) and 80× (tumour). Sequenced reads were aligned to the human reference GRCh37 (hg19) using BWA-MEM.

## Long-read sequencing

High-molecular weight (HMW) DNA was extracted from fresh and/or frozen tissue fragments using the MagAttract HMW DNA Kit (Qiagen) and size-selected using Blue Pippin for single long-molecule sequencing on the PromethION (Oxford Nanopore Technologies).

## Generation of single-cell suspensions and nuclei for scDNA-seq

Viably frozen aliquots of patient tissues and PDX tumours were thawed and either homogenized and lysed using Nuclei EZ Buffer (Sigma) or enzymatically dissociated using a collagenase/hyaluronidase 1:10 (10×) enzyme mix (STEMCELL Technologies), as described previously[3,18]. Cells and nuclei were stained with CellTrace CFSE (Life Technologies) and LIVE/DEAD Fixable Red Dead Cell Stain (Thermo Fisher Scientific) in a 0.04% BSA/PBS (Miltenyi Biotec 130-091-376) incubated at 37 °C for 20 min. Cells were pelleted and resuspended in 0.04% BSA/PBS. This single-cell suspension was loaded into a contactless piezoelectric dispenser (Cellenone or sciFLEXARRAYER S3, Scienion) and spotted into open nanowell arrays (SmartChip, TakaraBio) preprinted with unique dual index sequencing primer pairs. Occupancy and cell state were confirmed by fluorescent imaging and wells were selected for single-cell copy number profiling using the DLP+ method[3]. In brief, cell dispensing was followed by enzymatic and heat lysis. After cell lysis, tagmentation mix (14.335 nl TD buffer, 3.5 nl TDE1 and 0.165 nl 10% Tween-20) in PCR water was dispensed into each well followed by incubation and neutralization. For *BRCA1*[+/−] cells, the tagmentation mix consisted of 10 nl TB1 buffer and 10 nl BT1 enzyme without Tween-20 in PCR water. Final recovery and purification of single-cell libraries was done after eight cycles of PCR. Pooled single-cell libraries were analysed using the Agilent Bioanalyzer 2100 HS kit. Libraries were sequenced at the UBC Biomedical Research Centre on the Illumina NextSeq 550 (mid- or high-output, paired-end 150-bp reads), or at the Genome Sciences Centre on the Illumina HiSeq2500 (paired-end 125-bp reads) and Illumina HiSeqX (paired-end 150-bp reads). The data were then processed through a quantification and statistical analysis pipeline[3].

## Generation of 10X scRNA-seq data

The 184-hTERT cells were pelleted and gently resuspended in 200 µl PBS followed by 800 µl 100% methanol and incubation at −20 °C for 30 min to fix, dehydrate and shrink cells. PDX tumour fragments were dissociated into single cells using collagenase/hyaluronidase at 37 °C for 2 h for TNBC tumours or with cold active *Bacillus lichenformis* (Creative Enzymes NATE0633) in PBS supplemented with 5 mM CaCl$_2$ and 125 U ml$^{-1}$ DNAse for HGSC tumours, as described previously[45] with additional mechanical dissociation using a gentleMACS dissociator (Miltenyi Biotec). Cells were then pelleted and resuspended in 0.04% BSA/PBS and immediately loaded onto a 10X Genomics Chromium single-cell controller targeting 3,000 cells for recovery. Libraries were prepared according to the 10X Genomics Single Cell 3′ Reagent kit standard protocol. Libraries were then sequenced on an Illumina Nextseq500/550 with 42-bp paired-end reads, or a HiSeq2500 v4 with 125-bp paired-end reads. 10X Genomics Cell Ranger 3.0.2 was used to perform demultiplexing, counting and alignment to GRCh38 and mm10.

## Processing of bulk whole-genome data

SNV and SV calls for 121 HGSC samples were acquired from a previous study[12]. For new samples, reads were aligned to the hg19 reference genomes using BWA-MEM. Processing proceeded as per the aforementioned study[12] to maintain consistency.

SNVs were called with MutationSeq[46] (probability threshold = 0.9) and Strelka[47]. The intersection of calls from these methods were retained; however, SNVs falling in blacklist regions were removed. The blacklist regions include the UCSC Genome Browser Duke and DAC blacklists, and those in the CRG Alignability 36mer track that had

more than two mismatched nucleotides. SNVs were then annotated with OncoKB[48] for variant impact.

SVs were called using deStruct[49] and LUMPY[50], and breakpoints called by both methods were retained. We then filtered events with the following criteria: any breakpoints falling in the blacklists described above; ≤30-bp inter-breakpoint distance; <1,000-bp deletion; any breakpoints with fewer than 5 supporting reads in the tumour sample or any read support in the matched normal sample.

Gene mutation enrichment analysis was performed using the hypergeometric test for SNVs, amplifications and deep deletions separately, comparing each signature stratum to all other samples.

## Nanopore data analysis

For Nanopore sequence data, base calling and read alignment were performed using Guppy v.3 and Minimap2, respectively[51,52]. Reads that were likely to be derived from mouse were filtered by first aligning to a concatenated hg19 and mm10 reference, removing reads with alignments to mm10 and re-aligning the remaining reads to hg19. Signal artefact regions as well as alignments with mapping quality of less than 60 were excluded from the final alignments. Alignments were then phased using the PEPPER-Margin-DeepVariant pipeline, after which WhatsHap was used to tag reads in the filtered alignments using phasing information[53,54]. SV calling was performed using Sniffles (v.1.0.12) and cuteSV (v.1.0.11) with 5 read support, and subsequently merged using SURVIVOR for a union set of predicted variants[55,56]. Alignments and variants were visualized using IGV, Ribbon and the karyoploteR R package[57–59].

## HGSC and TNBC meta-cohort signature analysis

Signature analysis was performed according to a previous study[10]. The MMCTM model was run on the sample SNV and SV count matrices. The number of signatures to estimate in the HGSC and TNBC integrated cohort was chosen by running the above fitting procedure for $k = 2$–16 for both SNV and SV signatures with the number of restarts set to 500, in which $k$ is the number of signatures. We performed this step on approximately half the mutations in each sample, then computed the average per-mutation log likelihood on the other held-out half of the mutations. The elbow curve method on log-likelihood values was used to select the final number of signatures to fit to the entire dataset.

To estimate MMCTM parameters on the full dataset, $\alpha$ hyperparameters were set to 0.1. The model was initially fit to the data 1,500 times. Each restart was run for a maximum of 1,000 iterations or until the relative difference in predictive log likelihood on the training data was less than $10^{-4}$ between iterations. The restarts with the best predictive log likelihoods for SNVs and SVs were selected as seeds for the final fitting step. The model was again fit to the data 1,500 times. The model parameters for each restart were set to the parameters of the optimal models from the previous step described above, then run for a maximum of 1,000 iterations or until the relative difference in predictive log likelihood on the training data was less than $10^{-5}$ between iterations. The restart with the best mean rank of the SNV and SV predictive log likelihoods from this round was selected as the final model.

MMCTM estimated SNV signatures were matched to COSMIC signatures by solving the linear sum assignment problem for cosine distances between the MMCTM and COSMIC signatures v.3 (minus tobacco smoking-associated COSMIC SBS4) using the clue R package[60].

Samples were clustered by first applying UMAP[61] to the normalized signature probabilities for the HRD SNV signature and all SV signatures with n_neighbors = 20 and min_dist = 0 to produce two-dimensional sample embeddings. Next, HDBSCAN[62] was run on the sample embeddings with min_samples = 5, min_cluster_size = 5 and cluster_selection_epsilon = 0.75 to produce the sample clusters (strata).

## Survival analysis

For each patient, the number of days between diagnosis and death or last follow-up was collected. Patients were segregated into groups,

and a Kaplan–Meier curve was fitted for each group. Each cancer type was analysed separately and in two distinct grouping schemes. First, patients were split into HRD and 'Other' groups, in which the HRD group included patients whose cancers were identified as being in either the HRD-Dup or HRD-Del groups, and the 'Other' group included all other patients. Next, patients were grouped according to their assigned signature types: HRD-Dup, HRD-Del, TD or FBI.

## DLP+ WGS quantification and analysis
Single-cell copy number, SNV and SV calls were generated using a previously described approach[3], except that BWA-MEM[63] was used to align DLP+ reads to the hg19 reference genome. The genome was segregated into 500-kb bins, and GC-corrected read counts were calculated for each bin. These read counts were then input into HMMCopy[64] to produce integer copy number states for each bin.

## DLP+ data filtering
Cells were retained for further analysis if the cell quality was at least 0.75 (ref. [3]), and they passed both the S-phase and the contamination filters. The contamination filter uses FastQ Screen[65] to tag reads as matching human, mouse or salmon genomes. If more than 5% of reads in a cell are tagged as matching the mouse or salmon genomes, then the cell is flagged as contaminated. The S-phase filter uses the cell-cycle state Random Forest classifier from ref. [3] and removes cells for which S-phase is the most probable state. The HGSC and TNBC cells were also filtered to remove small numbers of contaminating diploid cells.

Finally, cell filtering was performed to remove putative early and late S-phase cells that passed the initial S-phase filter. This involved two steps: first, building a cell phylogeny with sitka[37] and manually identifying the minimal phylogeny branches in which the cycling cells have been clustered. The cells in these branches were then removed. Next, clustering cells according to their copy number profiles and removing manually identified clusters of cycling cells.

We removed potentially problematic genome bins from our copy number results that had a mappability score of 0.99 or below, or that were contained in the ENCODE hg19 blacklist[66].

To detect SNVs and SVs in each dataset, reads from all cells in a DLP+ library were merged to form 'pseudobulk' libraries. SNV calling was performed on these libraries individually using MutationSeq[46] (probability threshold = 0.9) and Strelka (score > 20) (ref. [47]). Only SNVs that were detected by both methods were retained. For each dataset, the union of SNVs was aggregated, then for each cell and each SNV, the sequencing reads of that cell were searched for evidence of that SNV. SV calling was performed in a similar manner, by forming pseudobulk libraries, then running LUMPY[50] and deStruct[49] on each pseudobulk library, and retaining events that were detected by both methods. LUMPY and deStruct predictions were considered matched if the breakpoints matched in orientation and the positions involved were each no more than 200 nucleotides apart on the genome. Only deStruct predictions with a matching LUMPY prediction were retained. Sparse per-cell breakpoint read counts were extracted from deStruct using the cell identity of read evidence for each predicted breakpoint. SNV and SV calls were further post-processed according to a previous study[12]. When performing pseudobulk analysis on groups of cells, a breakpoint is considered present in a clone if at least one cell that constitutes the clone contains evidence of the breakpoint. A subsampling experiment determined that this approach has 80% power to recover breakpoints at a cumulative coverage of 5× (100–150 cells) (see Supplementary Note).

## Analysis of mutation signatures in DLP+ data
Mutation signature probabilities were fit to DLP+ pseudobulk-derived SNV and SV counts for each patient using the MMCTM method and pre-computed mutation signatures from the HGSC and TNBC meta-cohort. Inference was performed as per the bulk sequencing

data, until the relative difference in predictive log likelihood was < 10⁻⁶ between iterations.

## Identifying clones in DLP+ WGS by clustering copy number profiles
For most datasets, clones were detected by first using UMAP on per-cell GC-corrected read count profiles, producing a two-dimensional embedding of the cell profiles. We then ran HDBSCAN on the two-dimensional embedding from UMAP to detect clusters of cells with similar copy number profiles.

UMAP was run with min_dist = 0.0, and metric = "correlation", whereas HDBSCAN was run with approx_min_span_tree = False, cluster_selection_epsilon = 0.2, and gen_min_span_tree = True. Dataset specific UMAP and HDBSCAN parameter settings are listed in Supplementary Table 3.

## Calculating cell ploidy
Cell ploidy was calculated by taking the most common copy number state. Copy number states were those determined by HMMCopy.

## Identifying missegregated chromosomes
The approach taken to identify putative chromosome missegregation events is similar to a previous one[3]. Cells were split into groups corresponding to their clones. Clone copy number profiles were generated for each clone. Cells with ploidy not equal to the clone consensus profile were normalized to match the clone ploidy. Cell copy number profiles were compared to the clone copy number profile for the matching clone to which the cell belongs. The result was assignment of an offset value for each genomic bin in each cell, that represented the copy number difference between the cell and the clone-level consensus profile. For each chromosome in each cell, if a particular copy number difference (that is, −1, 1, and so on) represented at least 75% of the chromosome, then we labelled that chromosome as having a missegregation event.

## Identifying CNA segments
Gain and loss segments in each cell were found by comparing the copy number state in each 500-kb bin to that cell's ploidy. A copy number higher than ploidy was labelled as a gain, and a copy number lower than ploidy was labelled as a loss. Gain and loss segments are a set of consecutive bins with the same gain–loss label. Segments ≤ 1.5 × 10⁶ bp were excluded to reduce segments potentially resulting from noise in the HMMCopy copy number states.

## Computing serriform variability scores in CNA breakpoints
For each dataset, consensus copy number profiles were generated for each clone. Copy number segments were identified as above for each consensus profile. Copy number segments were then identified for single-cell copy number profiles. The copy number profiles of each cell were normalized so that the adjusted cell ploidy matched the ploidy of the clone to which the cell belonged using the following formula: *cell_state = cell_state/cell_ploidy × clone_ploidy*

Cell copy number segments were matched to segments in the clone copy number profile as follows: for each segment in the clone copy number profile, inspect the copy number states of the adjacent segments. If the segment state was less than both adjacent states, then only cell segments whose state was less than both of the two adjacent clone segment states could be matched to that segment. If the clone segment state was higher than both adjacent states, only cell segments whose state was higher than both of the adjacent clone segment states could be matched to that segment. If the clone segment state was in between the two adjacent states, only cell segments whose state was in between the two adjacent clone segment states could be matched to that segment. Finally, each cell segment was matched to the compatible clone segment that it overlapped the most, in which compatibility

means that the cell segment state met the criteria described above and the cell belonged to the relevant clone.

Next, clone segment breakpoints were aggregated across all clones. For each breakpoint, matched cell breakpoints were identified. Stable cell breakpoints (that is, those cell breakpoints that matched the clone-level breakpoint position) and unstable cell breakpoints (all other cell breakpoints) were queried for their raw GC-corrected read count values up to five bins to the left and right of the breakpoint position. Breakpoint noise values were computed as the mean absolute value of the difference between these values and the integer copy number state inferred by HMMCopy. For each clone-level breakpoint event, cells were removed if their breakpoint noise values were higher than a threshold value, which was computed as the mean noise value of the stable cell breakpoints.

Serration scores for each event were calculated by first computing the frequency of cell-specific breakpoint positions. Each cell breakpoint position was considered 'rare' if it occurred in less than 5% of cells with the considered event. The final serration score was computed as the fraction of event cells whose breakpoint position was considered 'rare'.

For comparing serration rates between cases, breakpoints with at least 100 cells, and whose adjacent copy number segments were in total at least 20 Mbp (40 genomic bins) were retained. This was done to retain only those breakpoints for which serration could be reliably computed. As a result, SA605 was not included in comparisons as this case had fewer than 100 cells. A zero-inflated generalized linear mixed model with a beta response that accounted for case-specific and signature-type effects was fit to determine the effect of mutation signature type on serration scores.

## Comparison of HLAMP copy number variance

HLAMPs were identified by first selecting 500-kb genomic bins in which at least 10 cells have a raw copy number (adjusted per-bin read counts) of at least 10. Copy number variance for each bin was calculated using the raw copy number that was adjusted for cell ploidy and cell clone by first dividing the copy number by the cell ploidy, then subtracting the mean clone raw copy number. The cell ploidy is the most common HMMCopy copy number state as described above, and the mean clone copy number is computed for each bin in each clone across all cells in that clone. Mean HLAMP copy number variance was calculated for each dataset across all HLAMP bins, and these values were compared between signature type dataset groups.

## Clustering HLAMP genomic features

To explore plausible mechanistic origins of oncogenic HLAMPs we extracted genomic features proximal to the locus of interest. We took a region 15 Mbp either side of the locus of interest and pulled out copy number and SV features. We extracted the following features: entropy of haplotype-specific states; total number of SVs identified; proportion of SVs of each type (fold-back inversions, duplications, deletions and translocations); number of chromosomes involved in translocations; ratio of copy numbers between the bin containing the oncogene and the average copy number across the chromosome; average copy number state; average size of segments; average number of segments; and average minor allele copy number. All averages are across cells. We then performed hierarchical clustering on a scaled matrix of all features, using the silhouette width to determine the appropriate number of clusters.

## HSCN analysis

See the Supplementary Note for a detailed discussion of our method, SIGNALS, for HSCN analysis. This includes validation of the method and benchmarking against other methods. In brief, SIGNALS uses haplotype blocks genotyped in single cells and implements an hidden Markov model (HMM) based on a Beta-Binomial likelihood to infer the most probable haplotype-specific state. We used default parameters for all datasets apart from SA1292, in which we increased the self transition probability from 0.95 to 0.999 to mitigate against the noisier copy number data in this sample.

## Pseudobulk HSCN profiles

In numerous places in this study we construct 'pseudobulk' haplotype-specific or total copy number profiles either across all cells in a sample or subsets of cells that share some features of interest. To do this, we group the cells of interest and then compute an average profile by taking the median values of copy number and BAF and the mode of the haplotype-specific state. The function 'consensuscopynumber' provided in SIGNALS was used for this.

## Comparing segmentation profiles across cells

To facilitate comparisons of genomic profiles across cells, we inferred a set of disjoint segments from the consensus copy number profiles of clusters. For each clone or cluster we generated a consensus segmentation profile, and then used the 'disjoin_ranges' function from ply-ranges[67] to generate a non-overlapping disjoint segmentation profile. Each segment was then genotyped in each cell by taking a consensus across the bins within each segment, producing a consistent set of genomic segments and states that could be compared across cells.

## Identification of parallel copy number events

The set of genotyped disjoint segmentation profiles was used to calculate the number of parallel copy number CNAs. Parallel CNAs were defined as genomic regions greater than 4 Mbp in which gain or loss of both the maternal and paternal haplotype was observed in more than 1% of cells. Copy number breakpoints of segments do not need to match to be included.

## Phylogenetic analysis

We computed phylogenetic trees using sitka as previously described[37], using the consensus tree from the posterior distribution for downstream analysis. For visualization, clades with a high fraction of singletons (nodes with a single descendant) were removed. To remove nodes, nodes were ordered by the fraction of descendants that were singletons, and nodes were removed iteratively until a maximum of 3% of cells in the tree were removed. Trees were visualized using ggtree[68] and functionality in SIGNALS. Phylogenetic distances were computed as the mean pairwise distances between phylogenetic tips (cells) using the cophenetic function in APE[69]. Distances represent the number of copy number change points between two cells on the phylogeny.

## Event rates inferred from single-cell phylogenies

To compute the rates of gains and losses of whole chromosomes, chromosome arms and segmental aneuploidies we enumerated the number of events from our single-cell phylogenies using parsimony-based ancestral state reconstruction. We used the genotyped disjoint segmentation profiles for this.

We first defined states for each segment in each cell relative to the most common state across all cells. For each segment, cells can have one of two possible states for each class of interest: (gain, not gained), (loss, not lost). By casting the problem as reconstructing the ancestral states within the phylogeny, we can then compute the number of transitions between these states that most parsimoniously explains the phylogenetic tree. We used a simple transition matrix in which transitions between states incur a cost of 1. Ancestral state reconstruction then amounts to finding the reconstruction that minimizes this cost. The event frequency per sample is then calculated by dividing the parsimony score (number of events) by the number of cells. We used castor v.1.6.6 in R to perform the ancestral state reconstruction[70]. The unit of this quantity is the number of events per cell division, assuming no cell death. It is possible (perhaps likely) that many cells get segmental gains or losses but then die, we never sample such cells and our phylogenetic tree reconstructs ancestral relationships between cells that

survive and that we sample. It is challenging to decouple the death rate of cells from the true event rate per cell division[71]; thus, our event rate is an effective event rate; that is, the event rate scaled by the (unknown) death rate of cells. To contrast the rates across different types of events, we classified segments as whole chromosomes, chromosome arms or segmental aneuploidies.

## Calculation of copy number distance

The copy number distance calculates the number segments that need to be modified to transform one copy number profile into another[20]. We use this measure to compute cell-to-cell variation in our dataset. To compute this measure, we modified the code provided in a previous study[25] to take into account whole-genome doubling of cells (https://github.com/raphael-group/WCND). We did this as follows: given two copy number profiles (integer copy number states of individual haplotypes in bins across the genome) $CNP_A$ and $CNP_B$, we computed the following distances:

$$d_1 = f(CNP_A, CNP_B)$$

$$d_2 = f(2 \times CNP_A, CNP_B)$$

$$d_3 = f(CNP_A, 2 \times CNP_B)$$

in which 2× refers to doubling the copy number state across the whole genome. We then took the copy number distance to be

$$d = \min(d_1, d_2, d_3).$$

If the minimum was $d_2$ or $d_3$, we increased $d$ by 1 (that is, counting WGD as an additional event). Calculating all pairwise comparisons is computationally expensive, so for each dataset we subsampled 250 cells and calculated all pairwise distances for these 250 cells.

## 10X scRNA-seq processing

CellRanger software (v.3.1.0) was used to perform read alignment, barcode filtering and quantification of unique molecular identifiers (UMIs) using the 10X GRCh38 transcriptome (v.3.0.0) for FASTQ inputs. CellRanger filtered matrices were loaded into individual Seurat objects using the Seurat R package (v.4.1.0) (refs. [72,73]). The resulting gene-by-cell matrix was normalized and scaled for each sample. Cells retained for analysis had a minimum of 500 expressed genes and 1,000 UMI counts and less than 25% mitochondrial gene expression. Genes expressed in fewer than three cells were removed. Cell-cycle phase was assigned using the Seurat[73] CellCycleScoring function. Scrublet[74] (v.0.2.3) was used to calculate and filter cells predicted to be doublets. We then applied the standard Seurat processing pipeline using default parameters apart from using the first 20 principal component analysis (PCA) dimensions for nearest neighbour and UMAP calculations.

## Allelic imbalance in scRNA-seq

We called heterozygous SNPs in the scRNA-seq data using cellSNP v.1.2.2 (ref. [75]). As input, we used the same set of heterozygous SNPs identified in the scDNA-seq and the corresponding normal sample for each sample. The liftover script provided in cellSNP was used to lift over SNPs from hg19 to hg38. After genotyping, we phased the SNPs using the phasing information computed from the haplotype-specific inference in the scDNA-seq. As SNP counts are much more sparse in scRNA-seq than in scDNA-seq (around two orders of magnitude lower), we aggregated counts across segments (minimum size = 10 Mbp), computing the BAF for each segment. We then generated a cell by segment BAF matrix and incorporated this into our gene expression Seurat objects. We applied an additional filtering criterion here, removing cells with fewer than 200 SNP counts. Functionality to map scDNA-seq to scRNA-seq and call allelic imbalance is provided in SIGNALS.

## Differential expression analysis

Differential expression analysis between gene expression clusters was computed using the Wilcoxon rank sum test with the presto R package. Gene expression clusters were computed using the FindClusters function in Seurat. Only cells in G1 phase were included. To compare gene expression variability for oncogenes, we took the absolute maximum log-transformed fold change for each sample for each oncogene and contrasted this value in cases in which oncogene copy number was determined to be fixed or variable from DLP+ single-cell sequencing of the same samples. 'Variable' oncogenes were defined as those that had a minimum ratio of 2 between the maximum to minimum clone-level copy number, and 'non-variable' oncogenes as those that had a ratio of less than 2.

## Nearest neighbour gene expression analysis

To assess transcriptional convergence of losses of alleles we made use of the shared nearest neighbour graph computed using Seurat. This was done for chr. 2q in sample SA906b. For a given cell, an enrichment score was defined as the observed fraction of nearest neighbours divided by the expected fraction of nearest neighbours. Here, the expected fraction of neighbours with the same allelic state was defined as the global fraction of cells in each state. Hence, a positive enrichment score indicates an overrepresentation of cells in the allelic state of interest among its nearest neighbours, a negative score indicates an underrepresentation and a score of 0 would reflect a perfectly mixed neighbourhood of cells with different allelic states. To mitigate the influence of other technical or biological variability, for this analysis we only included cells in G1 phase, and removed cells with greater than 7.5% mitochondrial gene expression as we found that this was variable between gene expression clusters.

## Statistical tests

The statistical tests used were two-tailed unequal-variance $t$-tests unless otherwise specified: log-rank tests were used for comparing survival curves; Wilcoxon rank sum two-tailed tests were used for comparing segment lengths, segment counts, missegregations and ploidy percentages, copy variances, bin counts, gene copy number distributions, gene expression log-transformed fold changes, parallel copy number counts and breakpoint counts; and hypergeometric tests were used to identify enrichment of gene mutations. $P$ values from multiple comparisons were corrected using the Benjamini–Hochberg method[76].

## Box plot statistics

All box plots indicate the median, first and third quartiles (hinges), and the most extreme data points no farther than 1.5× the IQR from the hinge (whiskers).

## Reporting summary

Further information on research design is available in the Nature Research Reporting Summary linked to this article.

## Data availability

All data are available for general research use. Processed data including somatic mutation data for bulk WGS, total (and allele-specific) copy number profiles for DLP+ data and filtered count matrices for scRNA-seq data are available for download at https://zenodo.org/record/6998936. Raw scRNA-seq data are available for download at https://ega-archive.org/studies/EGAS00001006343. Raw single-cell sequencing data generated for this study are available from https://ega-archive.org/studies/EGAS00001006343, and previously published single-cell sequencing data used in this study are available at https://ega-archive.org/studies/EGAS00001004448 and https://ega-archive.org/studies/EGAS00001003190. Somatic mutation calls from bulk

WGS for 16 patients with TNBC for whom the IRB consent does not include public deposition of raw sequencing data are available at http://www.ncbi.nlm.nih.gov/projects/gap/cgi-bin/study.cgi?study_id=phs003038.v1.p1, and raw sequencing data can be provided upon request under material transfer agreement to shahs3@mskcc.org. Bulk WGS BAM files from patients under IRB consent protocols for public release of raw data are available for download at https://ega-archive.org/studies/EGAS00001006343, http://www.ncbi.nlm.nih.gov/projects/gap/cgi-bin/study.cgi?study_id=phs003036.v1.p1 and https://ega-archive.org/datasets/EGAD00001003268 (for previously published data[12]) or by request under material transfer agreement to shahs3@mskcc.org and saparicio@bccrc.ca.

## Code availability

MMCTM method: https://github.com/shahcompbio/MultiModal-MuSig.jl. DLP+ single-cell WGS pipeline: https://github.com/shahcompbio/single_cell_pipeline. Bulk WGS pipeline: https://github.com/shahcompbio/wgs. SIGNALS processing pipeline: https://github.com/marcjwilliams1/hscn_pipeline. SIGNALS: https://github.com/shahcompbio/signals; v.0.7.2 archived at https://doi.org/10.5281/zenodo.6642342.

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

**Acknowledgements** This project was supported by the BC Cancer Foundation at BC Cancer and Cycle for Survival supporting the Memorial Sloan Kettering Cancer Center. S.P.S. holds the Nicholls Biondi Chair in Computational Oncology and is a Susan G. Komen Scholar (GC233085). S.A. holds the Nan and Lorraine Robertson Chair in Breast Cancer and is a Canada Research Chair in Molecular Oncology (950–230610). Additional funding was provided by a Terry Fox Research Institute grant (1082), a Canadian Cancer Society Research Institute Impact program grant (705617), a CIHR grant (FDN-148429), Breast Cancer Research Foundation awards (BCRF-18-180, BCRF-19-180 and BCRF-20-180), a MSK Cancer Center Support Grant/Core Grant (P30 CA008748), National Institutes of Health grants (1RM1 HG011014-01 and P50 CA247749-01), a CCSRI grant (705636), the Cancer Research UK Grand Challenge Program and the Canada Foundation for Innovation (40044) to S.A. and S.P.S. M.J.W. is supported by a National Cancer Institute Pathway to Independence award (K99CA256508).

**Author contributions** S.P.S. and S.A.: project conception and oversight. T.F., C.H.O. and M.J.W.: led implementation of the study and performed all experiments and data analysis. A.M., S.M., S.S., I.V.-G., E.L., H.S., S.B., D.L., J.P., D.G., D.A., E.H., S.L., V.B., F.U., N.G., A.C.W., E.Z. and J.M.D.: computational biology and statistical analysis. F.K., H.L., T.M., P.E., D.Y., A.W.Z., J.L.P.L., B.W., J. Brimhall, J. Biele, J.T., V.A., M.V.V., Y.F.L., H.X., T.R.d.A., S.R.L., J.N.M., D.G.H., Y.L. and V.C.L.: single-cell sequencing, in vitro experiments, PDX generation and sample processing. S.D.M.: histopathology analysis. R.A.M., D.Z., B.W., S.H.K., A.D.C.P. and J.S.R.-F.: bulk WGS and sample procurement. N.R., S.P.S., S.A., T.F., C.H.O. and M.J.W.: wrote the manuscript.

**Competing interests** B.W. reports ad hoc membership of the advisory board of Repare Therapeutics, outside the scope of this study. J.S.R.-F. reports receiving personal or consultancy fees from Goldman Sachs, REPARE Therapeutics, Paige.AI and Eli Lilly, membership of the scientific advisory boards of VolitionRx, REPARE Therapeutics and Paige.AI, membership of the Board of Directors of Grupo Oncoclinicas and ad hoc membership of the scientific advisory boards of Roche Tissue Diagnostics, Ventana Medical Systems, Novartis, Genentech and InVicro, outside the scope of this study. D.G.H is the Chief Medical officer of Imagia Canexia Health, outside the scope of this study. S.P.S. and S.A. are shareholders and consultants of Canexia Health and shareholders of Imagia Canexia Health, outside the scope of this study The remaining authors declare no competing interests.

**Additional information**
**Correspondence and requests for materials** should be addressed to Marc J. Williams, Sohrab P. Shah or Samuel Aparicio.

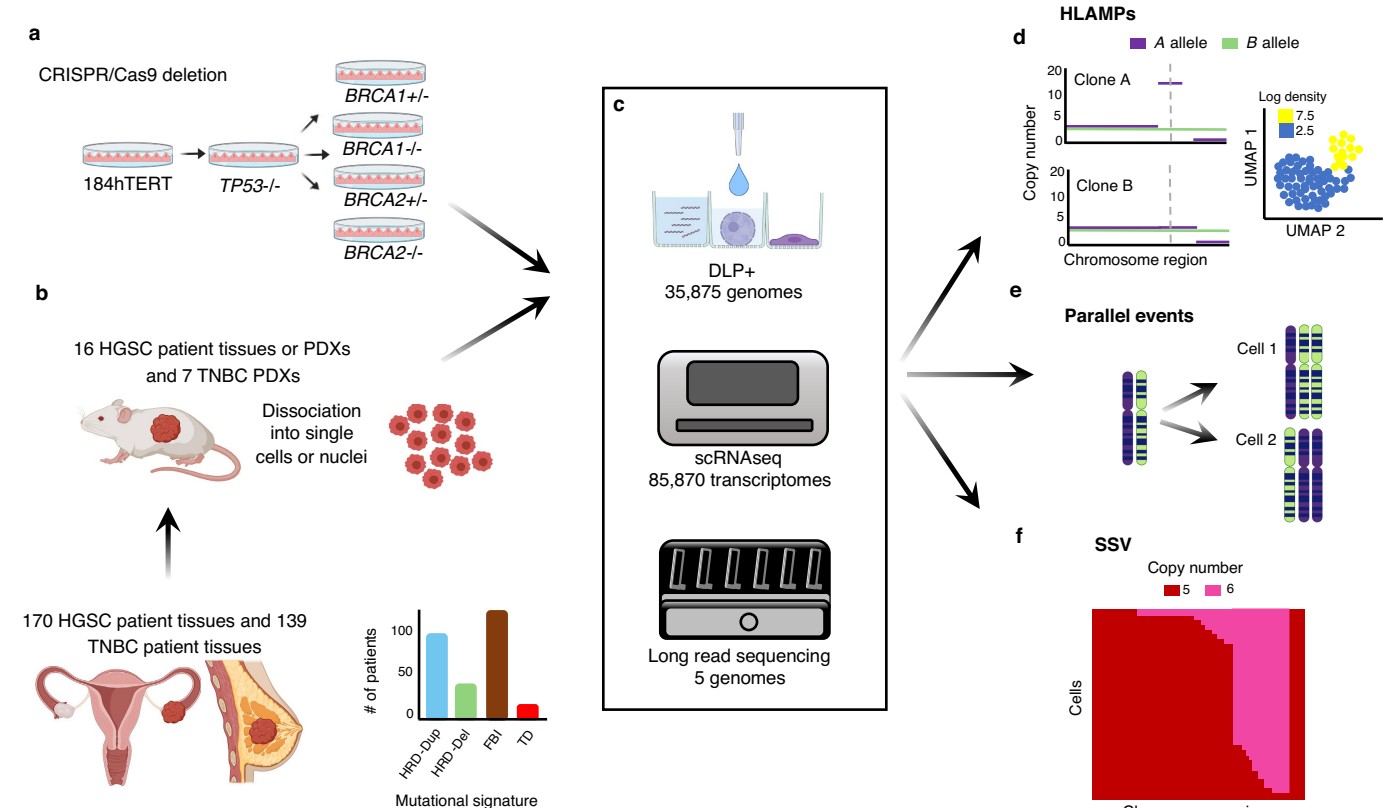

**Extended Data Fig. 1 | Study overview.** Experimental and cohort design. Single-cell genomes, transcriptomes and long-read sequencing libraries were generated from isogenic 184-hTERT cell lines (WT, or with *TP53*, *BRCA1* or *BRCA2* mutations) a) or PDX tissue from patients with TNBC and patients with HGSC from a meta-cohort with assigned SV or SNV mutational signatures b).

Single-cell and long-read sequencing (c) was used to examine mutational processes and haplotype-specific genomic diversity, including HLAMPs or rearrangements d), parallel events e) and SSVs f) at single-cell resolution and within clonal and subclonal populations. Generated using Biorender.com.

**a**

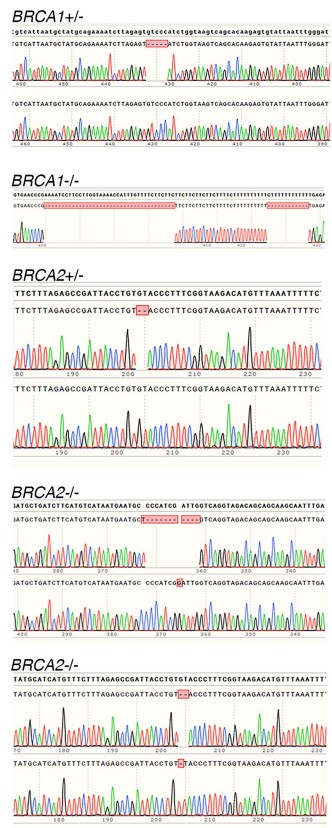

*BRCA1+/-*
gtcattaatgctatgcagaaaatcttagagtgtcccatctggtaagtcagcacaagagtgtattaatttgggat

*BRCA1-/-*

*BRCA2+/-*
TTCTTTAGAGCCGATTACCTGTGTACCCTTTCGGGTAAGACATGTTTAAATTTTTC

*BRCA2-/-*

*BRCA2-/-*

**b**

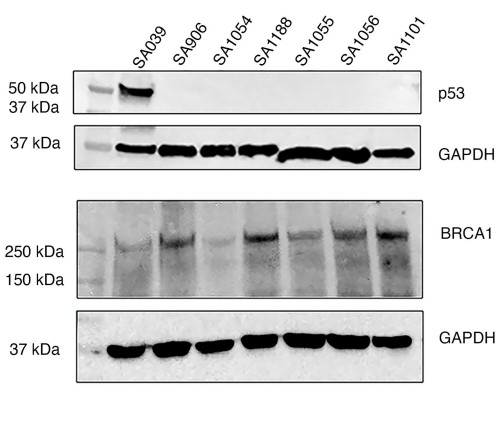

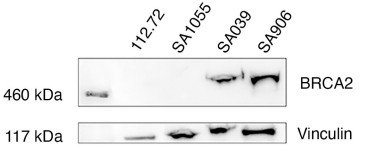

**c**

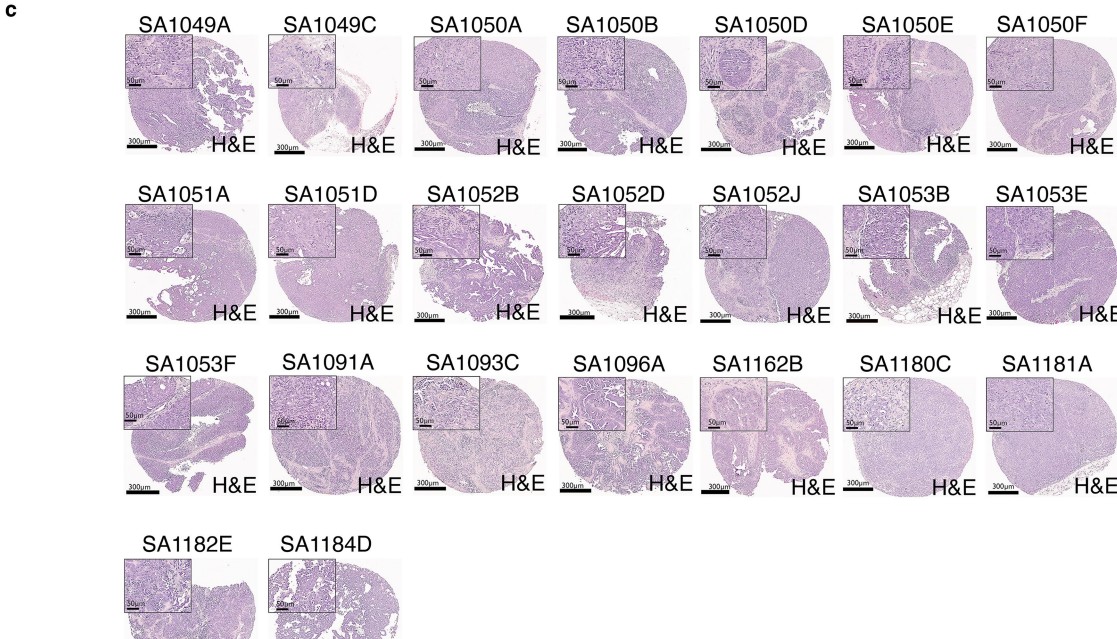

**Extended Data Fig. 2 | Sanger sequencing of cell lines and tumour histology. a,b)** Verification of CRISPR–Cas9 induced genotypes of 184-hTERT cell lines. **a)** Sanger sequencing of TOPO cloned *BRCA1* and *BRCA2* regions. b) Western blotting for p53, BRCA1 and BRCA2 proteins for 184-hTERT cell lines and including an additional *BRCA2*−/− clone, 112.72 and *TP53*−/− clone, SA1101. GAPDH and vinculin loading controls were performed on the same blot as p53, BRCA1 or BRCA2 probes. Blots shown are representative of n = 3 (WT), n = 3 (BRCA1) and n = 6 (BRCA2) independent experiments. For source blots, refer to Supplementary Fig. 1. **c)** Histology of HGSC PDXs in the dataset. Scale bars 300 μm and 50 μm as indicated. Images are representative of two cores stained from each PDX tissue.

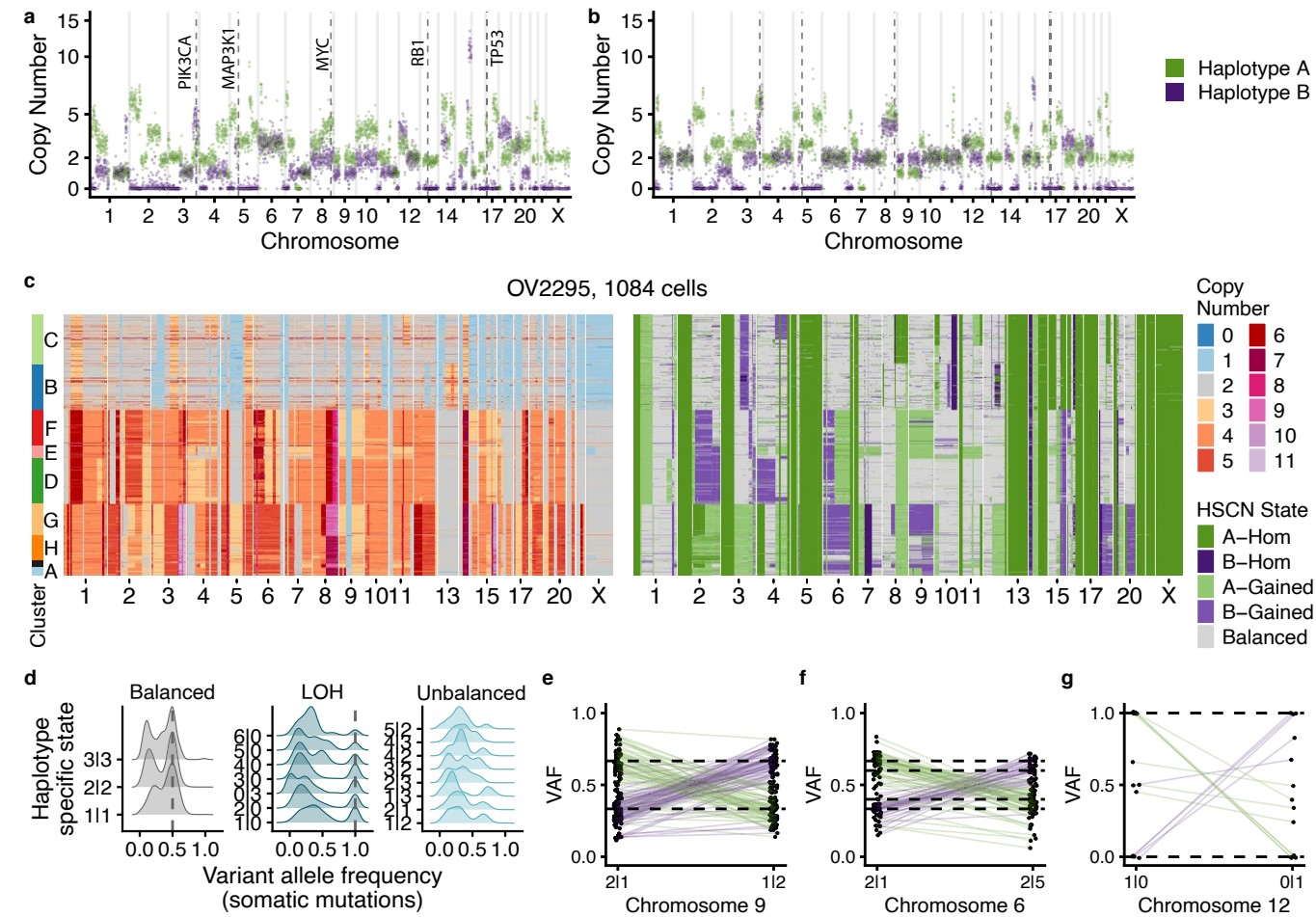

**Extended Data Fig. 3 | Validation of the SIGNALs method in an ovarian cancer cell line. a,b**) HSCN from 2 individual cells from the OV2295 cell line. **c**) Total copy number heat maps and HSCN heat map for 1084 cells. Each row is an individual cell. Rows are ordered according to a UMAP + HDBSCAN clustering, with clusters annotated on the left hand side. **d**) Distribution of VAFs as a function of haplotype-specific state. **e,f,g**) VAFs in clones where the dominant allele switches between A and B. Each point is the VAF of a mutation, with lines connecting the same mutation in different clones.

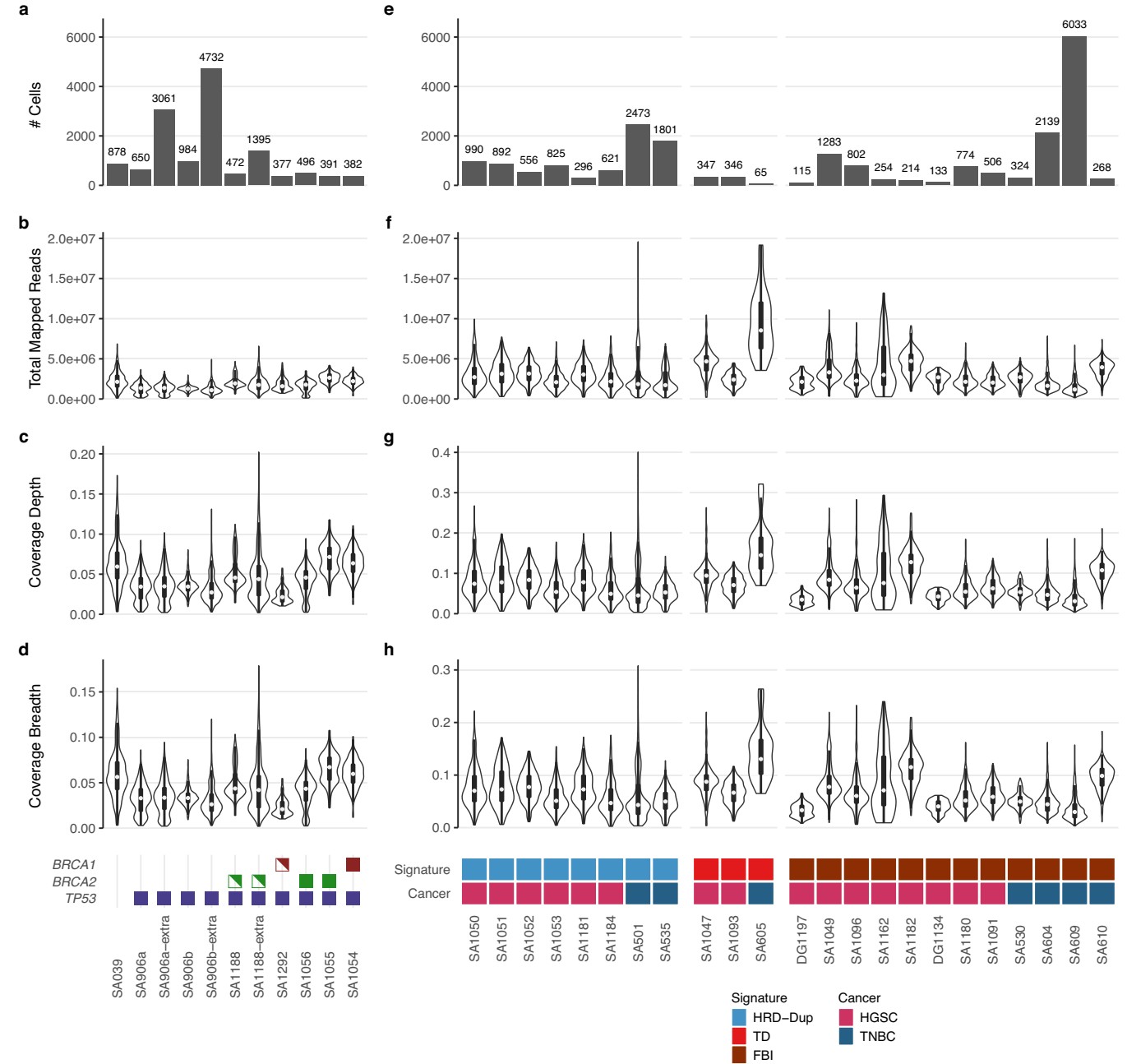

**Extended Data Fig. 4 | DLP summary statistics.** Summary of DLP+ sequencing statistics of data for 184-hTERT cell lines **a**–**d**) and HGSC and TNBC tumours **e**–**h**). Number of cells for each box plot as indicated in panels **a**) and **e**). Shared y axis labels shown at left. The legend for **e**–**h**) indicates the number of samples for each cancer and signature type. All box plots indicate the median, 1st and 3rd quartiles (hinges), and the most extreme data points no farther than 1.5x the IQR from the hinge (whiskers).

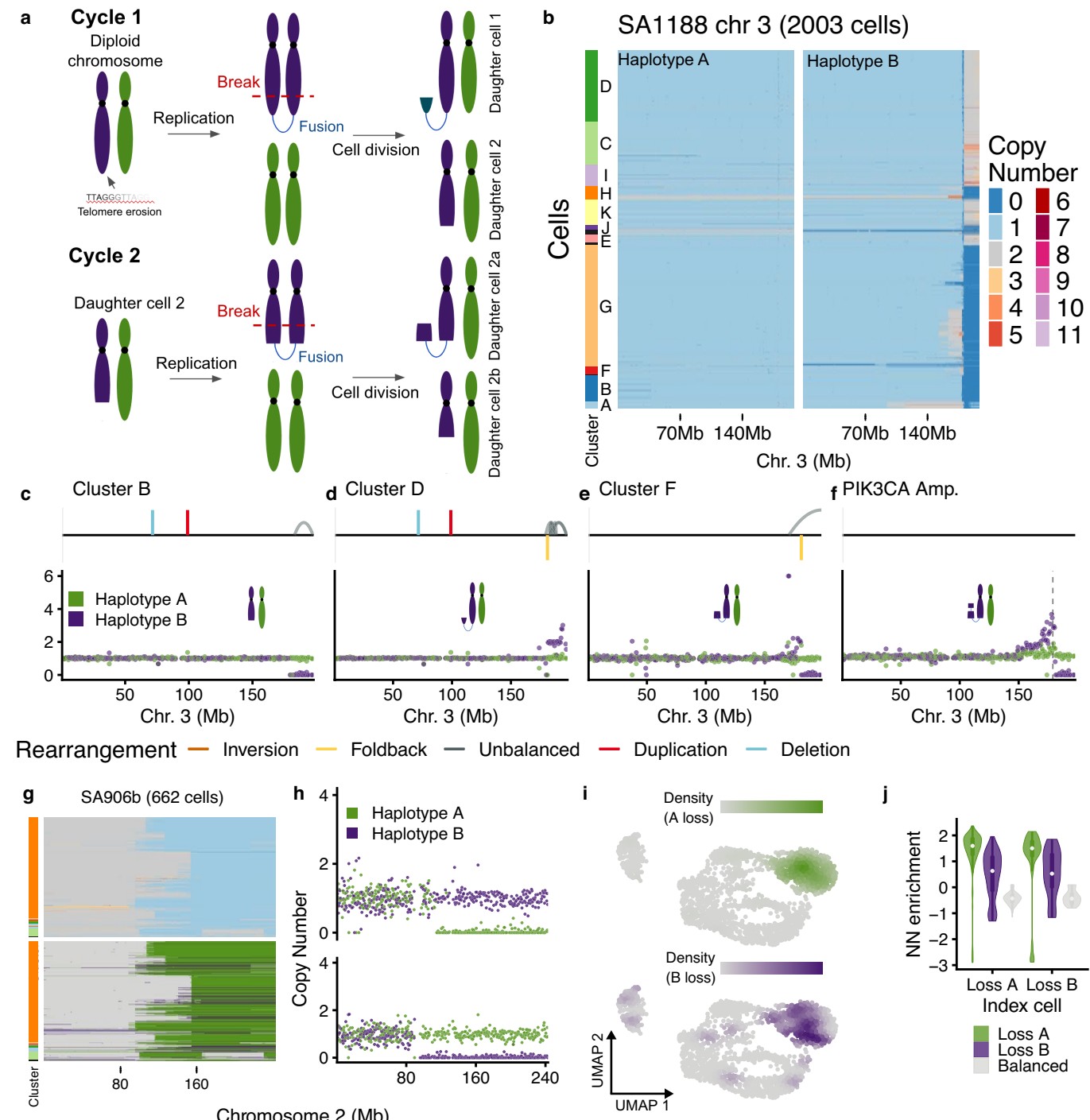

**Extended Data Fig. 5 | Haplotype-specific analysis reveals breakage–fusion–bridge processes and parallel losses. a**) Diagram of BFBCs **b**) Heat maps of the copy number of each homologue in SA1188. **c**–**f**) HSCN and structural variation in clusters B, I, F and the small subpopulation with *PIK3CA* amplification. Here we plot the copy number for each homologous chromosome in purple for homologue B and green for homologue A. **g**) Parallel copy number losses in SA906b: total copy number (top) and HSCN (bottom) heat maps for chr2 in SA906b **h**) two individual cells from **g**). **i**) UMAP dimensionality reduction plots of scRNA-seq data generated from SA906b, colours indicate the density of loss of chr 2q A *vs*. B haplotype. **j**) Enrichment of the haplotype-specific state on chr 2q of nearest neighbour cells (# cells with loss of A = 175, # of cells with loss of B = 34, # Balanced = 2066). All box plots indicate the median, 1st and 3rd quartiles (hinges), and the most extreme data points no farther than 1.5x the IQR from the hinge (whiskers).

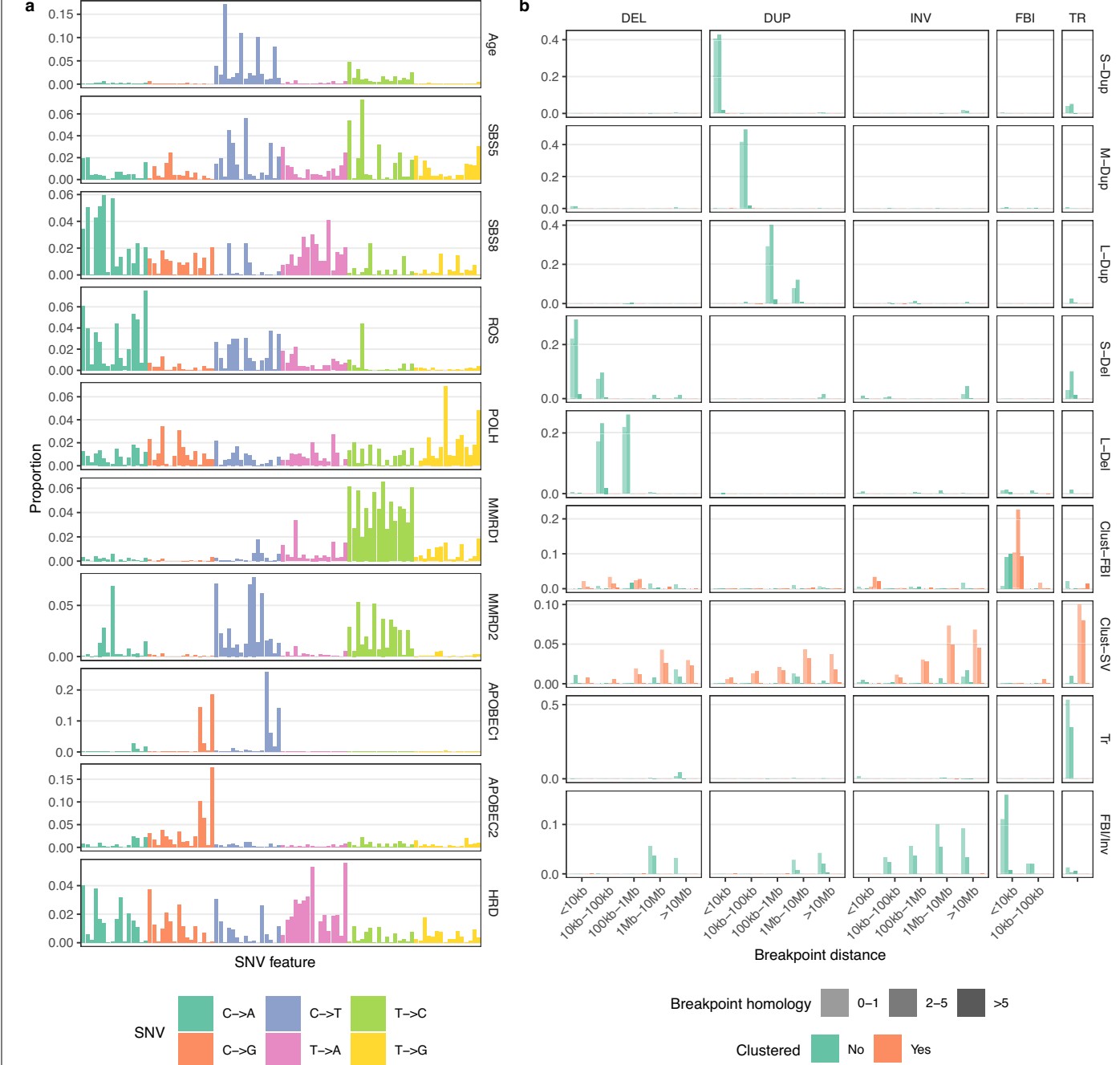

**Extended Data Fig. 6 | SNV and SV signatures. a)** SNV and **b)** SV mutation signatures estimated from HGSC and TNBC bulk tumour mutation catalogues using the MMCTM method. The x axis in a) is the 96-channel (*i.e.* A[C>A]A, ..., T[T>G]T) SNV types. SV types are DEL: interstitial deletions, DUP: tandem duplications, INV: inversions, FBI: fold-back inversions, TR: translocations. SV signature labels are S-Dup: small duplications, M-Dup: medium duplications, L-Dup: large duplications, S-Del: small deletions, L-Del: large deletions, Clust-FBI: clustered fold-back inversions, Clust-SV: clustered other structural variants, Tr: translocations, FBI/Inv: fold-back inversions and inversions.

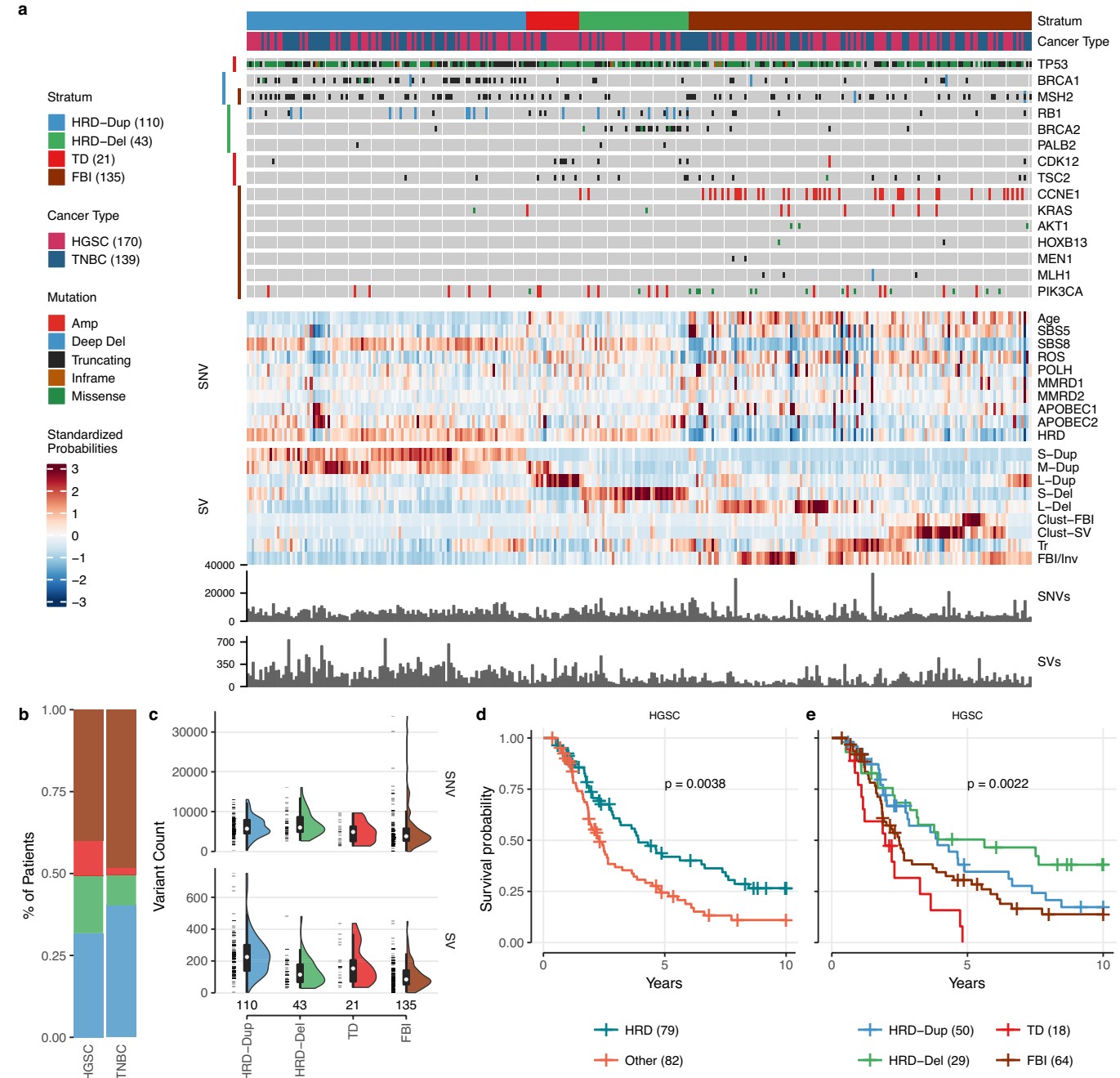

**Extended Data Fig. 7 | Meta-cohort signature analysis of 139 TNBC and 170 HGSC bulk whole genomes. a**) Heat map representing individual patients as columns, annotation tracks (top) including cancer type and mutation status of key genes (strata with adjusted p-values ≤ 0.1 shown as coloured bars on *left*), standardized signature probabilities of SNVs and SVs (middle) and event counts (bottom). **b**) Signature type (see stratum annotation track) proportions by cancer type. **c**) SNV and SV count distributions per signature type (number of samples shown below each violin, data points shown left of violins). Kaplan–Meier survival probability of HGSCs faceted by **d**) HRD and **e**) more granular signatures (p-values computed using the log-rank test, p = 0.0038 for **d**) and p = 0.0022 for **e**)). All box plots indicate the median, 1st and 3rd quartiles (hinges), and the most extreme data points no farther than 1.5x the IQR from the hinge (whiskers).

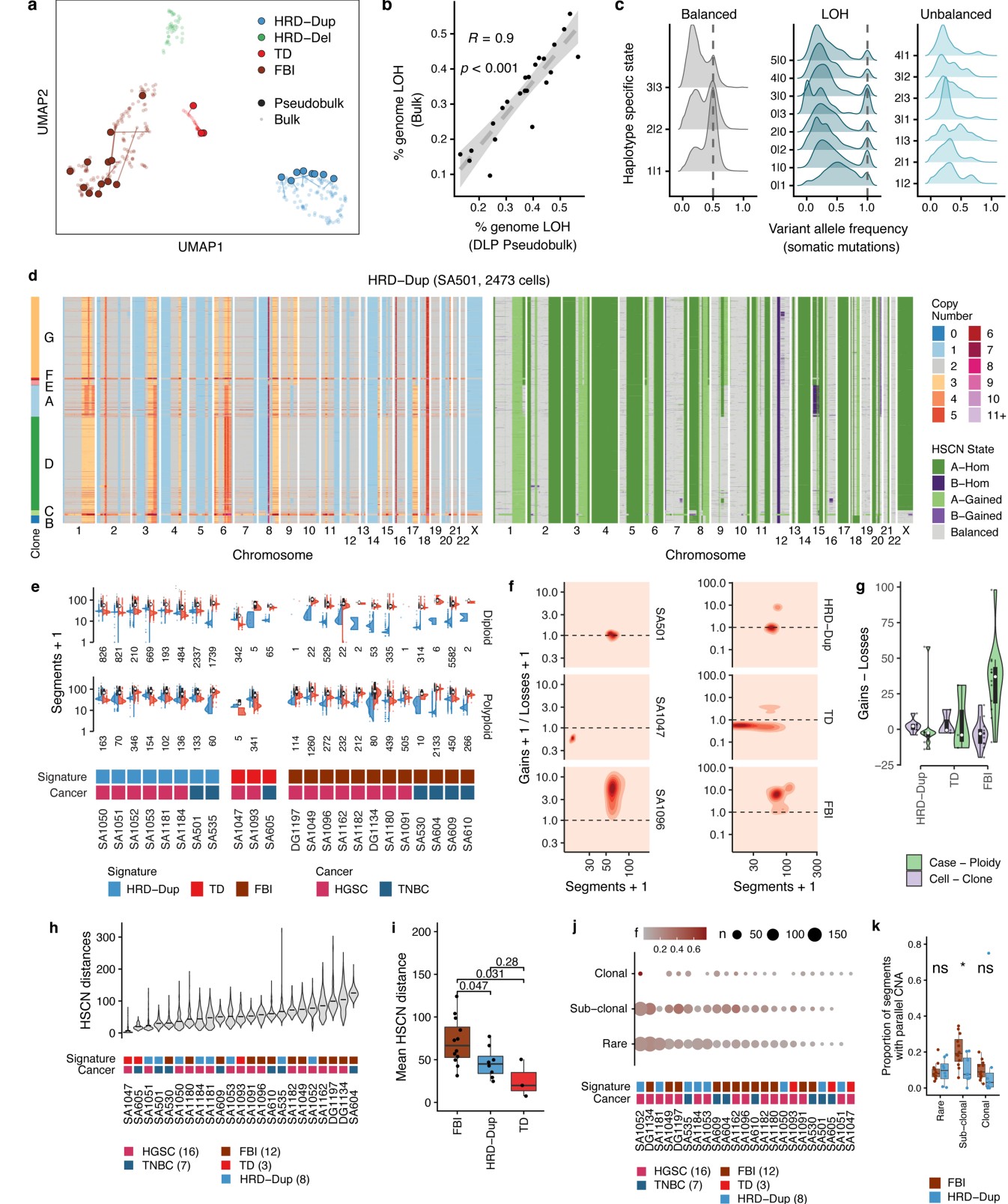

**Extended Data Fig. 8** | See next page for caption.

**Extended Data Fig. 8 | Summary, quality control and features of single-cell WGS of tumours. a**) UMAP of meta-cohort signature probabilities. Lines connect DLP-pseudobulk to their bulk data counterpart. **b**) Correlation of proportion of the genome that is LOH between DLP-pseudobulk (horizontal) and matched bulk WGS (vertical). Correlation coefficient (R) and p-value (p) derived from a linear regression in inset, shaded area shows the 95% CI of the linear regression. **c**) VAF distributions (horizontal) for somatic mutations called in single cells as a function of haplotype-specific state (vertical), coded as integer copy level allele A | integer copy level allele B. Data from all DLP samples are included. **d**) Heat map showing total copy number (*left*) and HSCN (*right*) of single cells from a TNBC HRD-Dup case (SA501). **e**) Chromosomal gains and losses across different ploidy states and mutational signature grouping. Total counts (*black*), gains (*red*), and losses (*blue*) shown. **f**) Relationship between gain/loss ratios and number of gained or lost segments for representative datasets from each signature type (*left*) and all HRD-Dup, TD, or FBI cases (*right*). **g**) Differences in copy number segmental gain and loss counts (n = 12 FBI, n = 8 HRD-Dup, n = 3 TD), comparing ploidy-relative case-level consensus copy number profiles (*green*) and mean cell-level changes relative to clone copy number profiles (*purple*). **h**) HSCN distance distributions for all PDX samples. Distribution is over $n = 1,000$ sampled pairwise HSCN distances. Horizontal black line shows the mean value of the distribution. **i**) HSCN distance distributions as a function of signature type, each dataset is summarized as the mean of the distributions on the left. P-values indicate per group comparisons using the two-sided Wilcoxon test (n = 12 FBI, n = 8 HRD-Dup, n = 3 TD). **j**) Number of parallel copy number segments (n, size of circle) and the proportion of segments containing parallel events (f, colour of circle) across all datasets as a function of clonality. Clonal: CCF > 80%, Subclonal: 20% < CCF ≤ 80%, Rare: 1% < CCF ≤ 20%. **k**) Proportion of segments with parallel CNA in HRD-Dup vs FBI, * = p < 0.05, ns = p > 0.05, two-sided Wilcoxon test (n = 12 FBI, n = 8 HRD-Dup). Exact p-values from left to right, p = 0.85, p = 0.031, p = 0.1. All box plots represent the median, 1st and 3rd quartiles (hinges), and the most extreme data points no farther than 1.5x the IQR from the hinge (whiskers).

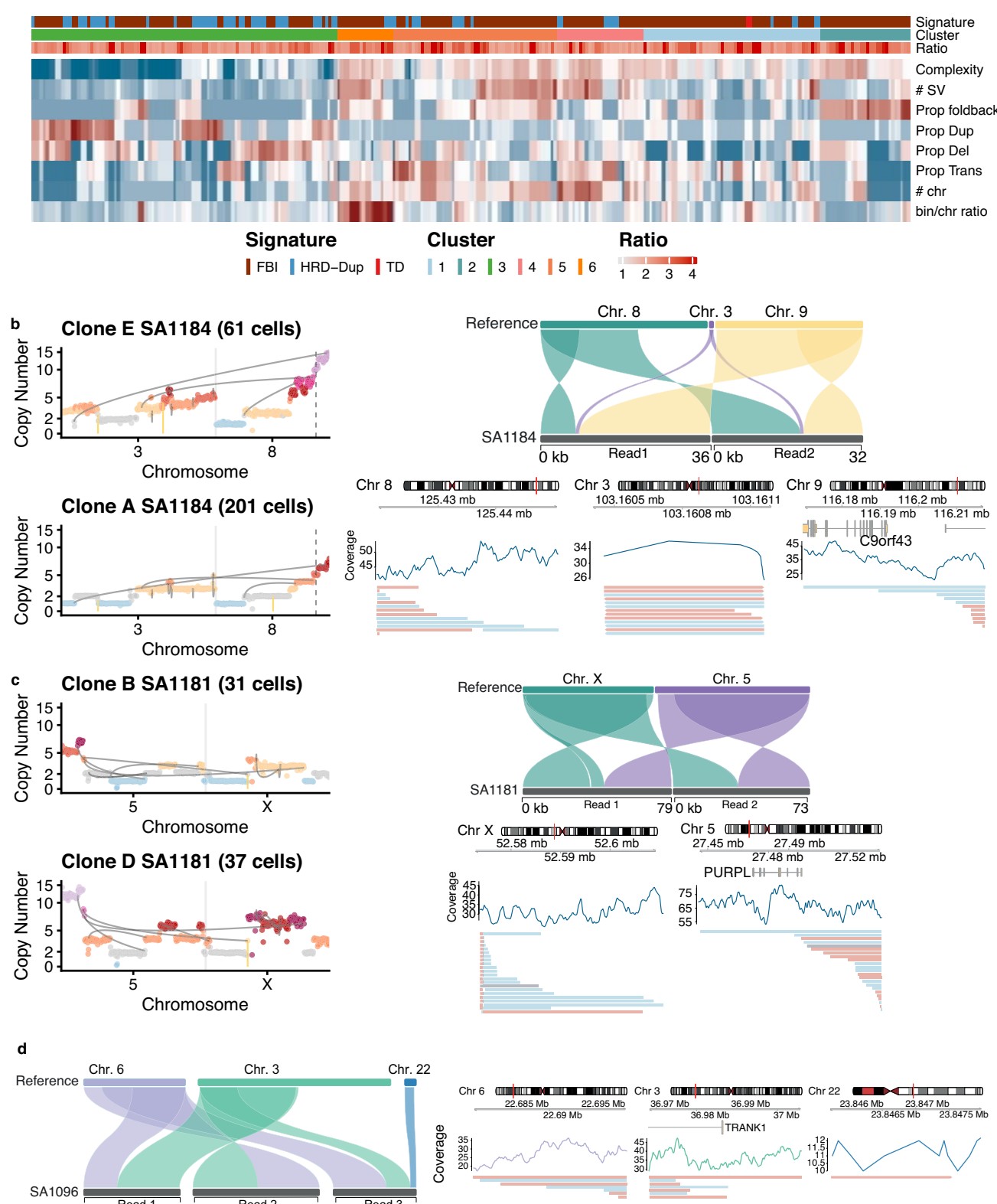

**Extended Data Fig. 9** | See next page for caption.

**Extended Data Fig. 9 | Genomic features of HLAMPS and long read sequencing validation. a)** Each column is a HLAMP that amplifies an oncogene. Each row is a feature extracted from a region 15Mb either side of the amplification. Complexity = entropy of haplotype-specific states, #SV = total number of structural variants identified, proportion of SVs of each type: fold-back inversions, duplications, deletions and translocations. #chr = number of chromosomes involved in translocation. bin/chr ratio copy number of the bin containing the oncogene to the average copy number across the chromosome. Ratio is the copy number ratio between the clone with the maximum copy number state and the minimum copy number state. **b–c)** HLAMPs involving multiple chromosomes, left plot shows copy number profiles from pseudobulk clones derived from DLP, lines indicate rearrangement breakpoints, right plot shows example long reads from Oxford nanopore technologies that support inter-chromosomal translocations. Example reads and their mapping to chromosomes of interest (top right), long-read coverage of genomic region and alignment of all supporting reads (bottom right). **b)** SA1184 *MYC* amplification **c)** SA1181 chr5q amplification. **d)** Long-read support for inter-chromosomal alterations involving chromosomes 3 and 6 in SA1096, DLP clone-level plots shown in Fig. 4j.

# Reporting Summary

## Statistics

For all statistical analyses, confirm that the following items are present in the figure legend, table legend, main text, or Methods section.

| n/a | Confirmed | |
|---|---|---|
| ☐ | ☒ | The exact sample size (*n*) for each experimental group/condition, given as a discrete number and unit of measurement |
| ☐ | ☒ | A statement on whether measurements were taken from distinct samples or whether the same sample was measured repeatedly |
| ☐ | ☒ | The statistical test(s) used AND whether they are one- or two-sided <br> *Only common tests should be described solely by name; describe more complex techniques in the Methods section.* |
| ☐ | ☒ | A description of all covariates tested |
| ☐ | ☒ | A description of any assumptions or corrections, such as tests of normality and adjustment for multiple comparisons |
| ☐ | ☒ | A full description of the statistical parameters including central tendency (e.g. means) or other basic estimates (e.g. regression coefficient) AND variation (e.g. standard deviation) or associated estimates of uncertainty (e.g. confidence intervals) |
| ☐ | ☒ | For null hypothesis testing, the test statistic (e.g. *F*, *t*, *r*) with confidence intervals, effect sizes, degrees of freedom and *P* value noted <br> *Give P values as exact values whenever suitable.* |
| ☐ | ☒ | For Bayesian analysis, information on the choice of priors and Markov chain Monte Carlo settings |
| ☒ | ☐ | For hierarchical and complex designs, identification of the appropriate level for tests and full reporting of outcomes |
| ☒ | ☐ | Estimates of effect sizes (e.g. Cohen's *d*, Pearson's *r*), indicating how they were calculated |

*Our web collection on statistics for biologists contains articles on many of the points above.*

## Software and code

Policy information about availability of computer code

| Data collection | DLP+ single cell WGS pipeline: https://github.com/shahcompbio/single_cell_pipeline <br> Bulk WGS pipeline: https://github.com/shahcompbio/wgs |
|---|---|
| Data analysis | MMCTM method: https://github.com/shahcompbio/MultiModalMuSig.jl v0.3.0 <br> SIGNALS: https://github.com/shahcompbio/signals v0.7.2 <br> CELLRANGER https://support.10xgenomics.com/single-cell-gene-expression/software v3.1.0 <br> cellSNP https://github.com/single-cell-genetics/cellsnp-lite v1.2.2 <br> seurat https://satijalab.org/seurat/ v4.1.0 <br> scrublet https://github.com/swolock/scrublet v0.2.3 <br> CHISEL https://github.com/raphael-group/chisel v1.0.0 <br> Alleloscope https://github.com/seasoncloud/Alleloscope <br> sitka https://github.com/UBC-Stat-ML/sitkatree <br> castor https://cran.r-project.org/web/packages/castor/index.html v1.6.6 <br> Guppy https://nanoporetech.com/nanopore-sequencing-data-analysis v3 <br> minmap2 https://github.com/lh3/minimap2 v2.24 <br> sniffles https://github.com/fritzsedlazeck/Sniffles v1.0.12 <br> cuteSV https://github.com/tjiangHIT/cuteSV v1.0.11 |

For manuscripts utilizing custom algorithms or software that are central to the research but not yet described in published literature, software must be made available to editors and reviewers. We strongly encourage code deposition in a community repository (e.g. GitHub). See the Nature Portfolio guidelines for submitting code & software for further information.

## Data

Policy information about availability of data

All manuscripts must include a data availability statement. This statement should provide the following information, where applicable:

- Accession codes, unique identifiers, or web links for publicly available datasets
- A description of any restrictions on data availability
- For clinical datasets or third party data, please ensure that the statement adheres to our policy

All data are available for general research use. Processed data including somatic mutation data for bulk WGS, total (and allele-specific) copy number profiles for DLP + data and filtered count matrices for scRNA-seq data are available for download at https://zenodo.org/record/6998936 . Raw scRNA-seq data are available for download at https://ega-archive.org/studies/EGAS00001006343 . Raw single-cell sequencing data generated for this study are available from https://ega-archive.org/studies/EGAS00001006343 , and previously published single-cell sequencing data used in this study are available at https://ega-archive.org/studies/EGAS00001004448  and https://ega-archive.org/studies/EGAS00001003190 . Somatic mutation calls from bulk WGS for 16 patients with TNBC for whom the IRB consent does not include public deposition of raw sequencing data are available at http://www.ncbi.nlm.nih.gov/projects/gap/cgi-bin/study.cgi?study_id=phs003038.v1.p1, and raw sequencing data can be provided upon request under material transfer agreement to shahs3@mskcc.org. Bulk WGS BAM files from patients under IRB consent protocols for public release of raw data are available for download at https://ega-archive.org/studies/EGAS00001006343 , http://www.ncbi.nlm.nih.gov/projects/gap/cgi-bin/study.cgi?study_id=phs003036.v1.p1 and https://ega-archive.org/datasets/EGAD00001003268  (for previously published data) or by request under material transfer agreement to shahs3@mskcc.org and saparicio@bccrc.ca.

# Field-specific reporting

Please select the one below that is the best fit for your research. If you are not sure, read the appropriate sections before making your selection.

☒ Life sciences         ☐ Behavioural & social sciences         ☐ Ecological, evolutionary & environmental sciences

For a reference copy of the document with all sections, see nature.com/documents/nr-reporting-summary-flat.pdf

# Life sciences study design

All studies must disclose on these points even when the disclosure is negative.

| | |
|---|---|
| Sample size | 309 patients, 309 with bulk WGS and 23 with scWGSor RNAseq. 8 hTERT lines, 2 ovarian cancer cell lines. Sample size was determined by the number of patient tissues accrued and number of 184hTERT clones with confirmed knockout identified. |
| Data exclusions | poor quality and s-phase cells were removed from analysis as per Methods |
| Replication | Where possible, biological duplicates were sequenced by scDNAseq. 11 patients with scWGS have more than 1 replicate tissue sequenced by DLP+ |
| Randomization | Patients were stratified into groups according to their mutational signature as described in the manuscript. No grouping of animals was performed, so no randomization was necessary |
| Blinding | Patient tissues were stratified by mutational signature during analysis, blinding was not necessary during data collection or analysis as no groups were assigned. |

# Reporting for specific materials, systems and methods

We require information from authors about some types of materials, experimental systems and methods used in many studies. Here, indicate whether each material, system or method listed is relevant to your study. If you are not sure if a list item applies to your research, read the appropriate section before selecting a response.

## Materials & experimental systems

| n/a | Involved in the study |
|---|---|
| ☐ | ☒ Antibodies |
| ☐ | ☒ Eukaryotic cell lines |
| ☒ | ☐ Palaeontology and archaeology |
| ☐ | ☒ Animals and other organisms |
| ☐ | ☒ Human research participants |
| ☒ | ☐ Clinical data |
| ☒ | ☐ Dual use research of concern |

## Methods

| n/a | Involved in the study |
|---|---|
| ☒ | ☐ ChIP-seq |
| ☒ | ☐ Flow cytometry |
| ☒ | ☐ MRI-based neuroimaging |

# Antibodies

| | |
|---|---|
| Antibodies used | mouse anti-p53 (Santa Cruz SC-126), mouse anti-BRCA1 (Santa Cruz SC-6954), mouse anti-BRCA2 (Millipore OP95), goat anti-GAPDH (SC-48166), rabbit KRAS (Lifespan Bioscience, LS-B4683) |
| Validation | KRAS - Validated by our lab using human uterine tissue and by Lifespan Bioscience using human uterine and placental tissues. GAPDH - Validated by Santa Cruz. p53 - Validated by Santa Cruz using A549, Daudi, NTERA2, SW480 cell lines and human bladder carcinoma tissue. BRCA1 - Validated by Santa Cruz using A-431, HeLa, U2OS and MCF7 cell lines. BRCA2 - Validated by MIllipore using MCF7 cell line. |

# Eukaryotic cell lines

Policy information about cell lines

| | |
|---|---|
| Cell line source(s) | 184hTERT L9 (SA039), 184-hTERTTP53-/- (SA906a and SA906b), 184-hTERTTP53-/-;BRCA1+/- (SA1292), 184-hTERTTP53-/-;BRCA1-/- (SA1054),  184-hTERTTP53-/-;BRCA2-/- (SA1055 and SA1056)  and 184-hTERTTP53-/-;BRCA2+/- (SA1188), OV2295, TOV2295 |
| Authentication | All cell lines were authenticated by short tandem repeat testing by Genetica Lab Corp DNA Identity |
| Mycoplasma contamination | All cells were tested for mycoplasma contamination by Genetica Lab Corp DNA Identity and tested negative |
| Commonly misidentified lines (See ICLAC register) | No commonly misidentified cell lines were used in this study. |

# Animals and other organisms

Policy information about studies involving animals; ARRIVE guidelines recommended for reporting animal research

| | |
|---|---|
| Laboratory animals | NOD/SCID/IL2-/- (NSG) and NOD/Rag/Il2-/- (NRG) mice were used in this study. Housing was maintained in a 18–25 °C temperature range and 20–70% humidity range, with a 12 hour daylight cycle (on at 6:00am, off at 6:00pm). Surgery was carried out on female mice between the ages of 5–12 weeks. All experimental procedures were approved by the University of British Columbia Animal Care Committee |
| Wild animals | This study did not involve wild animals |
| Field-collected samples | This study did not involve samples collected from the field |
| Ethics oversight | The animal care committee of University of British Columbia approved all experimental procedures in this study |

Note that full information on the approval of the study protocol must also be provided in the manuscript.

# Human research participants

Policy information about studies involving human research participants

| | |
|---|---|
| Population characteristics | Age, gender (female), past and current diagnosis, hormone receptor status, past and current treatment, survival. Only survival was used in this manuscript. |
| Recruitment | Patients were recruited with informed consent during surgery or biopsy procedures. Patients were not selected for or excluded based on genomic information, so would not impact results. |
| Ethics oversight | approved by the Ethics Committees at the University of British Columbia |

Note that full information on the approval of the study protocol must also be provided in the manuscript.

