## [Peer Review File · Nature]

Manuscript Title: Single cell genomic variation induced by mutational processes in cancer

Editorial Note: The work described in this article was originally reviewed as two separate manuscripts and merged into a single manuscript following the first round of review. This peer review file contains reviewer comments on both original manuscripts, as indicated (Funnell et al. and Williams et al.).

Reviewer Comments & Author Rebuttals

Reviewer Reports on the Initial Version of Funnell et al.:

Referees' comments:

Referee #1 (Remarks to the Author):

In this manuscript, Funnell and O'Flanagan et al. use bulk WGS and single-cell WGS (scWGS) to study how single nucleotide mutations (SNVs), copy number alterations (CNAs) and structural variants (SVs) shape the genomic heterogeneity of high-grade serous ovarian cancer (HGSOC) and triple-negative breast cancer (TNBC). The authors combined state-of-the-art experimental and computational tools to co-analyze the mutational patterns across the analyzed tumors. Next, they sequenced an isogenic in vitro model of immortalized mammary epithelial cells with various combinations of TP53, BRCA1 and BRCA2 mutations, and found distinct mutational signatures for each of these genotypes. Finally, the authors generated PDXs and performed scWGS of 22 primary tumors and PDXs, focusing on the comparison of tumors from two specific mutational groups: HRD-Dup (enriched for BRCA1 mutation and patterns reflecting short/medium tandem duplications) and FBI (enriched for CCNE1 amplification and foldback inversions). They report different CNA patterns between these two groups, consistent with distinct CNA acquisition processes. FBI tumors accrued more gains than HRD-Dup tumors, and also exhibited more high-level amplifications (HLAMP). Intra-tumor variation in the exact genomic breakpoints of specific CNAs was observed and termed 'serriform structural variation' (SSV), and this phenomenon was also more common in FBI vs. HRD-Dup tumors. The FBI tumors therefore seem to be more structurally unstable, suggesting higher evolvability of potential clinical relevance.

Overall, this study is a tour de force of current cancer genomics. However, I have several doubts regarding its suitability for Nature. First, the biological novelty of the findings and the conceptual advance over current knowledge is rather moderate. Second, in its current form the paper is oriented toward a specialized audience, and the writing ought to be somewhat simplified if it were to target a wider audience. Third, the three parts of the paper (bulk sequencing of tumors, sequencing of engineered cell lines, single-cell sequencing of tumors) seem to be a bit detached: the BRCA1 vs. BRCA2 comparison (which is highly interesting) is limited to the in vitro analysis, whereas the HDR-Dup vs. FBI comparison and the description of the SSV phenomenon are studied only in the clinical data; a more integrative analysis would make the results and conclusions stronger and more

exciting.

In addition, below are a few specific suggestions:

Fig.2:

* While the vast majority of tumors have an inactivating TP53 mutation, not all of them do. However, the TP53-WT tumors do not cluster separately from the TP53-mut tumors. Do these tumors have mutations in other genes of the p53 pathway? A direct comparison of the TP53-WT vs. TP53-mut tumors would be of interest.

Fig. 3:

* The analysis finds that “despite skewing towards accumulation of losses at the single cell level, human cancers nevertheless exhibit tandem duplication enrichment at the bulk genome level in BRCA1 associated cancers, but enrichment of interstitial deletions in BRCA2 associated cancers.” Could this reflect different evolutionary pressures in vitro and in vivo?

* The increased levels of CNAs found in TP53-null/BRCA1-null and TP53-null/BRCA2-null cells in comparison to TP53-null cells are interesting. Is this trend also statistically significant in the bulk-population tumor data shown in Fig. 2? And in the single-cell tumor data shown in Fig. 4? (It’s hard to tell from the plots.)

* The TP53-null/BRCA2-null show a very specific aneuploidy pattern – loss of chromosomes 2q and 14 (Fig. 3e). Is the same pattern seen with these specific mutations in the primary BRCA tumors?

* How was the rate of chromosome mis-segregation determined (e.g., Fig. 3h)? This should be briefly clarified in the text.

Fig. 6:

* Is the ‘serration’ pattern enriched for any genomic feature (nucleotide composition, distance from centromere/telomere, etc.)? It is likely that there is no sufficient statistical power for such an analysis with the current cohort size, however maybe there is an interesting trend? Also, could this be analyzed using previously published scDNAseq data from tumors, generated by the authors or by other groups?

* The authors analyzed a few thousand single cells in vitro. Can the SSV pattern be observed in these data?

Referee #2 (Remarks to the Author):

Funnell and colleagues investigate the consequences of SNV and SV mutational signatures on genomic instability at single cell resolution. They apply their previously developed computational algorithm that jointly infers signatures from both SNV and SV mutation types on bulk WGS samples from ovarian (HGSC) and breast (TNBC) tumors. From these results, they delineate clusters of tumors associated with specific SNV and SV features that were also found to harbor genomic alterations for individual genes of interest (e.g. HRD tandem duplication phenotype in BRCA1 mutated tumors and HRD deletion phenotype in BRCA2 mutated tumors). The authors further examine the impact of compound biallelic loss of TP53 and BRCA1/2 at the single cell level in vitro with CRISPR KO models

where biallelic loss of BRCA1/2 induced greater genomic instability and aneuploidy compared to a background of biallelic TP53 loss. Subsequent single-cell WGS profiling of PDX tumors showed HRD-tandem duplicator phenotypes with more even distribution of losses and gains vs foldback inversion (FBI) phenotypes. FBI tumors accrued copy number gains at a higher rate, with high-level amplifications co-occurring with FBI tumors in known driver genes such as ERBB2, KRAS, MYC, CCNE1, and FGFR1, with clone-specific variation in amplitude of these high-level amplifications, suggesting therapeutic implications.

Although the impact of structural variations together with mutations at the single-cell level on genomic instability is certainly interesting, the therapeutic significance of the manuscript is limited as all results are associative and not supported by pharmacological experiments. Clinical significance is limited to observations of modest prognostic effects which are largely already known and there is no data regarding specific drug responses. The issue for oncologists is TNBC and ovarian cancer treatment is only partially effective, patients can be given more than 5 different chemotherapy agents with different interaction with DNA repair, DNA replication and cell division processes. While the authors speculate about therapeutic effects of investigational drugs they do not conduct any studies of this type. They do have access to TNBC PDX and so this is something the authors could consider to increase impact. Another under-discussed issue is the effect of different mutational mechanisms on the immunogenicity of individual tumors. Finally there is also an issue of novelty since the group's SNV/SV algorithm was applied to HGSC and breast tumors in a previous publication where association with BRCA1/2 was discussed.

Major comments:

1. Their previous publication (PMID: 30794536) applied the same computational algorithm already to HGSC and breast samples where association with BRCA1/2 mutations were also examined. How much of Fig 2 in the current submission represent new observations?
2. It is not clear how gene list for mutation/amp analysis in Fig 2 were picked. Were these the top most mutated or amplified genes in respective group (Stratum). An unbiased enrichment analysis is warranted.
3. The authors mention HRD signature enriched in BRCA1 mutation and CCNE1 amplification in FBI clusters, but no statistical basis of enrichment or associated p-value reflecting how strong or weak the enrichment is shown.
4. Some clusters seem to be rich in higher age patients in FBI and L-Del clusters. A location and telomeric repeat analysis is required to ensure these genomic aberrations are not age or telomeric loss related.
5. The authors associate FBI tumors with high rates of copy number gains/high level amplification events, but seems counterintuitive given their previous findings of deletions associated with FBI. This discrepancy should be discussed.
6. The authors state the HRD-tandem duplicator phenotype with BRCA1 mutations leads to a higher rate of genomic instability as compared to HRD-deletion phenotypes with BRCA2 mutations which may explain differences to cisplatin treatment. The cisplatin pharmacological experiments in vitro should be extended to in vivo models.
7. To further increase clinical and therapeutic relevance, pharmacological inhibition of targets such as ERBB2, FGFR1, etc, that were found to have clone-specific high-level amplifications should be assessed for FBI tumors.

Minor comments:

1. Add brief description of FBI to introduction section.
2. How were tumors exhibiting strong score for multiple clusters treated ? For example there are many samples where there appears to be equally strong signal for FBI/INV and L-DEL but these are categorized as L-Del.

Referee #5 (Remarks to the Author):

[Reviews to the manuscripts (Funnel et al.) and (Williams et al) are presented together below.]

The studies led by Shah and Aparicio (in the following we will refer to them by their respective first authors) build upon single-cell whole-genome sequencing (scWGS) analyses of human primary tumors, patient-derived xenografts, and engineered cell lines to explore features of chromosomal instability during tumor evolution. These studies certainly benefit from a compelling dataset (although already partly published in PMID: 34163070) and rigorous computational biology analyses, including the development of novel tools that we can see of interest and useful to the scientific community.

However, where these studies seem to suffer is in the biological novelty introduced by their finding. As we discuss in more detail below, the results presented are largely confirmatory and descriptive. In particular, observations from one manuscript could often benefit the other and the choice of separating the two studies seems rather counterproductive. Below, we separately discuss in more detail the two studies, highlighting where integration of the two would be most beneficial.

Funnel et al.)

In our opinion, this study is the weakest of the two in terms of biological findings and should be integrated into the second study. This work focused on mutational signatures associated with the emergence of structural variants rather than single nucleotide mutations and how these signatures determine chromosomal instability in different models of serous ovarian and triple-negative breast cancer. In particular, by scWGS, the authors explored these mutational processes within single cells.

The stratification of patient samples in Figure 2 sets the stage for future analyses but signatures and associations were previously shown, e.g. the differences between BRCA1 and BRCA2 mutations (e.g., see previous work from the authors PMID: 30794536 or PMID: 32118208) or the similarity between TNBC and HGSC (PMID: 23000897 or PMID: 23636398). This figure could either be supplemental or substantially reduced to show the stratification of patient subgroups. (Figure 1 should be substantially edited as it currently includes figure panels representing actual data - so it seems - but it is unclear what data.)

The analysis in Figure 3 is rather intriguing as it delves into the differences between BRCA1 and BRCA2 chromosomal instability at the single-cell level. Here we have a few questions:

* First of all, the choice of models is sometimes not sufficiently motivated. Why do the authors provide a heterozygous and a homozygous model for BRCA2, but not for BRCA1?

* Similarly, for BRCA2^{-/-} the authors analyze two independent replicates at different time points, should the differences between the two samples be attributed to the different times they were kept in culture? If so, why BRCA2^{+/-} cells were analyzed at a much earlier time point? Would these cells at a later time point accumulate more chromosomal aberrations? Importantly, what is the replication time of these cells? The role of time-in-culture is not sufficiently discussed and analyzed in these results and the authors should consider the rate of accumulation of chromosomal aberration w.r.t. cell cycle divisions.

Moreover, some results from this analysis are expected, e.g., the association between number altered segments and polyploidy, and could be moved to supplement, whereas it would be great to show here how BRCA1 and BRCA2 SV signature are implemented at the level of single alleles, which is part of the analyses in the second manuscript (Williams et al.).

The authors then move to a more detailed comparison between HRD-dup (i.e. the SV signature associated with BRCA1 mutations) and FBI-associated tumors (another SV signature independent of both BRCA1 and BRCA2) in human samples.

(We should note that although 5 additional samples not exhibiting either of these signatures are shown in the figure, these are never discussed in the manuscript. We suggest removing them from the main figure as they just add more complexity to an already complex figure.)

Here, it is difficult to exactly understand what is the point the authors want to make. Indeed, the features of FBI and HRD-dup signatures have already been characterized, in part by the authors themselves (see for example PMID: 28436987, PMID: 29754820, PMID: 30104763) and these results simply confirm in single cells what was already known from bulk tumor analyses.

In addition, the figure is difficult to read as it misses color legends in individual panels, and some of the colors are used multiple times with different meanings. Results in Figure 4e are expected and results in Figure 4f and 4g are redundant. Additionally, the sample ordering in Figure 4d,e,f,g is not consistent, making it very difficult to match the results in the various panels. In Figure 4j there is a great overlap in the segment length distributions between HRD-Dup and FBI in interiors, while the only tangible difference at termini is between HRD-Dup gains and HRD-Dup losses and FBI gains/losses, so there is no clear distinction between the two signatures even in this case.

Given the previously reported association between FBI and HLAMP (see refs above), the authors search for known oncogenes in highly amplified regions from FBI samples. Here they found a few interesting targets, however, more details are needed.

First and foremost it is unclear how focal these events are. Indeed, instead of providing a genome-wide map of copy number statuses, the authors should zoom in on the affected loci and clearly show the size of these events. Indeed for some of the proposed targets (see BCL11A and MET), copy number gains seem to be arm-events. In these cases, it is difficult to say that these events are selected because they amplify a specific oncogene. Here the authors should have used matched RNA-seq data to show that these oncogenes or their downstream targets are up-regulated when amplified. In the absence of this data, I wouldn't make a big deal about these events.

Furthermore, given the association of the FBI signature with bridge fusion break cycle (BFBC) events (which was shown in previous studies from the authors) this analysis would gain robustness and completeness if combined with the analysis of the same amplicons done in Williams et al. where BFBC events are analyzed in detail. Here it seems the authors did an overall analysis of these events to then split the results into two studies, but each half feels incomplete without the other.

Finally, the authors highlight what they call a serriform pattern for chromosomal breakpoints, where the same/matched structural variant exhibits slightly different breakpoints in different single cells. As discussed by the authors, this pattern was already shown in vitro (PMID: 32299917) but was here first shown in human tumors. Was it named differently before? or simply not named?

This analysis is rather interesting, here we would like however to have a better feeling of the incidence of this phenomenon across different samples. What is the fraction of breakpoints that exhibits this pattern?

The association with FBI, although significant, is rather weak and supported by a limited number of samples. Do the authors think it would be possible to develop an approach to estimate the presence of this pattern from bulk sequencing data (e.g. by detecting reads supporting a breakpoint in different positions)? Pseudo-bulk of their samples could be used for testing. This could be rather intriguing and possibly a way to validate their findings in a larger cohort of samples.

Importantly, it would have been nice here to show this event at the level of single alleles using the approach proposed in the Williams et al. manuscript.

Overall, this study shows interesting features of tumor evolution under the effect of distinct mutational signatures. However, the findings are limited and descriptive and alone do not reach the level of novelty one expects from Nature. The authors could instead consider migrating the most innovative parts to the manuscript by Williams et al..

Williams et al.)

In this manuscript, the authors introduce a new approach to determine the copy number status of single cells at the level of single haplotypes. They then analyze haplotype-specific copy number alterations in the same datasets introduced above.

At first, the authors present their algorithm, called schnapps, and show its performance on a previously published dataset comprising multiple samples derived from the same patient and analyzed by scWGS. The analysis shows coherent B-allele frequencies (BAF) and variant allele frequencies (VAF) for SNPs, validating the approach. Here we have a few questions:

* In Figure 1B: given that the BAF is part of the inference procedure of schnapps, isn't it expected that values are distributed around the expected values for each allele combination?

* the description of the algorithm in the method section could be improved, especially in the use of notation and terminology. Is there a difference between "block allele" and "haplotype block"? What is the difference between the (Ch, Dh) and (Ca, Cb) notation?

- * The code description is great but at least the processed data should be made available for reproducibility (bin-level copy number status)
- * What is the effect of the delta parameter in the transition matrix of the HMM? It is currently set at 0.95, but how was this value determined? How robust are the results of the method to changing this parameter?
- * A comparison with previous approaches is warranted here. The authors have simply described the differences between schnapps and CHISEL and Alleloscope, but they should have actually tried to run these tools on the same dataset and show the differences in performance.
- * The authors do not provide access to the processed data. We understand that patient raw data should be accessed only under proper access control, but having access to the anonymized processed data (for example, bin level copy number estimates of each cell of the study together with their allelic status) could be very valuable. Additionally, we do not understand the inaccessibility of the engineered cell line raw data. Where can we find the processed data to reproduce the analysis presented in the Github repository?

Figure 1d is really nice but we have trouble understanding the color code of the second heatmap. If we read correctly the color legend, the allelic imbalance in the heatmap should range between 0 and 0.5 (only blue hues are shown), but this is in contradiction with Figure 1c, isn't it?

The analysis of clonal and subclonal events that follows (Figure 2) is an intriguing negative result, i.e. subclonal events do not show evidence of selection. A question that comes to mind is whether there is any association here with the genetic context.

Is there any difference between BRCA1/2 mutated tumors and others?

Most notably, given this information is already available to the authors, what about the association with mutational signatures?

This analysis feels a bit out of scope here, it does not focus on allele-specific changes and seems rather more connected to the other manuscript by Funnel et al.. Combining the two would make a much nicer chapter.

The analysis of parallel events is also quite interesting but very difficult to follow in Figure 3. Color legends should be more clearly shown in association with the figure panels and if possible I would refrain from using the same/similar colors for total copy number and allelic imbalance. A suggestion: would it be possible/simpler to show two separate heatmaps for the estimated copy number status of each allele?

Also, it is unclear from the methods what are the requirements to call a parallel gain or loss. It looks like several events are quite different in size between the two alleles, although intersecting in a region (e.g. Figure 3g,h,i). The authors should clarify how the overlapping of changes in maternal and paternal haplotypes are computed.

The correlations in Figure 3k are rather weak with points clearly deviating from the line of fit. Anything to note in these samples? Is there an association with polyploidy or mutational signature?

Was the UMAP in Figure 3m generated using expression data for all genes or only for those in chr 2 (or 2q)? If the former, the authors cannot conclude that the clustering shows convergent

phenotypes as it could be driven by a myriad of other signals independent of chr. 2q. Can the authors actually test for allelic expression imbalance for genes in the lost part of the chromosome? Bulk RNA-seq data could be used for that.

Lastly, the remaining figures of the manuscript are highly curated reconstructions of specific events. Whereas the effort to reconstruct some of these events is commendable, the value seems rather anecdotal. The analyses in Figures 4 and 5 could be combined and should be put in the context of the mutational signature that generated these events. Figure 6 is unreadable and, again, as much as we understand the difficulty of representing this data, it is unclear what we can learn from it.

In conclusion, we feel that this manuscript (Williams et al.) brings forward a new methodology of possibly broad applicability, and its potential is nicely shown through multiple analyses revealing high allelic imbalance in chromosomal instability and patterns of convergent evolution between events in the homologous chromosomes. These results could benefit from the mutational signature analysis performed in Funnell et al., which here are completely ignored.

Overall, we strongly encourage the authors to propose a substantially revised and harmonized combination of the two manuscripts into one, streamlined from the many confirmatory results and focused instead on what new we can learn from these scWGS datasets and computational approaches.

Reviewer Reports on the Initial Version of Williams et al.

Referees' comments:

Referee #3 (Remarks to the Author):

This paper studies allele-specific copy number alterations (CNA) in the tumor setting. The authors developed a new tool, called schnapps, that detects allele-specific copy number alterations from low-coverage single cell DNA sequencing data. By applying schnapps to scDNA-seq data from a set of 20 high grade serous ovarian and triple negative breast cancer samples, they were able to study the frequency of copy number changes in the cancer genome. They reported strong evidence for positive selection of alterations in specific regions and parallel evolution through convergent alterations in both the paternal and maternal alleles. They showed, quite convincingly, that some of the intra-tissue diversity of some of the events are generated through breakage fusion bridge cycles.

The paper claims to make novel contributions in: (1) the development of the schnapps method, and (2) the biological findings pertaining to CNA's role in tumor evolution. Here are my assessment of the rigor and novelty of its findings along these two dimensions:

1. On the methodological contributions:

- A. Schnapps is conceptually very similar to CHISEL (Zaccaria and Raphael, 2021, Nature Biotechnology 39, 207-214), in that local smoothing and phasing is conducted to aggregate information from RDR and BAF (Schnapps uses a HMM, while CHISEL uses a bin-wise smoothing). Although Schnapps claims to be higher resolution (using smaller 500KB bins), the bin-size in CHISEL can be user adjusted. There was no rigorous benchmarking of Schnapps against CHISEL.
- B. Other than CHISEL, the recently published Alleloscope (Wu et al., 2021, Nature Biotechnology DOI: 10.1038/s41587-021-00911-w) also has the same goals and operates on the same data type. Other than scDNA-seq data, also Alleloscope works for scATAC-seq data.
- C. Although the authors claim that Schnapps works for scRNA-seq data, they only applied it at the chromosome arm level, and for one event. More extensive benchmarking and illustrations would be needed to establish its usefulness beyond scDNA-seq.
- D. No formal validation of the detected CNA events was performed to establish the accuracy of schnapps. In contrast, CHISEL corroborated their events using somatic SNVs and the spatial distribution of inferred clones. Wu et al. used matched linked-reads data to establish that the mirrored subclones are real and not an experimental or computational artifact. Wu et al. also used data downsampling to benchmark CHISEL and Alleloscope at varying coverage. Thus, this paper is not on par with published papers in terms of rigor.
- E. No formal benchmarking is done comparing Schnapps with either CHISEL or Alleloscope. These two methods are only cited in passing in the Introduction, without stating their close connections to schnapps.
- F. The scDNA-seq data sets used to check the accuracy of Schnapps in this paper have a mean coverage of 0.16x, which is much higher than most current scDNA-seq data sets (0.01-0.1x). Also, schnapps was only shown to work on data from this DLP protocol, which is not commonly used by the community. How does schnapps work on other platforms?

2. On the biological findings:

A. The data analyzed in this paper were generated by Funnell et al.

(<https://www.biorxiv.org/content/10.1101/2021.06.03.446999v1.full>) which was also attached as supplementary material to this paper. Thus, this paper does not contribute new data. Funnell et al. analyzed this data set in depth, but did not focus on allele-specific copy number events.

B. One of the main findings in this paper (stated in the abstract, and presented in Figure 3) is the high number of cases of parallel copy number evolution. The prevalence of such events were reported by Wu et al. (2021), through the analysis of >10 breast, gastric, and colon cancer samples. Raphael and Zaccaria also found such events in 2 breast cancer samples. Wu et al. further established the inter-clonal complexity at such hypermutable loci. Yet, both papers were only cited in passing in the introduction, and the findings of Wu et al. in this context were not mentioned.

C. This paper establishes that breakage fusion bridge cycles (BFBC) are responsible for some of the more complex copy number aberrations in their data. Although BFBC, as a mutational mechanism, is well known, their analysis linking it to the complexity observed in their data is novel and convincing.

D. The integration and analysis of the scRNA-seq data needs to be more rigorous – in the current draft, it is only done for one sample. The proposed method of CNA detection in scRNA-seq data is crude, with no evaluation of accuracy. How do we know whether the CNA detection is confounded by RNA expression variation? The scRNA-seq data is used to show that gene expression does not differ between two subclones carrying alternate alleles at a CNA region. But the only way that the evidence is presented is through a UMAP feature plot, where the localization of the B-allele loss and A-allele loss do seem to differ (global trends are the same, but there are small local differences). Can this hypothesis be examined more rigorously?

In light of these concerns, I believe that the paper, in its current form, is limited in its methodological and scientific contributions.

Minor Comments:

1. No sample “2295” mentioned in the main text, but it is used in the figure legend (Fig.1 and Supplementary table 1)
2. More description of the engineered cell line data, and scientific hypotheses regarding them, in the main text may be helpful in interpreting the results.
3. Wrong legend for chr3m, 3l
4. The structural variant plots are hard to read without some explanation, especially Figure 6.
5. They claimed in line 457 that “The resolution of schnapps is 0.5Mb whereas CHISEL uses 5Mb bins”. The bin size is actually flexible for CHISEL.
6. The A, B, C... labeling needs to be revised in Supplementary Fig. 6.
7. Software details:
 - The software can be run with the example datasets
 - schnapps was developed to work with Direct Library Preparation + (DLP+) data. works using the output of the pipeline developed in their lab. Maybe they can provide guides to help prepare input data for scDNA-seq of other platforms.
 - Need more explanation about the input data (what is “allele_id”, “hap_lab”..and etc?)
 - In the main page: one of the required input data is haplotype block counts per cell “(SNP counts may also work)”. What does the results look like when using SNP counts?

Referee #4 (Remarks to the Author):

The authors apply DLP+, an exciting technology that allows sparse sequencing of single cells and is scalable to thousands of cells (introduced previously by the authors in Laks et al., Cell, 2019). This method enables a deep dive into tumor heterogeneity. In the present paper, the authors analyze a data set on 11,097 cells from cell lines and to 21,852 cells from ovarian and breast cancers. The focus of this work is on performing analyses in a haplotype-specific manner and the authors introduce a computational framework to this effect. They apply it to study known processes, particularly breakage-fusion-bridge cycles, in cancer genomes. I list my comments and concerns in the following.

Major

- The authors discuss related work on CHISEL and Alleloscope (L457-470), which indeed are similar in spirit. I'd like to add that the idea to phase from B-allele frequencing goes back to at least 2014, where it was used by Schwartz et al. (PLoS Comp Biol), even though that was not in a single cell setting. I suggest to mention all these methods in the introduction.
- The point out differences in resolution between schnapps (500kbp) and Chisel (5 Mbp). Given the great scalability of DLP+, I am actually surprised that the authors do not attempt to leverage the large number of cells to improve the resolution. Given the cumulative coverage of cell carrying a joint event, that should be possible computationally. Comparing to other works, I see that Sanders et al., reported 200kbp resolution with fewer cells.
- The claim that the majority of the genome is target to alterations (gains, losses, LOH) when lowering the cell fration cutoff to 1% is certainly interesting (and relevant to our understanding of tumor evolution), but I'm not yet convinced that the authors have ruled out all possible artifacts. The number of cells per samples varies widely (around two orders of magnitude according to Table S1) and different cell counts could come with different challenges: For samples with very high cell numbers, I wonder whether there might be a multiple testing issues; i.e. could a subset of copy number diverged cells emerge by random fluctions in coverage? For a sample with low numbers of cells (e.g. only 49 cells), >1% translates into an event seen in only one cell and that could introduce noise. Could the authors comment on the possible contributions of these effects and how they safeguard agains this?
- Also, I would be interested to see this analysis stratified by event size, which is important for the interpretation. In other words, how finegrained is the background of CNAs upon which evolution can act?
- L167-176: Zooming into Supp Fig 3, it appears that there are a number of chromosomes where the fraction of cells with both paternal and maternal gains is >0 for all seven lines. So the authors should expand on why they see chr20 as an outlier, which is not evident for me from looking at this figure.

What do you propose drives selection for chr20 gains?

- Regarding the BFBC analysis: I am not an expert for mutational mechanisms, but the schematic in Figure 4 is not consistent with my understanding of a BFBC. In the topmost row, the fusion appears to happen between the telomeres, which (to my understanding) is exactly what the telomeres should prevent; i.e. the common model is that BFBC processes happen *because* of the loss of a telomere. So the fusion is between the "blunt" ends of the sister chromatids. However, the model in Figure 4a appears to be consistent with Figure 4b in terms of losses (Cluster F) gains (clusters A+B) of the same terminal part of the chromosome. Could the authors comment on the nature of this fusion, i.e. how could the derived chromosomes look like? Looking at Cluster F in Supp Fig 6, I start wondering whether in that subclone chr3 as acquired a telomere from chr17? Another aspect that makes me wonder about the consistency of BFBCs with the shown data is the absence of high amplifications.

- Related to this: I was not able to get a good sense on how reliable the "SV track" in Fig 4c-h (and later figures) are? The Methods part says that "SV calling was performed in a similar manner, by forming pseudobulk libraries, then running LUMPY and DESTRUCT on each pseudobulk library." Given the low per-cell coverage and how vastly the number of single cells varies across libraries, some evaluation of reliability vs pseudobulk coverage would be helpful. In general, more details are needed here. How are the calls between the two tools merged? Do I interpret the figures correctly that this process was run separately for each cluster? I went to Laks et al., 2019 because of the pointer in Line 479, but the respective method part was also sparse on the SV calling part.

- The author elevate the term "haplotypes" to the title and emphasize the phased nature of their analyses. In my view, that warrants a detailed evaluation of the completeness and accuracy of the phasing. Both are not obvious given that a) one can only phase regions with gains, losses, or LOHs in at least a subpopulation of cells and b) the data is also very sparse per single cell, translating into sparse phasing of regions with CNA in few cells only. The notion of haplotype blocks and interaction with Shapelt should be described in more detail. Ideally, an evaluation would be done from sequencing data from the parents of a cancer patient, but that data can obviously be difficult to acquire. Alternatively one could resort to sequencing technologies providing long(er) range phasing information such as long read sequencing, linked reads, Hi-C or Strand-seq. One aspect that I'm particularly concerned about is the phasing along an individual chromosome: Looking at Figure 4b again, I wonder how confident the authors can be that the different segments affected by the (putative) BFBC are indeed all from the same homolog? Given the nature of the phasing algorithm, additional confirmation is warranted in my view.

- To my understanding, the performance of the phasing method is substantially influenced by the heterogeneity of the tumor (in a homogeneous tissue without gains/losses/LOHs, one cannot use B-allele frequencies). The only signal left in such a setting is statistical phasing using a reference panel, which has a limited reach (typically 100kbp to 1Mbp, depending on the sample ancestry and size of the reference panel). So an evaluation on the influence of the tumor heterogeneity on phasing capabilities would be good.

Minor comments:

- Methods / from L392: Please completely define all notation so that a reader does not need to guess. What exactly is a "block allele" and what is a "haplotype block" (is "block" define based on the bin size or otherwise)? What is the j in $C_{\{h,j\}}$ and what is N ?
- L425/426: How is the overdispersion parameter estimated from read counts?
- L231-233: Did the tree construction process in Supp Fig 6 omit chr3? (If it didn't, then it would not be independent).
- There seem to be some mismatches between numbers reported in the manuscript and the supplementary table, e.g.: "We next investigated the landscape of copy number alterations in the tumour cohort from single cell whole genome sequencing (median 697 cells per sample, range 49-2,627), ..." Table S1 reports 4214 cells for SA609, so the range seems to be larger (and the median lower).
- Figure 1d: I did not formally count them, but there seem to be way fewer rows of pixels than 1031. So is that image actually displaying some kind of average across adjacent cells? Please indicate the used bin size in the caption.

Referee #5 (Remarks to the Author):

[Reviews to the manuscripts (Funnel et al.) and (Williams et al) are presented together below.]

The studies led by Shah and Aparicio (in the following we will refer to them by their respective first authors) build upon single-cell whole-genome sequencing (scWGS) analyses of human primary tumors, patient-derived xenografts, and engineered cell lines to explore features of chromosomal instability during tumor evolution. These studies certainly benefit from a compelling dataset (although already partly published in PMID: 34163070) and rigorous computational biology analyses, including the development of novel tools that we can see of interest and useful to the scientific community.

However, where these studies seem to suffer is in the biological novelty introduced by their finding. As we discuss in more detail below, the results presented are largely confirmatory and descriptive. In particular, observations from one manuscript could often benefit the other and the choice of separating the two studies seems rather counterproductive. Below, we separately discuss in more detail the two studies, highlighting where integration of the two would be most beneficial.

Funnel et al.)

In our opinion, this study is the weakest of the two in terms of biological findings and should be integrated into the second study. This work focused on mutational signatures associated with the emergence of structural variants rather than single nucleotide mutations and how these signatures determine chromosomal instability in different models of serous ovarian and triple-negative breast

cancer. In particular, by scWGS, the authors explored these mutational processes within single cells.

The stratification of patient samples in Figure 2 sets the stage for future analyses but signatures and associations were previously shown, e.g. the differences between BRCA1 and BRCA2 mutations (e.g., see previous work from the authors PMID: 30794536 or PMID: 32118208) or the similarity between TNBC and HGSC (PMID: 23000897 or PMID: 23636398). This figure could either be supplemental or substantially reduced to show the stratification of patient subgroups. (Figure 1 should be substantially edited as it currently includes figure panels representing actual data - so it seems - but it is unclear what data.)

The analysis in Figure 3 is rather intriguing as it delves into the differences between BRCA1 and BRCA2 chromosomal instability at the single-cell level. Here we have a few questions:

* First of all, the choice of models is sometimes not sufficiently motivated. Why do the authors provide a heterozygous and a homozygous model for BRCA2, but not for BRCA1?

* Similarly, for BRCA2^{-/-} the authors analyze two independent replicates at different time points, should the differences between the two samples be attributed to the different times they were kept in culture? If so, why BRCA2^{+/-} cells were analyzed at a much earlier time point? Would these cells at a later time point accumulate more chromosomal aberrations? Importantly, what is the replication time of these cells? The role of time-in-culture is not sufficiently discussed and analyzed in these results and the authors should consider the rate of accumulation of chromosomal aberration w.r.t. cell cycle divisions.

Moreover, some results from this analysis are expected, e.g., the association between number altered segments and polyploidy, and could be moved to supplement, whereas it would be great to show here how BRCA1 and BRCA2 SV signature are implemented at the level of single alleles, which is part of the analyses in the second manuscript (Williams et al.).

The authors then move to a more detailed comparison between HRD-dup (i.e. the SV signature associated with BRCA1 mutations) and FBI-associated tumors (another SV signature independent of both BRCA1 and BRCA2) in human samples.

(We should note that although 5 additional samples not exhibiting either of these signatures are shown in the figure, these are never discussed in the manuscript. We suggest removing them from the main figure as they just add more complexity to an already complex figure.)

Here, it is difficult to exactly understand what is the point the authors want to make. Indeed, the features of FBI and HRD-dup signatures have already been characterized, in part by the authors themselves (see for example PMID: 28436987, PMID: 29754820, PMID: 30104763) and these results simply confirm in single cells what was already known from bulk tumor analyses.

In addition, the figure is difficult to read as it misses color legends in individual panels, and some of the colors are used multiple times with different meanings. Results in Figure 4e are expected and results in Figure 4f and 4g are redundant. Additionally, the sample ordering in Figure 4d,e,f,g is not consistent, making it very difficult to match the results in the various panels. In Figure 4j there is a great overlap in the segment length distributions between HRD-Dup and FBI in interiors, while the only tangible difference at termini is between HRD-Dup gains and HRD-Dup losses and FBI gains/losses, so there is no clear distinction between the two signatures even in this case.

Given the previously reported association between FBI and HLAMP (see refs above), the authors search for known oncogenes in highly amplified regions from FBI samples. Here they found a few interesting targets, however, more details are needed.

First and foremost it is unclear how focal these events are. Indeed, instead of providing a genome-wide map of copy number statuses, the authors should zoom in on the affected loci and clearly show the size of these events. Indeed for some of the proposed targets (see BCL11A and MET), copy number gains seem to be arm-events. In these cases, it is difficult to say that these events are selected because they amplify a specific oncogene. Here the authors should have used matched RNA-seq data to show that these oncogenes or their downstream targets are up-regulated when amplified. In the absence of this data, I wouldn't make a big deal about these events.

Furthermore, given the association of the FBI signature with bridge fusion break cycle (BFBC) events (which was shown in previous studies from the authors) this analysis would gain robustness and completeness if combined with the analysis of the same amplicons done in Williams et al. where BFBC events are analyzed in detail. Here it seems the authors did an overall analysis of these events to then split the results into two studies, but each half feels incomplete without the other.

Finally, the authors highlight what they call a serriform pattern for chromosomal breakpoints, where the same/matched structural variant exhibits slightly different breakpoints in different single cells. As discussed by the authors, this pattern was already shown in vitro (PMID: 32299917) but was here first shown in human tumors. Was it named differently before? or simply not named?

This analysis is rather interesting, here we would like however to have a better feeling of the incidence of this phenomenon across different samples. What is the fraction of breakpoints that exhibits this pattern?

The association with FBI, although significant, is rather weak and supported by a limited number of samples. Do the authors think it would be possible to develop an approach to estimate the presence of this pattern from bulk sequencing data (e.g. by detecting reads supporting a breakpoint in different positions)? Pseudo-bulk of their samples could be used for testing. This could be rather intriguing and possibly a way to validate their findings in a larger cohort of samples.

Importantly, it would have been nice here to show this event at the level of single alleles using the approach proposed in the Williams et al. manuscript.

Overall, this study shows interesting features of tumor evolution under the effect of distinct mutational signatures. However, the findings are limited and descriptive and alone do not reach the level of novelty one expects from Nature. The authors could instead consider migrating the most innovative parts to the manuscript by Williams et al..

Williams et al.)

In this manuscript, the authors introduce a new approach to determine the copy number status of

single cells at the level of single haplotypes. They then analyze haplotype-specific copy number alterations in the same datasets introduced above.

At first, the authors present their algorithm, called schnapps, and show its performance on a previously published dataset comprising multiple samples derived from the same patient and analyzed by scWGS. The analysis shows coherent B-allele frequencies (BAF) and variant allele frequencies (VAF) for SNPs, validating the approach. Here we have a few questions:

- * In Figure 1B: given that the BAF is part of the inference procedure of schnapps, isn't it expected that values are distributed around the expected values for each allele combination?
- * the description of the algorithm in the method section could be improved, especially in the use of notation and terminology. Is there a difference between "block allele" and "haplotype block"? What is the difference between the (Ch, Dh) and (Ca, Cb) notation?
- * The code description is great but at least the processed data should be made available for reproducibility (bin-level copy number status)
- * What is the effect of the delta parameter in the transition matrix of the HMM? It is currently set at 0.95, but how was this value determined? How robust are the results of the method to changing this parameter?
- * A comparison with previous approaches is warranted here. The authors have simply described the differences between schnapps and CHISEL and Alleloscope, but they should have actually tried to run these tools on the same dataset and show the differences in performance.
- * The authors do not provide access to the processed data. We understand that patient raw data should be accessed only under proper access control, but having access to the anonymized processed data (for example, bin level copy number estimates of each cell of the study together with their allelic status) could be very valuable. Additionally, we do not understand the inaccessibility of the engineered cell line raw data. Where can we find the processed data to reproduce the analysis presented in the Github repository?

Figure 1d is really nice but we have trouble understanding the color code of the second heatmap. If we read correctly the color legend, the allelic imbalance in the heatmap should range between 0 and 0.5 (only blue hues are shown), but this is in contradiction with Figure 1c, isn't it?

The analysis of clonal and subclonal events that follows (Figure 2) is an intriguing negative result, i.e. subclonal events do not show evidence of selection. A question that comes to mind is whether there is any association here with the genetic context.

Is there any difference between BRCA1/2 mutated tumors and others?

Most notably, given this information is already available to the authors, what about the association with mutational signatures?

This analysis feels a bit out of scope here, it does not focus on allele-specific changes and seems rather more connected to the other manuscript by Funnel et al.. Combining the two would make a much nicer chapter.

The analysis of parallel events is also quite interesting but very difficult to follow in Figure 3. Color legends should be more clearly shown in association with the figure panels and if possible I would refrain from using the same/similar colors for total copy number and allelic imbalance. A suggestion: would it be possible/simpler to show two separate heatmaps for the estimated copy number status

of each allele?

Also, it is unclear from the methods what are the requirements to call a parallel gain or loss. It looks like several events are quite different in size between the two alleles, although intersecting in a region (e.g. Figure 3g,h,i). The authors should clarify how the overlapping of changes in maternal and paternal haplotypes are computed.

The correlations in Figure 3k are rather weak with points clearly deviating from the line of fit. Anything to note in these samples? Is there an association with polyploidy or mutational signature?

Was the UMAP in Figure 3m generated using expression data for all genes or only for those in chr 2 (or 2q)? If the former, the authors cannot conclude that the clustering shows convergent phenotypes as it could be driven by a myriad of other signals independent of chr. 2q. Can the authors actually test for allelic expression imbalance for genes in the lost part of the chromosome? Bulk RNA-seq data could be used for that.

Lastly, the remaining figures of the manuscript are highly curated reconstructions of specific events. Whereas the effort to reconstruct some of these events is commendable, the value seems rather anecdotal. The analyses in Figures 4 and 5 could be combined and should be put in the context of the mutational signature that generated these events. Figure 6 is unreadable and, again, as much as we understand the difficulty of representing this data, it is unclear what we can learn from it.

In conclusion, we feel that this manuscript (Williams et al.) brings forward a new methodology of possibly broad applicability, and its potential is nicely shown through multiple analyses revealing high allelic imbalance in chromosomal instability and patterns of convergent evolution between events in the homologous chromosomes. These results could benefit from the mutational signature analysis performed in Funnel et al., which here are completely ignored.

Overall, we strongly encourage the authors to propose a substantially revised and harmonized combination of the two manuscripts into one, streamlined from the many confirmatory results and focused instead on what new we can learn from these scWGS datasets and computational approaches.

Author Rebuttals to Initial Comments:

Point-by-point response to reviewer comments:

The impacts of single cell genomic variation induced by mutational processes in cancer

Table of contents

1. Summary	2
2. Reviews for Funnell et al.	3
Referee #1	3
Referee #2	11
Major comments	12
Minor comments	16
3. Reviews for Williams et al.	19
Referee #3	19
Minor Comments:	32
Referee #4	34
Major comments	35
Minor comments	46
4. Joint review of both papers	48
Referee #5	48
Funnell et al.)	48
Williams et al.)	55

1. Summary

We thank the reviewers for careful reading of the original submissions and for constructive and insightful comments which have helped to considerably improve the study. In response to the comments, we have substantially revised the submissions, with the strongest findings joined together in a single manuscript. Below, we include point-by-point responses to the 5 reviewers whose comments were solicited across the two original submissions. The main approach we took when merging the papers was to integrate cell-to-cell variation of high level amplifications, allele-specific alterations and serrate as ‘foreground’ processes into the context of ‘background’ mutational processes found in bulk tumors, focusing on HRD-Dup, FBI. The highlights of the **revised single manuscript** are:

- 1) The characterization of 3 ‘foreground’ mutational patterns derived from cell-to-cell variations identified through single cell whole genome sequencing: (a) Clone-specific high level alterations impacting known oncogenes; (b) Parallel copy number alterations on different alleles; c) Serrate structural variations (SSV)
- 1) All 3 mutational patterns are introduced in the *in vitro* data in mammary epithelial cells with induced genomic instability through *TP53* and *BRCA1/2* ablation. They are then subsequently described in human tumors from high grade serous ovarian and triple negative breast cancer patients.
- 2) We have analyzed the evolutionary and phenotypic impacts of foreground processes and present these in context of both *in vitro* and human tumor data, including a more comprehensive integration of matched scRNASeq data from the same samples. The integration of scRNASeq demonstrates the phenotypic impact of variable high level amplifications and mono-allelic expression of the parallel copy number alterations.
- 3) We have added an extensive benchmarking of our allele specific copy number detection method to other related published methods and demonstrate that we can identify events both at higher genomic resolution (0.5Mb vs 5Mb) and higher cellular resolution (identifying aberrations in single cells) compared to other methods which is central to the biological findings we present. In addition, benchmarking at several other levels of granularity including phasing of mutations, concordance with bulk WGS and haplotype phasing with long-read ONT data is now incorporated. We re-emphasize however that the presentation of a new method was not the main contribution of the paper and focus on biological findings derived from using the method.
- 4) The findings of clone and cell-specific high level amplifications and the SSV patterns are now presented with the additional granularity of allele specificity, yielding a more integrated synthesis of the two original papers. Allele specificity is also now included in the context of the *in vitro* data and in the context of mutational processes in the tumors.
- 5) We have added orthogonal features of the mutational patterns in order to more richly describe their nature and draw conclusions about their mechanism and impact. Combining the haplotype and structural variation inference we illuminate that i) clone-specific high level amplifications are impacting gene expression programs ii) clone specific high level amplifications are generated by several different rearrangement processes; iii) SSV and parallel copy number alterations contribute significantly to evolutionary diversity within cell populations

We have added **additional data** to corroborate key points originally raised by the reviewers, within scope of the merged manuscript as follows: (i) An additional 11,946 single cell whole genomes from an additional 24 patient samples or cell line passages (including *BRCA1* heterozygous mutant), which brings the total number of single

cell whole genomes analysed to 35,875; (ii) An additional 85,870 single cell transcriptomes from matched samples to detail the transcriptional impacts of the foreground patterns; (iii) Immunohistochemistry of PDX tissues illustrating clonal variable copy number driven *KRAS* expression; (iv) Long read sequencing from ONT to confirm structural variants and haplotype phasing from 4 patients and WT cell line.

Accordingly, the presentation has been altered substantially. Many of the original figures and claims have been either moved to supplemental or omitted as they are replaced with merged figures and are thus no longer relevant. Following a discussion with the editor, we did not add drug sensitivity/impact experiments as we agreed this was beyond the scope of an already fairly dense paper, especially in the merged format. Given the complexity and volume of data and the point-by-point response, we included some figures that are only in the rebuttal document and not in the main text. These are indexed by **R***. All other figures below are indexed by how they appear in the manuscript, relevant panels are included in this document. The responses document is indexed and the TOC hyperlinked for ease of navigation.

We appreciate the point-by-point response is lengthy and thank the reviewers for their time in advance.

2. Reviews for Funnell et al.

Referee #1

In this manuscript, Funnell and O'Flanagan et al. use bulk WGS and single-cell WGS (scWGS) to study how single nucleotide mutations (SNVs), copy number alterations (CNAs) and structural variants (SVs) shape the genomic heterogeneity of high-grade serous ovarian cancer (HGSOC) and triple-negative breast cancer (TNBC). The authors combined state-of-the-art experimental and computational tools to co-analyze the mutational patterns across the analyzed tumors. Next, they sequenced an isogenic in vitro model of immortalized mammary epithelial cells with various combinations of TP53, BRCA1 and BRCA2 mutations, and found distinct mutational signatures for each of these genotypes. Finally, the authors generated PDXs and performed scWGS of 22 primary tumors and PDXs, focusing on the comparison of tumors from two specific mutational groups: HRD-Dup (enriched for BRCA1 mutation and patterns reflecting short/medium tandem duplications) and FBI (enriched for CCNE1 amplification and foldback inversions). They report different CNA patterns between these two groups, consistent with distinct CNA acquisition processes. FBI tumors accrued more gains than HRD-Dup tumors, and also exhibited more high-level amplifications (HLAMP). Intra-tumor variation in the exact genomic breakpoints of specific CNAs was observed and termed 'serriform structural variation' (SSV), and this phenomenon was also more common in FBI vs. HRD-Dup tumors. The FBI tumors therefore seem to be more structurally unstable, suggesting higher evolvability of potential clinical relevance.

Overall, this study is a tour de force of current cancer genomics. However, I have several doubts regarding its suitability for Nature. First, the biological novelty of the findings and the conceptual advance over current knowledge is rather moderate. Second, in its current form the paper is oriented toward a specialized audience, and the writing ought to be somewhat simplified if it were to target a wider audience. Third, the three parts of the paper (bulk sequencing of tumors, sequencing of engineered cell lines, single-cell sequencing of tumors) seem to be a bit detached: the BRCA1 vs. BRCA2 comparison (which is highly interesting) is limited to the in vitro analysis, whereas the HDR-Dup vs. FBI comparison and the description of the SSV phenomenon are studied only in the clinical data; a more integrative analysis would make the results and conclusions stronger and more exciting.

In addition, below are a few specific suggestions:

Fig.2:

* While the vast majority of tumors have an inactivating TP53 mutation, not all of them do. However, the TP53-WT tumors do not cluster separately from the TP53-mut tumors. Do these tumors have mutations in other genes of the p53 pathway? A direct comparison of the TP53-WT vs. TP53-mut tumors would be of interest.

We note that the bulk analysis has been de-emphasized in the refactoring of the manuscript. However, we nevertheless inspected the non-TP53-mutated tumors for putative driver mutations in TP53 target genes collected from the literature¹. We found only one case without driver mutations in any of these genes. We also

found that there was no significant difference (all BH-adjusted p-values > 0.05) in mutation signature probabilities between *TP53* mutated and non-mutated tumors. See **Figure R1**.

Figure R1

a Oncoprint of 309 tumors profiled with whole genome sequencing. b Signature probabilities for each feature for *TP53*mt and wt.

* The analysis finds that “despite skewing towards accumulation of losses at the single cell level, human cancers nevertheless exhibit tandem duplication enrichment at the bulk genome level in BRCA1 associated cancers, but enrichment of interstitial deletions in BRCA2 associated cancers.” Could this reflect different evolutionary pressures in vitro and in vivo?

This sentence in its original form has been removed in the resubmission. We agree with the reviewer this is an interesting question, but we feel to adequately address this would require experimental systems beyond the scope of this contribution. In addition, tandem duplications and interstitial deletions associated with loss of *BRCA1* and *BRCA2*, respectively, are typically less than 100 kbp long (Extended Data Fig. 6), and smaller than the 500 kbp genomic resolution used in this study to detect copy number variation. Accordingly, we leave the resolution of this provocative question for future work.

* The increased levels of CNAs found in TP53-null/BRCA1-null and TP53-null/BRCA2-null cells in comparison to TP53-null cells are interesting. Is this trend also statistically significant in the bulk-population tumor data shown in Fig. 2? And in the single-cell tumor data shown in Fig. 4? (It’s hard to tell from the plots.)

We have now calculated the proportion of the genome + the number of genomic segments subject to gains/losses/loh for each tumor as a function of genotype in our bulk WGS cohort. We find that TP53wt tumors consistently have fewer segments and a smaller proportion of the genome altered relative to TP53-null. There is a trend for further increases in instability in the TP53-/- BRCA1-/- but this does not attain statistical significance, see **Figure R2**. Due to space constraints we do not include this result in the manuscript but do include it here for review purposes.

Figure R2
Distribution of the number of segments and proportion of the genome altered as a function of genotype in bulk WGS.

* The TP53-null/BRCA2-null show a very specific aneuploidy pattern – loss of chromosomes 2q and 14 (Fig. 3e). Is the same pattern seen with these specific mutations in the primary BRCA tumors?

While we do observe some tumors with 2q + 14 loss in our bulk tumor cohort, these alterations do not appear to be enriched in the BRCA mutant tumors, see **Figure R3** for the landscape of CNA's stratified by BRCA mutation. We would speculate that this specific aneuploidy pattern likely arose by chance in our *in vitro* system. Due to space constraints we do not include this result in the manuscript but do include it here for review purposes.

Figure R3

Recurrent gains and losses in the bulk tumor cohort as a function of genotype. Gains shown in red, losses in blue.

* How was the rate of chromosome mis-segregation determined (e.g., Fig. 3h)? This should be briefly clarified in the text.

We have now referred the reader to the Methods section for details on determining mis-segregation events and updated the text accordingly (line 100) :

“Per-cell copy number distributions exhibited progressive increase in CNA rates as a function of TP53 and BRCA1/2 loss (Fig. 1b-e). In addition, we observed increasing whole genome polyploidy (Fig. 1f), chromosomal missegregation (Fig. 1g, Methods) and per-cell alteration counts of TP53-/-, BRCA2-/- and BRCA1-/- respectively over WT cells.”

The text in the Methods section now reads (line 265 methods document):

“The approach taken to identify putative chromosome missegregation events is similar to Laks et al. Cells were split into groups corresponding to their clones. Clone copy number profiles were generated for each clone. Cells with ploidy not equal to the clone consensus profile were normalized to match the clone ploidy. Cell copy number profiles were compared to the clone copy number profile for the matching clone to which the cell belongs. The result was assignment of an offset value for each genomic bin in each cell, that represented the copy number difference between cell and clone-level consensus profile. For each chromosome in each cell, if a particular copy number difference (i.e. -1, 1, etc.) represented at least 75% of the chromosome then we labeled that chromosome as having a missegregation event.”

Fig. 6:

*** Is the ‘serration’ pattern enriched for any genomic feature (nucleotide composition, distance from centromere/telomere, etc.)? It is likely that there is no sufficient statistical power for such an analysis with the current cohort size, however maybe there is an interesting trend? Also, could this be analyzed using previously published scDNAseq data from tumors, generated by the authors or by other groups?**

We thank the reviewer for the suggestion, we looked but did not find any significant association between SSVs and distance from centromeres or telomeres, or with replication timing. We do observe other relationships of SSVs, documented in the revised manuscript. We attempted to examine nucleotide composition by identifying all split reads mapped to the genomic bins adjacent to cell-specific copy number change points, as this could potentially identify the actual cell-specific breakpoint location. However, identifying cell-specific rearrangement SVs remains an open problem and is fraught with confounding levels of likely false positive predictions. We have identified this as a key area for algorithm development in future work. Accordingly we leave the identification and study of cell-specific SSV breakpoints to future research. We emphasize that our study was set up to compare the cell-to-cell variation patterns in HRD and non-HRD cancers (as presented in **Fig. 6d**) - we are not aware of other datasets where this could be adequately addressed.

*** The authors analyzed a few thousand single cells in vitro. Can the SSV pattern be observed in these data?**

Thank you for the suggestion. We do indeed observe this pattern in our *in vitro* data, examples shown below in **Main Figure 2j**. In the restructuring of our paper we first characterize 3 novel mutation patterns (including the SSV pattern) in our *in vitro* data before going on to analyze their prevalence and impact in the PDX data.

Figure 2j Examples of 'serrate structural variants' SSVs found in our cell line data.

Referee #2

We highlight that in order to meet the request to merge the two submitted manuscripts into a single submission, we have de-emphasized many of the previous observations, de-prioritizing those that could be construed as lacking novelty. Accordingly many of the comments regarding the bulk data are less relevant in the revised paper to appropriately highlight the single cell observations. For completeness we have addressed the comments below, but suggest that responses to comments on the bulk observations be down-weighted in the re-assessment of this work.

Funnell and colleagues investigate the consequences of SNV and SV mutational signatures on genomic instability at single cell resolution. They apply their previously developed computational algorithm that jointly infers signatures from both SNV and SV mutation types on bulk WGS samples from ovarian (HGSC) and breast (TNBC) tumors. From these results, they delineate clusters of tumors associated with specific SNV and SV features that were also found to harbor genomic alterations for individual genes of interest (e.g. HRD tandem duplication phenotype in BRCA1 mutated tumors and HRD deletion phenotype in BRCA2 mutated tumors). The authors further examine the impact of compound biallelic loss of TP53 and BRCA1/2 at the single cell level in vitro with CRISPR KO models where biallelic loss of BRCA1/2 induced greater genomic instability and aneuploidy compared to a background of biallelic TP53 loss. Subsequent single-cell WGS profiling of PDX tumors showed HRD-tandem duplicator phenotypes with more even distribution of losses and gains vs foldback inversion (FBI) phenotypes. FBI tumors accrued copy number gains at a higher rate, with high-level amplifications co-occurring with FBI tumors in known driver genes such as ERBB2, KRAS, MYC, CCNE1, and FGFR1, with clone-specific variation in amplitude of these high-level amplifications, suggesting therapeutic implications.

Although the impact of structural variations together with mutations at the single-cell level on genomic instability is certainly interesting, the therapeutic significance of the manuscript is limited as all results are associative and not supported by pharmacological experiments. Clinical significance is limited to observations of modest prognostic effects which are largely already known and there is no data regarding specific drug responses. The issue for oncologists is TNBC and ovarian cancer treatment is only partially effective, patients can be given more than 5 different chemotherapy agents with different interaction with DNA repair, DNA replication and cell division processes. While the authors speculate about therapeutic effects of investigational drugs they do not conduct any studies of this type. They do have access to TNBC PDX and so this is something the authors could consider to increase impact.

Another under-discussed issue is the effect of different mutational mechanisms on the immunogenicity of individual tumors. Finally there is also an issue of novelty since the group's SNV/SV algorithm was applied to HGSC and breast tumors in a previous publication where association with BRCA1/2 was discussed.

We appreciate this comment and agree that investigating how the cell-to-cell variation we identify in this study impacts treatment response would be of considerable interest. We have recently published data on this topic in an initial study (Salehi et al, Nature 2021²) covering cisplatin. We point out that Salehi *et al* constituted a multi-year series of experiments to robustly define the therapeutic responses and fitness of single cells/clones

in PDX and therefore feel this is well beyond the scope of the current study (see also comment below), where our focus is on first identifying and characterizing the extent of genome instability at **single-cell resolution**. We also emphasize that previous work³ has been solely in bulk data. Here we structure the study and contrasts from bulk 'background' mutational processes (e.g. Fig 3a - described in the next comment) which enable the comparisons between HRD-Dup and FBI tumors at single cell resolution, identifying and characterizing the nature of the foreground processes. All 6 main figures now focus almost exclusively on the novel single cell data to minimize any conception of overlapping material with previous work.

In relation to immunogenicity we would point the reviewer to a complementary study available as a preprint where we use single cell RNAseq data to profile the tumor microenvironment as a function of mutational process⁴ in patient samples, currently in revision. Given the request to merge the two submitted manuscripts into a single submission, discussion of immunogenicity is left to the Vazquez-Garcia et al paper⁴ with a fulsome discussion and exposition of the tumor microenvironment in patient samples with intact immune systems. We anticipate this work will appear in the next six months.

Major comments

1. Their previous publication (PMID: 30794536) applied the same computational algorithm already to HGSC and breast samples where association with BRCA1/2 mutations were also examined. How much of Fig 2 in the current submission represent new observations?

This analysis includes 128 new tumor samples not included in any previous publications and more importantly describes how the mutational subtypes are distributed in TNBC (the previous papers were focused on HGSC). We note that we used this analysis in order to stratify tumors into mutational subtypes and from this we generated PDX models and single cell whole genome sequencing across the spectrum of mutational subtypes. The relationship of the full 'bulk sequencing' cohort to the samples sequenced with single cell WGS is depicted in **Fig 3a** to orient the reader. We agree that the bulk data in and of itself does not contribute significant novel insight beyond their initial description in TNBC and have therefore moved this analysis to Extended Data Figure 7 since it nevertheless is an important element of the study design and the provenance of the samples presented in the single cell sections. In addition, the bulk data serves as a useful validation of the single cell allele specific analysis as depicted in **Fig 3b**.

Figure 3

a) UMAP of metacohort signature probabilities. Lines connect DLP-pseudobulk to their bulk data counterpart. b) Correlation of proportion of the genome that is LOH between DLP-pseudobulk (horizontal) and matched bulk whole genome sequencing (vertical).

We have reworked the text in this section to clarify (line 178):

“To identify appropriate patient tumor samples for this comparison, we first constructed a ‘meta-cohort’ of 309 patients comprising 170 HGSCs and 139 TNBCs with bulk tumor-normal paired WGS to infer the distribution of established mutational processes, of which 106 TNBC and 22 HGSC genomes were newly sequenced for this study and combined with published HGSC^{8,29–31} and TNBC^{11,16,32–34} datasets (Extended Data Fig. 1).”

2. It is not clear how gene list for mutation/amp analysis in Fig 2 were picked. Were these the top most mutated or amplified genes in respective group (Stratum). An unbiased enrichment analysis is warranted.

We have updated the bulk tumor library oncoprint (**Extended Data Figure 7** in the revised manuscript) to include genes with enrichment of mutations, amplifications, or deletions in any stratum compared to all other samples. Genes included in the enrichment analysis were chosen by adding those with known biological/clinical relevance to the BROCA Cancer Risk Panel genes (<https://testguide.labmed.uw.edu/public/view/BROCA>).

Extended data figure 7

Meta-cohort signature analysis of 139 TNBC and 170 HGSC bulk whole genomes. Mutation status of key genes shown above heatmap, with stratum enrichment status shown as coloured bars on left.

3. The authors mention HRD signature enriched in BRCA1 mutation and CCNE1 amplification in FBI clusters , but no statistical basis of enrichment or associated p-value reflecting how strong or weak the enrichment is shown.

We appreciate this comment and have updated this analysis as shown in Extended Data Figure 7. Genes were selected with known biological relevance or presence in the BROCA Cancer Risk Panel genes (<https://testguide.labmed.uw.edu/public/view/BROCA>). Each gene was tested for enrichment of mutations, amplifications, or deletions in any stratum compared to all other samples. Tests were performed using the hypergeometric test, and p-values corrected using the Benjamini-Hochberg procedure. We have included the stratum gene mutation enrichment analysis results as **Supplementary Table 5**. *BRCA1* mutations and *CCNE1* amplifications are enriched in the HRD-Dup and FBI strata with Benjamini-Hochberg adjusted p-values 8.45e-12 and 1.29e-12, respectively.

4. Some clusters seem to be rich in higher age patients in FBI and L-Del clusters. A location and telomeric repeat analysis is required to ensure these genomic aberrations are not age or telomeric loss related.

We compared telomere lengths in both tumor and normal samples between cancer and signatures types, see **Figure R4**. While TNBC tumor samples had longer telomere lengths on average, possibly indicating active telomere lengthening, we found no significant differences between signature strata. Telomere lengths in units of base pairs were inferred from bulk WGS data using Telomerecat⁵. Bonferroni p-value correction was used to determine which groupings were significant. Each point represents the telomere length of a bulk WGS sample. The left column represents tumor and right column represents normal.

Figure R4

Telomere length distributions inferred from bulk whole genome sequenced tumor and normal data. Each data point is a single tumor/normal sample.

5. The authors associate FBI tumors with high rates of copy number gains/high level amplification events, but seems counterintuitive given their previous findings of deletions associated with FBI. This discrepancy should be discussed.

Our updated signature strata now includes an FBI type which contains some cases also enriched for the L-Del (Large Deletion) SV signature. We found higher rates of polyploidy and segmental copy number gains in FBI cancers, which may provide greater opportunity for, and tolerance of, large deletion-type SV breakpoints due to additional genomic redundancy, which were detected with a copy number-agnostic algorithm. This is discussed in several figure panels in this resubmission and included in this passage in the main text (line 206):

"In addition, FBI tumors accrued gains at a significantly higher rate than HRD-Dup, with higher skewing of the gain/loss ratio (4.9 vs. 2.1, p-value: 0.04, Fig. 3h, Extended Data Fig. 9b,c). This

was more pronounced when considering baseline ploidy of the tumors (Extended Data Fig. 9d, p-value: 0.0012). Indeed, higher rates of polyploidy and segmental copy number gains may provide greater opportunity for, and tolerance of, large interstitial deletions found in some FBI cancers (Extended Data Fig. 6b)."

6. The authors state the HRD-tandem duplicator phenotype with BRCA1 mutations leads to a higher rate of genomic instability as compared to HRD-deletion phenotypes with BRCA2 mutations which may explain differences to cisplatin treatment. The cisplatin pharmacological experiments in vitro should be extended to in vivo models.

We appreciate this comment, however, in the refocused merged manuscript we have removed the cisplatin treatment experiment, which was only originally included to show the isogenic epithelial knockout had an expected phenotype. Since this is not the main point of our findings in the revised manuscript, we refer the reviewer to recent work from our group in this area² published in *Nature*. We note that this latter study was a significant effort that took place over a multi-year period to reach definitive conclusions on polyclonal nature drug response. We suggest this is out of scope of the current 'merged' manuscript due to space constraints To keep a cohesive narrative we therefore elected to remove the pharmacological experiments from the merged paper.

7. To further increase clinical and therapeutic relevance, pharmacological inhibition of targets such as ERBB2, FGFR1, etc, that were found to have clone-specific high-level amplifications should be assessed for FBI tumors.

We do appreciate the nature of this comment and refer the reviewer to our response on the previous comment. To exemplify the biological relevance of the clone-specific high level amplifications we have conducted 3 additional analyses:

1. We have demonstrated the variation in gene expression of the affected genes with co-registered scRNASeq data, see new **Fig 4**
2. We have conducted immunohistochemistry assays to demonstrate the variation in expression at the protein level. We show that in both the PDX and the matching source patient tissue the pattern of highly variable KRAS protein expression is evident, see new **Fig 4**
3. We have integrated structural variant classes to reveal that most high level amplifications in HRD tumors accrue with different mechanisms than FBI, see **Extended Data Figure 11**

Minor comments

1. Add brief description of FBI to introduction section.

We have refactored the introduction to include the following statement (line 64):

"For example, breakage-fusion-bridge cycles (BFBC) and homologous recombination deficiency (HRD) are endogenous mutational processes that accrue structural variations with specific patterns

including tandem duplications, interstitial deletions and foldback inversions (FBI) that generate high level copy number amplifications^{2,7,8,10.}

In addition, we have included the following sentence when we introduce the mutational subgroups in the results section (line 182):

*"We applied a previously described correlated topic model machine learning approach (MMCTM)⁷ and recapitulated previously described groups of tumors. Distinct structural-copy number mutational features in both TNBC and HGSC were observed as follows: HRD-Dup (enriched in small tandem duplications and BRCA1 mutations), HRD-Del (enriched in deletions, BRCA2 mutations), FBI (enriched in foldback inversions and CCNE1 amplification), TD (enriched in large tandem duplications, CDK12 mutations) (**Extended Data Figs. 6,7, Supplemental Tables 4,5**)."*

2. How were tumors exhibiting strong score for multiple clusters treated ? For example there are many samples where there appears to be equally strong signal for FBI/INV and L-DEL but these are categorized as L-Del.

We agree with this observation and have resolved this in the latest sample clustering, with an updated FBI group enriched for CCNE1 amplification, including cases with FBI/Inv and/or L-Del signal as shown in **Extended Data Figure 7**. We also note that the FBI/Inv signature includes some large deletions as shown in **Extended Data Figure 6**. The signatures inference and updated clustering procedure is described in detail in the **Methods** section **HGSC & TNBC metacohort signature analysis**. The clustering method text is as follows (Methods line 181):

"Samples were clustered by first applying UMAP to the normalized signature probabilities for the HRD SNV signature and all SV signatures with $n_neighbors = 20$ and $min_dist = 0$ to produce 2-dimensional sample embeddings. Next, HDBSCAN was run on the sample embeddings with $min_samples = 5$, $min_cluster_size = 5$, and $cluster_selection_epsilon = 0.75$ to produce the sample clusters (strata)."

Extended Data Figure 6b

SV mutation signatures. SV types are DEL: interstitial deletions, DUP: tandem duplications, INV: inversions, FBI: foldback inversions, TR: translocations. SV signature labels are S-Dup: small duplications, M-Dup: medium duplications, L-Dup: large duplications, S-Del: small deletions, L-Del: large deletions, Clust-FBI: clustered foldback inversions, Clust-SV: clustered other structural variants, Tr: translocations, FBI/Inv: foldback inversions and inversions.

3. Reviews for Williams et al.

Referee #3

This paper studies allele-specific copy number alterations (CNA) in the tumor setting. The authors developed a new tool, called schnapps, that detects allele-specific copy number alterations from low-coverage single cell DNA sequencing data. By applying schnapps to scDNA-seq data from a set of 20 high grade serous ovarian and triple negative breast cancer samples, they were able to study the frequency of copy number changes in the cancer genome. They reported strong evidence for positive selection of alterations in specific regions and parallel evolution through convergent alterations in both the paternal and maternal alleles. They showed, quite convincingly, that some of the intra-tissue diversity of some of the events are generated through breakage fusion bridge cycles.

The paper claims to make novel contributions in: (1) the development of the schnapps method, and (2) the biological findings pertaining to CNA's role in tumor evolution. Here are my assessment of the rigor and novelty of its findings along these two dimensions:

We thank the reviewer for their constructive comments. In our revised manuscript we have included additional analyses validating the results of our allele-specific copy number method as well as comparisons to previously published methods. In addition, our new manuscript combines the results of both Funnell *et al* and Williams *et al* strengthening the biological conclusions we present. We note that we have changed the name of our tool from schnapps to SIGNALS at the request of the editor, and use SIGNALS in our reply and in the figures.

We first discuss the overall summary of our benchmarking efforts before going on to address the specific comments below. Our results suggest that while the results of SIGNALS are broadly similar to both chisel and alleloscope (highest similarity between SIGNALS and chisel), SIGNALS has the following advantages.

1. Increased resolution of haplotype specific copy number calls down to 0.5Mb
2. Increased accuracy of haplotype specific copy number calls in rare cell populations
3. Relative to alleloscope, our pipeline better resolves baseline cell ploidy, which is often underestimated.

We find that chisel performs equally well in this regard.

Taken together these present the following significant advantages in terms of the biological findings in our study:

1. The ability to identify rare parallel CNAs and therefore compute rates of sporadic gains and losses
2. Increased allelic resolution in focal high level amplifications, enabling reconstruction of possible mechanisms such as breakage fusion bridge.

We now provide a detailed **Supplementary Note** which includes extensive validation and benchmarking of the method. We include the most relevant points that directly relate to reviewer comments in our reply below, so we note that many figures indexed by R* are included in the **Supplementary Note**.

1. On the methodological contributions:

A. Schnapps is conceptually very similar to CHISEL (Zaccaria and Raphael, 2021, Nature Biotechnology 39, 207-214), in that local smoothing and phasing is conducted to aggregate information from RDR and BAF (Schnapps uses a HMM, while CHISEL uses a bin-wise smoothing). Although Schnapps claims to be higher resolution (using smaller 500KB bins), the bin-size in CHISEL can be user adjusted. There was no rigorous benchmarking of Schnapps against CHISEL.

B. Other than CHISEL, the recently published Alleloscope (Wu et al., 2021, Nature Biotechnology DOI: 10.1038/s41587-021-00911-w) also has the same goals and operates on the same data type. Other than scDNA-seq data, also Alleloscope works for scATAC-seq data.

We agree that CHISEL is conceptually similar, however in our experience the resolution was not sufficient for the downstream analysis we presented in our study. We have now run both CHISEL and alleloscope on 11 of our datasets. We chose 11 datasets to encompass a range of tumor types and baseline ploidies. We ran CHISEL at both a recommended bin size of 5Mb and a smaller bin size of 0.5Mb (chiselsmall in the plots below). In addition, we also ran our pipeline on breast cancer patient S0, which was studied extensively in the chisel paper. We obtained the chisel results for this data from the original publication.

A summary of our benchmarking is shown in **Figure R5**, where we found the following:

1. SIGNALS and CHISEL with 5Mb were the most similar with an average minor allele copy number similarity of more than 90%.
2. Alleloscope and chisel with small bin sizes were the least similar.
3. The similarity between single-cell pseudobulk and matched bulk copy number profiles (inferred with Remixt⁶) was highest for CHISEL and SIGNALS. Some Alleloscope samples had large discrepancies due to incorrect baseline ploidy assignment.
4. CHISEL and SIGNALS had the smallest distance between expected and observed variant allele frequency peaks derived from somatic SNVs
5. We observed a larger proportion of SNVs with VAF > 0.95 in non-LOH states in alleloscope and CHISEL (0.5Mb bins) relative to SIGNALS and CHISEL (5Mb bins). Mutations with VAF > 0.95 in non-LOH states would be indicative of incorrect calls.

Together, these results suggest that CHISEL with 5Mb bins and SIGNALS (0.5Mb) perform equally well at estimating allele-specific copy number. We did not observe a statistically observable difference in any of our metrics for these two methods. We note however that these metrics will be dominated by copy number alterations at high frequency and will be insensitive to the accuracy of rare aneuploidies.

Figure R5 Comparison of methods

a Similarity of total and minor allele copy number calls between methods across 12 datasets. Each bar represents the mean per cell similarity, lines indicate the 90% quantiles. Comparisons are ordered by the average most similar (SIGNALS + CHISEL) to least similar (alleloscope and chisel with 0.5Mb bins). b Difference in bulk derived vs single cell pseudobulk derived ploidy estimates. Each data point is one of the 11 samples with a matched bulk WGS (not available for OV2295). c correlation between the minor allele copy number in bulk vs single-cell derived pseudobulk d Distance of expected VAF peak to empirically observed VAF peaks. e Proportion of SNVs in balanced or unbalanced states that have VAF > 0.95 indicative of missassigned states. In each boxplot panel, the numbers and labels indicate the p-values from pairwise wilcoxon tests.

As chisel (5Mb) and SIGNALS appeared to perform similarly in our benchmarking assessment, we focused on comparing the results of these two methods in detail, with particular attention on rare aneuploidies. We used the data originally published in the chisel paper (patient S0), reasoning that this would represent the best use-case scenario. Overall we found that the results from SIGNALS and chisel had a similarity of 93%. The remaining 7% of inconsistent calls could be attributed to differences in calls of rarer aneuploidies present at low cancer cell fractions and differences in baseline ploidy assignment.

Assessing the correctness of rare CNAs is challenging, the metrics we report above will only be informative of clonal CNAs (bulk tumor comparison) or CNAs present in high-frequency subclones with sufficient cumulative coverage for mutation calling (VAF distributions). We therefore resort to checking consistency of BAF distributions and the number of times the haplotype phase switches within a particular CNA, which would be indicative of incorrect phasing and calls.

We found numerous examples where CHISEL switches between two states - for example 2|1 and 1|2 in total copy number 3 states. In such events, the BAF distributions were centred around 0.5 rather than the expected $\frac{1}{3}$ or $\frac{2}{3}$. Conversely, SIGNALS appeared to perform better in these scenarios, with fewer switches and expected BAF distributions, **Figure R6**. Using the number of switches per chromosome event as a metric, we found that SIGNALS consistently had fewer switches than CHISEL This was particularly evident when the proportion of cells harbouring a CNA < 10%, **Figure R6**.

Figure R6

a Heatmap representation of all cells from section E patient S0 from chromosomes with parallel CNAs. Left, total copy number from HMMcopy pipeline, right haplotype specific states from SIGNALS b) Same data as in a) with states inferred using CHISEL c,d Two single cells with mirrored allelic imbalance in chromosome 9 inferred with SIGNALS. e BAF distributions of cells with allelic imbalance in chromosome 9 from SIGNALS. Black horizontal line indicate the expected means of the distributions. f,g The same 2 single cells inferred with CHISEL showing inconsistent BAF given triploid state and switching of phase across the chromosome. h BAF distributions of cells with allelic imbalance in chromosome 9 from CHISEL i The number of switches in phase in individual cells across chromosomes. Each data represent the number of switches in a chromosome in an individual cell. Number across the top indicate the number and fraction of cells that are aneuploid in that chromosome according to SIGNALS.

C. Although the authors claim that Schnapps works for scRNA-seq data, they only applied it at the chromosome arm level, and for one event. More extensive benchmarking and illustrations would be needed to establish its usefulness beyond scDNA-seq.

We have now adapted our scRNAseq algorithm to use genomic segments rather than chromosome arms. Genomic segments are derived from pseudobulk scDNAseq profiles, segments <10Mb are merged with adjacent segments. Phased SNPs are then genotyped in each single cell transcriptome and counts are merged per segment to compute allelic imbalance. See the **Supplementary Note** for further details.

To further validate our ability to recover allelic imbalance in scRNAseq we included previously published scRNAseq data² (SA1035, SA609, SA535) and generated additional data from n=11 samples with matched scDNAseq. We first investigated allele imbalance in chromosome 17. Chromosome 17 would be expected to be clonally homozygous in 100% of these samples. Allele imbalance in scRNA confirms this and is consistent with allele imbalance from scDNA. We also found that both the average and the variance of the BAF per segment were highly correlated with values from scDNAseq, **Figure R7**. A summary of this figure is included in main **Figure 5**.

Figure R7

a Distribution of BAF within chr17 per sample, for each violin plot, the distribution of BAF in scRNA is shown on the right in red and the distribution in scDNA is shown in blue on the left. b The mean BAF per segment per sample in DNA vs RNA c The variance of BAF per segment per sample in DNA vs RNA

D. No formal validations of the detected CNA events was performed to establish the accuracy of schnapps. In contrast, CHISEL corroborated their events using somatic SNVs and the spatial distribution of inferred clones. Wu et al. used matched linked-reads data to establish that the mirrored subclones are real and not an experimental or computational artifact. Wu et al. also used data downsampling to benchmark CHISEL and Alleloscope at varying coverage. Thus, this paper is not on par with published papers in terms of rigor.

We thank the reviewer for bringing this to our attention. We have now extended the formal validation section in **four distinct ways** which we contend exceed the state of the field in terms of rigor.

- First, similarly to CHISEL, we also use somatic SNVs to validate our allele specific calls. We assigned all SNV's to haplotype-specific states in single cells and computed variant allele fractions (VAFs) by summing read counts across cells. The distribution of VAF's we observed supports our haplotype-specific inferences, where we consistently observed a VAF peak at 1.0 for LOH states, a peak at 0.5 for balanced states and expected modes in the distribution corresponding to pre-and-post CNA acquisition in unbalanced states. For example, modes of 0.33 and 0.67 for 2|1 states, see **Figure 3c**.

Figure 3

c) VAF distributions per haplotype-specific state. Shown are distribution across the whole cohort of single-cell data.

- Second, we also used SNVs to check the validity of our haplotype phasing. We reasoned that ancestral mutations acquired when the chromosome was in a diploid state would originally be present in 1/2 copies, subsequent aneuploidies will then shift the mutation copy number and VAF up or down. For example, for a parallel copy number event where 1 clone has a haplotype specific copy number = 2|1 and a second clone has a haplotype specific copy number = 1|2, the VAF of an SNV would be expected to flip between $\frac{1}{3}$ and $\frac{2}{3}$ in the 2 clones. This analysis is limited to cases where we can accurately infer ancestral mutations, and when we have sufficient depth to compute accurate VAFs across cells. When these conditions were satisfied, this analysis supports our haplotype specific copy number inference. See **Figure R8** below.

Figure R8 VAF of somatic mutations as a function of haplotype-specific state

For each panel, we plot the VAF of each somatic mutation in 2 haplotype-specific states where the dominant allele switches between the 2 states. Each point is the VAF of a single SNV, lines connect the same SNV in the 2 states. If the line decreases from left to right it is colored green, if it increases it is colored purple. Dashed lines indicate where we expect the VAF to be based on the states. The title of each plot gives the dataset and chromosome.

- Third, we also leveraged whole genome sequencing from bulk tissue to compare single-cell derived pseudobulk estimates to those derived from bulk. We found a high correlation with the fraction of the genome LOH, consistent ploidy estimates and high correlation in BAF values between modalities, **Figure R9**.

Figure R9 Comparison of bulk WGS copy number calls with pseudobulk single cell copy number calls
 a Fraction of genome LOH in the modalities, dashed line indicates $y = x$. b inferred ploidy in the 2 modalities c histogram of the correlation between mirrored BAF

- Fourth, we also generated long read sequencing data using Oxford Nanopore Technologies (ONT) from the wild type hTERT cell line. We ran the PEPPER-Margin-DeepVariant pipeline⁷ to phase heterozygous SNPs. Using the ONT data as ground truth, we contrasted the phasing produced from SIGNALS with random phasing and the phasing from SHAPEIT (used as input to SIGNALS). We used standard metrics to assess the quality of phasing; the switch error rate and the blockwise hamming distance. We found that SIGNALS had lower values for both these metrics, indicating improved phasing, **Figure R10**. We note that the switch error rate from the ONT data and pipeline is of the order ~ 0.01 (see Shafin et al), similar to what we observe here with SIGNALS, meaning we cannot distinguish accuracy beyond this level.

Figure R10 Assessment of SNP phasing using blockwise hamming distance (a) and switch error rate (b) vs ONT used as ground truth.

E. No formal benchmarking is done comparing Schnapps with either CHISEL or Alleloscope. These two methods are only cited in passing in the Introduction, without stating their close connections to schnapps.

Please see above for the details of our benchmarking assessment. We have also include a detailed discussion of the algorithms in our supplementary note. We have updated the main text as follows which refers to both methods, de-emphasizes claims of novelty, and points the reader to a detailed exposition should they be interested (line 87):

*“In addition, we developed a computational method called SIGNALS, a hidden Markov model which phases copy number events to individual homologs¹⁸ in single cell genomes in order to quantify allele-specific CNA as a source of cell-to-cell variation. Originally benchmarked on ovarian cancer cell line OV2295, SIGNALS evaluation across different technologies and multiple tumor types showed increased genomic and cellular resolution over previously published methods^{19,20} to 0.5 Mb, identified cell-to-cell diversity that would be obscured when relying on total copy number, and exhibited expected distributions of phased somatic point mutation variant allele fractions resulting from allele-specific gains and losses (**Extended Data Fig. 1,3, Supplementary Note**).”*

F. The scDNA-seq data sets used to check the accuracy of Schnapps in this paper have a mean coverage of 0.16x, which is much higher than most current scDNA-seq data sets (0.01-0.1x).

Our validation assessment now includes data from all samples in our study which has mean coverage ~0.04X. **Extended Data Fig. 3** and **Fig 3b** are especially relevant here in addition to the **Supplementary Note**.

Also, schnapps was only shown to work on data from this DLP protocol, which is not commonly used by the community. How does schnapps work on other platforms?

We have now run SIGNALS on data from the 10X CNV platform. We ran SIGNALS on patient S0, originally published in the CHISEL manuscript. We found an overall similarity of 93% between our results and those from CHISEL, however as discussed above we found that SIGNALS was better able to phase rare CNAs present in a few cells relative to CHISEL. We also provide a pipeline to produce allele-specific copy number estimates starting from BAM files which is agnostic to the underlying technology, see https://github.com/marcjwilliams1/hscn_pipeline.

Figure 10

Histograms of the similarity per cell between minor allele copy number and total copy number between CHISEL and SIGNALS.

2. On the biological findings:

A. The data analyzed in this paper were generated by Funnell et al. (<https://www.biorxiv.org/content/10.1101/2021.06.03.446999v1.full>) which was also attached as supplementary material to this paper. Thus, this paper does not contribute new data. Funnell et al. analyzed this data set in depth, but did not focus on allele-specific copy number events.

As recommended by the editor and the joint reviewer, our revised manuscript we have combined the results of Funnell et al and Williams et al into a single manuscript and focus on the new biological findings which can be described using our novel dataset and methods.

B. One of the main findings in this paper (stated in the abstract, and presented in Figure 3) is the high number of cases of parallel copy number evolution. The prevalence of such events were reported by Wu et al. (2021), through the analysis of >10 breast, gastric, and colon cancer samples. Raphael and Zaccaria also found such events in 2 breast cancer samples. Wu et al. further established the inter-clonal complexity at such hypermutable loci. Yet, both papers were only cited in passing in the introduction, and the findings of Wu et al. in this context were not mentioned.

We thank the reviewer for bringing this to our attention, it was an oversight on our part not to mention the contribution of Wu et al in this regard. We have now included the following sentence to the manuscript in the Discussion to better reflect the previous work (line 358):

“Second, we demonstrate that multi-allelic variation within the same locus is a highly prevalent feature of genomically unstable breast and ovarian cancers, consistent with some previous observations in human cancers^{19,20,27}.”

Given our new manuscript combines the results of Funnell et al. and Williams et al we would stress the following novelties not reported in previous studies

- We present the first description of parallel copy number evolution at single-cell resolution in high-grade serous ovarian cancer and our study reports (to our knowledge) the largest single-cell dataset where this has been explored to date (>22,000 cells across 23 patient tumors & >13,000 cells across 8 cell lines) and contrast how the three new foreground mutational processes distribute in the background of HRD and FBI tumors.
- The increased resolution of smaller populations provided by SIGNALS is better able to reveal the extent of this phenomenon. For example, CHISEL reported parallel copy number evolution only on chromosomes 2 and 3 in breast cancer patient S0, as demonstrated above we show that our method additionally identifies parallel copy number evolution in an additional 12 chromosomes. This allows us to compute the rate of sporadic gains and losses and contrast them on diploid vs tetraploid background, see main **Figure 5**.
- We show that parallel copy number evolution results in a convergent effect on transcriptional phenotype using matched scRNAseq (see main **Figure 2**)
- We integrate allele-specific information to reconstruct complex structural alterations such as BFBC leading to clone-specific high level amplifications (see main **Figure 2**)
- We measure the impact of parallel copy number events on evolutionary diversity
- We report that parallel copy number events impact allele specific expression from matched scRNASeq data

Taken together, overall we believe our study sheds new light on copy number evolution and phenotypic diversity implications of parallel allele-specific alterations, which is presented in **Figure 5** in the paper.

C. This paper establishes that breakage fusion bridge cycles (BFBC) are responsible for some of the more complex copy number aberrations in their data. Although BFBC, as a mutational mechanism, is well known, their analysis linking it to the complexity observed in their data is novel and convincing.

We thank the reviewer for this positive assessment of our work. In our revised paper, we have integrated these observations with those presented in Funnell et al. showing that BFBC in some cases is likely responsible for variable oncogenic amplification and variable CNA breakpoints between cells, as shown in **Fig 4**.

D. The integration and analysis of the scRNA-seq data needs to be more rigorous – in the current draft, it is only done for one sample. The proposed method of CNA detection in scRNA-seq data is crude, with no evaluation of accuracy. How do we know whether the CNA detection is confounded by RNA expression variation? The scRNA-seq data is used to show that gene expression does not differ between two subclones carrying alternate alleles at a CNA region. But the only way that the evidence is presented is through a UMAP feature plot, where the localization of the B-allele loss and A-allele loss do seem to differ (global trends are the same, but there are small local differences). Can this hypothesis be examined more rigorously?

We have now adapted our scRNAseq analysis and included further validation as discussed above. We have also included a new analysis, leveraging the shared nearest neighbor graph of gene expression values to

assess whether neighbors of cells with loss of allele A or allele B are more likely to be in the same state or not (see section titled **Nearest neighbour gene expression analysis** in **Methods** for further details). We find that the nearest neighbors of cells with either type of loss were enriched for both cells with loss of A and loss of B, consistent with a convergent phenotypic effect, see **Fig2i**.

Fig 2

h) UMAP dimensionality reduction plots of scRNAseq data generated from SA906b, colors indicate the density of loss of chr 2q A vs. B haplotype. i) Enrichment of the haplotype-specific state on chr 2q of nearest neighbor cells.

Minor Comments:

1. No sample "2295" mentioned in the main text, but it is used in the figure legend (Fig.1 and Supplementary table 1)

The main text has been updated as follows (line 89):

"Originally benchmarked on ovarian cancer cell line OV2295, SIGNALS evaluation across different technologies and multiple tumor types showed increased genomic and cellular resolution over previously published methods^{19,20} to 0.5 Mb, identified cell-to-cell diversity that would be obscured when relying on total copy number, and exhibited expected distributions of phased somatic point mutation variant allele fractions resulting from allele-specific gains and losses (Extended Data Fig. 1,3, Supplementary Note)."

2. More description of the engineered cell line data, and scientific hypotheses regarding them, in the main text may be helpful in interpreting the results.

In our revised manuscript a more detailed discussion of the cell line data is included and can be found on line 81:

"We first developed a combined experimental and computational approach for studying genome scale cell-to-cell variation in human cells, establishing an in vitro isogenic system of breast epithelium with induced HRD and defined temporal passaging. We generated TP53¹⁶, TP53/BRCA1 and TP53/BRCA2 loss of function alleles from diploid non-transformed 184-hTERT mammary epithelial cells¹⁷ using CRISPR-Cas9 editing (Fig. 1a, Extended Data Fig. 1,2,

Supplemental Table 1). We then subjected these cells to tagmentation whole genome single cell sequencing (DLP+), enabling scaled analysis of each population and inference of cell-specific rates of structural alterations¹¹.

3. Wrong legend for chr3m, 3l

This is no longer relevant due to restructuring.

4. The structural variant plots are hard to read without some explanation, especially Figure 6.

In our revised manuscript we have attempted to present this data in a more considered way being careful to only include relevant chromosomes when this visualization is used. In addition, these complex events are now validated with long-read nanopore technology as presented in **Fig 4i** and **Extended Data Figure 12**.

Figure 4

h) Consensus copy number profiles in 2 clones in SA1096 overlaid with lines indicating rearrangement breakpoints. i) ONT long read data from SA1096, confirming translocation between chr3 and chr6.

5. They claimed in line 457 that “The resolution of schnapps is 0.5Mb whereas CHISEL uses 5Mb bins”. The bin size is actually flexible for CHISEL.

While it’s true that the bin size is flexible, in our benchmarking analysis we found that running CHISEL at this resolution results in a significant drop in performance as outlined above. This was expected given the recommendations in the original CHISEL paper and from our discussions with the developer of CHISEL.

6. The A, B, C.... labeling needs to be revised in Supplementary Fig. 6.

No longer relevant due to restructuring.

7. Software details:

- The software can be run with the example datasets

- schnapps was developed to work with Direct Library Preparation + (DLP+) data. works using the output of the pipeline developed in their lab. Maybe they can provide guides to help prepare input data for scDNA-seq of other platforms.

We agree with the reviewer that we should attempt to make the software usable with other single-cell whole-genome technologies. We therefore provide a Snakemake pipeline and guide on how to generate inputs to SIGNALS, starting from single-cell bam files. In addition, we have included scripts to demultiplex 10X barcoded bams into single-cell bams required by our pipeline. This is provided at the following link - https://github.com/marcjwilliams1/hscn_pipeline, and a more in depth discussion on the inputs required by SIGNALS is provided on the software landing page (<https://github.com/shahcompbio/SIGNALS>).

- Need more explanation about the input data (what is "allele_id", "hap_lab"..and etc?)

We have added a much more detailed discussion of the input data on the SIGNALS documentation website (see **Input Data** section at <https://github.com/shahcompbio/SIGNALS>) as well as in our new **Supplementary Note** document.

- In the main page: one of the required input data is haplotype block counts per cell "(SNP counts may also work)". What does the results look like when using SNP counts?

We have now tested our ability to call haplotype-specific copy number using SNPs rather than haplotype blocks. We applied SIGNALS to sample SA1049 using both raw SNP counts and haplotype block counts. We observed broadly similar results, with a per cell minor allele copy number similarity of 95%, **Figure R11**. Overall we would recommend using reference panel based phasing as this allows for more accurate BAF estimates of rarer aneuploidies but leave this as an option to users of our software package.

Figure R11

Distribution of per cell minor allele copy number similarity when applying SIGNALS using raw SNP counts vs haplotype block counts.

Referee #4

The authors apply DLP+, an exciting technology that allows sparse sequencing of single cells and is scalable to thousands of cells (introduced previously by the authors in Laks et al., Cell, 2019). This method enables a deep dive into tumor heterogeneity. In the present paper, the authors analyze a data set on 11,097 cells from cell lines and to 21,852 cells from ovarian and breast cancers. The focus of this work is on performing analyses in a haplotype-specific manner and the authors introduce a computational framework to this effect. They apply it to study known processes, particularly breakage-fusion-bridge cycles, in cancer genomes. I list my comments and concerns in the following.

Major comments

- The authors discuss related work on CHISEL and Alleloscope (L457-470), which indeed are similar in spirit. I'd like to add that the idea to phase from B-allele frequencing goes back to at least 2014, where it was used by Schwartz et al. (PLoS Comp Biol), even though that was not in a single cell setting. I suggest to mention all these methods in the introduction.

We thank the reviewer for reminding us about this paper, now referenced. We are aware of the previous literature and having led 3 studies and methods that incorporated measurements of allelic fraction in WGS data. (Ha et al Genome Research 2012 PMID: 22637570; Ha et al Genome Research 2014 PMID: 25060187; and McPherson et al Genome Biology 2017 PMID: 28750660). We stress that the single cell data is materially different primarily due to coverage limitations for each cell and due to the actual observation of hundreds to thousands of individual cells. Our focus lies squarely on analysis of allele-specific changes that induce cell-to-cell variation in this paper and do not make any claims about computational advances over bulk methods. Rather we indeed borrow from previous literature to develop the SIGNALS approach and accordingly appreciate the conceptual nature of Schwartz et al contribution and include a citation to this work. In addition we use and cite the copy number edit distance approach proposed by Schwarz et al to compute the evolutionary distances as presented in Fig 1j, Fig 5m, and Fig 6f. Text updated as follows (line 112):

"Analysis of cell-to-cell pairwise copy number distances^{18,23}, computed with allele-specific profiles, found that TP53^{-/-} induced an 3.6-fold (SA906a) and 2-fold (SA906b) increase in cell-to-cell divergence, BRCA2^{-/-} a 4.8-fold (SA1055) and 3.6-fold (SA1056) increase, and BRCA1^{-/-} an 15.7-fold increase (Fig. 1j, p-values <10e-10) relative to pairwise distances in WT cells."

- The point out differences in resolution between schnapps (500kbp) and Chisel (5 Mbp). Given the great scalability of DLP+, I am actually surprised that the authors do not attempt to leverage the large number of cells to improve the resolution. Given the cumulative coverage of cell carrying a joint event, that should be possible computationally. Comparing to other works, I see that Sanders et al., reported 200kbp resolution with fewer cells.

While it's true that by merging cells we may be able to achieve higher resolution at a clone level, many of our analyses explore cell-to-cell variation and hence the important determinant is the average coverage in single cells rather than the number of cells. We find that 500kbp provides a reasonable trade-off between resolution and noise for robust copy number inference.

Additionally, we also note that we have now compared SIGNALS to chisel (with standard settings of 5Mb bins and as well as 0.5Mb bins) and alleloscope and find that SIGNALS does provide improved accuracy at higher resolution, see Figure R5 and our new Supplementary Note document.

- The claim that the majority of the genome is target to alterations (gains, losses, LOH) when lowering the cell fraction cutoff to 1% is certainly interesting (and relevant to our understanding of tumor evolution), but I'm not yet convinced that the authors have ruled out all possible artifacts. The number of cells per samples varies widely (around two orders of magnitude according to Table S1) and different cell counts could come with different challenges: For samples with very high cell numbers, I wonder whether there might be a multiple testing issues; i.e. could a subset of copy number diverged cells emerge by random fluctuations in coverage? For a sample with low numbers of cells (e.g. only 49 cells), >1% translates into an event seen in only one cell and that could introduce noise. Could the authors comment on the possible contributions of these effects and how they safeguard against this?

- Also, I would be interested to see this analysis stratified by event size, which is important for the interpretation. In other words, how finegrained is the background of CNAs upon which evolution can act?

In the restructuring of the manuscript, this analysis was removed. Nonetheless we thank the reviewer for bringing this to our attention and will bear this in mind for future work.

- L167-176: Zooming into Supp Fig 3, it appears that there are a number of chromosomes where the fraction of cells with both paternal and maternal gains is >0 for all seven lines. So the authors should expand on why they see chr20 as an outlier, which is not evident for me from looking at this figure. What do you propose drives selection for chr20 gains?

We agree that on closer inspection there are other chromosomes where both alleles are gained or lost across all or most cell lines. Notably, gains on chromosomes 13 and 17 and losses on chromosomes 2, 14 and 22. We have replaced this figure with a new figure to better draw out these comparisons, shown below in Figure R12. We would maintain that chr20 seems to be a striking example, particularly given this is also observed in the WT setting where genome instability is low. As to the mechanism, no canonical breast or ovarian cancer oncogenes are present on chr20, we note however that chr20 gains are common in these cancer types (~40% frequency in PCAWG), consistent with this event providing a selective advantage to cancer cells. It's possible that chr20 gain is beneficial to cells by modulating the gene dosage of a number of genes, alternatively there may be as yet undiscovered oncogenic drivers on chr20. Due to space constraints we do not include this result in the manuscript but do include it here for completeness.

Figure R12

Fraction of cells (CCF) that have gains or losses of each allele in each chromosome across all the cell line data.

- Regarding the BFBC analysis: I am not an expert for mutational mechanisms, but the schematic in Figure 4 is not consistent with my understanding of a BFBC. In the topmost row, the fusion appears to happen between the telomeres, which (to my understanding) is exactly what the telomeres should prevent; i.e. the common model is that BFBC processes happen **because** of the loss of a telomere. So the fusion is between the "blunt" ends of the sister chromatids.

We apologize for the lack of clarity here, the reviewer is correct to point out that the fusion between sister chromatids would ordinarily be due to blunt ends of the chromosomes following telomere erosion, we have amended the figure to include this, see Extended Data Figure 5a.

Extended Data Figure 5
a Diagram of breakage fusion bridge cycles.

However, the model in Figure 4a appears to be consistent with Figure 4b in terms of losses (Cluster F) gains (clusters A+B) of the same terminal part of the chromosome. Could the authors comment on the nature of this fusion, i.e. how could the derived chromosomes look like? Looking at Cluster F in Supp Fig 6, I start wondering whether in that subclone chr3 as acquired a telomere from chr17?

We did not find any breakpoints suggesting a fusion with chromosome 17. The only inter-chromosomal rearrangement we observed was a translocation with the X chromosome in Cluster F. In the remaining clusters of cells, rearrangements were intra-chromosomal, suggesting that the additional genetic material is appended to the chromosome as expected in a simple BFBC process. As to how the derivative chromosome stabilizes to avoid further changes we would speculate that our data is consistent with the following scenarios:

- i) In some cases the chromosome may not have stabilized and we are observing a BFBC mid-process.
- ii) The chromosome acquires a new telomere via telomerase activity.
- iii) The chromosomes stabilize via a rearrangement undetectable with short read sequencing such as with the telomeric end of another chromosome.

Another aspect that makes me wonder about the consistency of BFBCs with the shown data is the absence of high amplifications.

We note that in this case, what we are observing is the very beginning of a breakage fusion bridge cycle as this is an early passage in what was originally a diploid cell line. We also note that there is a small cluster of cells that has a modest amplification (copy number = 5) of *PIK3CA*, **Extended Data Figure 5**. We also note that we observe high level amplifications in cells in other lines of this system consistent with BFBC, these are included in a new main **Figure 2**.

f PIK3CA Amp.

Extended Data Figure 5

f Amplification of PIK3CA in TP53^{-/-} BRCA2^{+/-} cell line. Dashed line indicates the location of PIK3CA. Top track shows the total copy number, bottom track shows haplotype specific copy number states.

- Related to this: I was not able to get a good sense on how reliable the "SV track" in Fig 4c-h (and later figures) are? The Methods parts says that "SV calling was performed in a similar manner, by forming pseudobulk libraries, then running LUMPY and DESTRUCT on each pseudobulk library." Given the low per-cell coverage and how vastly the number of single cells varies across libraries, some evaluation of reliability vs pseudobulk coverage would be helpful. In general, more details are needed here. How are the calls between the two tools merged?

In order to assess our ability to recover SV's in pseudobulk clones we calculated the sensitivity of our approach using data from an ovarian cancer cell line. This data consists of cell lines generated from 3 sites at different time points (referred to as OV2295 in the text), meaning we can confidently identify breakpoints present in 100% of cells (those that are found in all 3 samples). We sampled groups of cells of varying size, calculated the cumulative coverage of these groups and then assessed the fraction of breakpoints we recover. We classified a breakpoint as "recovered" if any cell within the group of sampled cells had evidence of the breakpoint. **Figure R13** shows the sensitivity as a function of cumulative coverage. At a cumulative coverage of ~5X we recover 80% of the breakpoints that are present, ~5X coverage translates to approximately 100-150 cells given our median coverage of 0.04X per cell. This analysis is included in the **Supplementary Note**.

Figure R13

Sensitivity of structural variant recall as a function of the cumulative coverage of randomly sampled cells that are merged to form a pseudobulk.

Do I interpret the figures correctly that this process was run separately for each cluster?

This process is not run per cluster, rather we run the calling on the combined set of cells in a library and then look for evidence for breakpoints at a read level in individual cells. If any cell in a cluster has evidence for the breakpoint then the breakpoint is included in these figures. This has been clarified in the methods section (line 231 Methods).

“When performing pseudobulk analysis on groups of cells, a breakpoint is considered present in a clone if at least 1 cell that constitutes the clone contains evidence of the breakpoint. A subsampling experiment determined that this approach has 80% power to recover breakpoints at a cumulative coverage of 5X (100-150 cells), see Supplementary Note.”

I went to Laks et al., 2019 because of the pointer in Line 479, but the respective method part was also sparse on the SV calling part.

We have now extended the methods section on the breakpoint calling (line 226):

“LUMPY and deStruct predictions were considered matched if the breakpoints matched in orientation and the positions involved were each no more than 200 nucleotides apart on the genome. Only deStruct predictions with a matching LUMPY prediction were retained. Sparse per cell breakpoint read counts were extracted from deStruct using the cell identity of read evidence for each predicted breakpoint. SNV and SV calls were further post-processed according to Wang et al.”

- The author elevate the term "haplotypes" to the title and emphasize the phased nature of their analyses. In my view, that warrants a detailed evaluation of the completeness and accuracy of the phasing. Both are not obvious given that a) one can only phase regions with gains, losses, or LOHs in at least a subpopulation of cells and b) the data is also very sparse per single cell, translating into sparse phasing of regions with CNA in few cells only. The notion of haplotype blocks and interaction with Shapelt should be described in more detail. Ideally, an evaluation would be done from sequencing data from the parents of a cancer patient, but that data can obviously be difficult to acquire. Alternatively one could resort to sequencing

technologies providing long(er) range phasing information such as long read sequencing, linked reads, Hi-C or Strand-seq. One aspect that I'm particularly concerned about is the phasing along an individual chromosome: Looking at Figure 4b again,

I wonder how confident the authors can be that the different segments affected by the (putative) BFBC are indeed all from the same homolog? Given the nature of the phasing algorithm, additional confirmation is warranted in my view.

This is an important question and the reviewer is correct to point out that this is key to much of our results. We have now included a number of additional analyses to validate our haplotype inferences. Firstly we looked at the distribution of SNV VAFs in subclones with different haplotype-specific copy number and demonstrate that these change as expected, see reply to reviewer #3 and **Figure R8**.

Secondly we generated long read data using oxford nanopore technologies (ONT) from the wild type hTERT cell line and ran the PEPPER-Margin-DeepVariant pipeline⁷ to phase heterozygous SNPs. These results showed that SIGNALS improves phasing along the genome relative to reference based approaches see reply to reviewer #3 and **Figure R10**.

Finally, on the specific point of the phasing along chromosome 3 in the cell line, we are confident in our ability to accurately phase in this cell line data because we run our algorithm on the combined set of all cells across all the cell line data (~13,000 cells). We are able to do this because all the cells are derived from the same common ancestor. Due to the large number of cells in this dataset, we always observe a cluster of cells with a whole chromosome aneuploidy which can be used for unambiguous phasing. Specifically for chromosome 3, SIGNALS identifies a cluster of 28 cells in a triploid state for phasing, see **Figure R14a**. As further confirmation of the phasing, we also show examples of a diploid single cell, a whole chromosome loss of haplotype A and a whole chromosome loss of haplotype B, **Figure R14b-c**.

Figure R14 Chromosome 3 phasing

a Cluster of 28 cells used for phasing chromosome 3 in the cell line data here shown as a pseudobulk average over all 28 cells b Example of a diploid single-cell from the cell line data c Example of a whole chromosome loss of allele B from the cell line data d Example of a whole chromosome loss of allele A from the cell line data. For each panel we show the BAF in 0.5Mb bins (top) colored by state and the corrected read counts colored by total copy number state (bottom).

- To my understanding, the performance of the phasing method is substantially influenced by the heterogeneity of the tumor (in a homogeneous tissue without gains/losses/LOHs, one cannot use B-allele frequencies). The only signal left in such a setting is statistical phasing using a reference panel, which has a limited reach (typically 100kbp to 1Mbp, depending on the sample ancestry and size of the reference panel). So an evaluation on the influence of the tumor heterogeneity on phasing capabilities would be good.

This is an important point that the reviewer raises and they are indeed correct to point out that the phasing relies on observing cells with allelic imbalance. In the absence of cells with allelic imbalance we cannot phase the chromosomes. To assess our ability to accurately infer haplotype specific copy number in datasets with limited heterogeneity we i) generated simulations with varying levels of heterogeneity and ii) undertook a detailed investigation of the quality of copy number calls in a sample that is homogeneous but harbors a number of rare chromosomal aneuploidies.

i) Simulations

We generated a simulation scheme that allowed us to introduce subclonal whole chromosome aneuploidies at varying cancer cell fractions (see Supplementary Note for further details). We simulated a dataset with 500

cells and introduced aneuploidies present in 1,2,5,10 and 15 cells and assessed our ability to recover the correct haplotype-specific copy number in these subclones. We found a small drop in accuracy when the event was only present in 1 or 2 cells but overall accuracy remained above 99%, **Figure R15**.

Figure R15

Accuracy of inferred haplotype specific calls as a function of the size of the smallest clone in a simulated dataset of 500 cells.

ii) Sample with limited heterogeneity

While they can be useful as a way of generating ground truth data, simulations of single-cell data likely miss key aspects of noise that may influence the accuracy of calling copy number. To assess our ability to call haplotype-specific copy number we made use of sample SA1047, one of our datasets with minimal heterogeneity but a number of rare aneuploidies present in < 10 cells. Strategies that we used to check the overall consistency of calls such as using matched bulk copy number calls and distributions of SNVs are not possible here however. We therefore resort to contrasting calls made when haplotype blocks are unphased vs phased, and heuristics such as the consistency of BAF distributions in called states.

SA1047 contained aneuploidies present in a small minority of cells on chromosomes 8 (7 cells), 11 (3 cells) and 13 (2 cells), **Figure R16a**. Here we focus on the whole chromosome losses on chr13. We contrasted the results from the following scenarios:

- haplotype blocks remain unphased across chromosomes
- using 1 cell as the minimum cluster size
- using 10 cells as the minimum cluster size

In the case of unphased haplotype blocks, called states oscillated between 1|0 and 0|1 and BAF values were distributed randomly between 0 and 1, inconsistent with a single copy loss. When using 1 cell for clustering, BAF's were heavily skewed towards either 0 or 1 indicating that these cells lost different homologs, and consistent with the read depth based total copy number state of 1, **Figure R16b,c**. When we used 10 cells for clustering, there was a bias toward 0 or 1 but it was not as pronounced. Investigating the BAF distribution

across all rare aneuploidies showed distributions consistent with whole chromosome losses, **Figure R16d**. Overall, these results provide evidence that SIGNALS is able to correctly phase haplotypes and call haplotype specific copy number even when CNAs are rare.

Figure R16 Phasing rare populations in SA1047

a Heatmap representation of sample SA1047. Left shows total copy number, right shows haplotype specific copy number states. Arrows show small aneuploid populations b The same single-cell with a chromosome 13 loss with results shown when SIGNALS was run with a 1 cell cutoff, a 10 cell cutoff and unphased haplotypes. c A different cell with a chromosome loss on 13, this time of the opposite haplotype. d BAF distributions in the cells highlighted in panel a with chromosome losses

We also demonstrate increased reliability of phasing and haplotype specific copy number in rare CNAs over CHISEL, see **Figure R6** reply to reviewer #3.

Minor comments

- **Methods / from L392: Please completely define all notation so that a reader does not need to guess. What exactly is a "block allele" and what is a "haplotype block" (is "block" define based on the bin size or otherwise)? What is the j in $C_{\{h,j\}}$ and what is N ?**

This has been modified in the new Supplementary Note and we have now also included a schematic diagram outlining our approach, see **Figure R19** in reply to reviewer #5.

- **L425/426: How is the overdispersion parameter estimated from read counts?**

The overdispersion parameter is estimated using maximum likelihood estimation using the VGAM package in R. This takes in total read counts and alternate read counts as input and computes the overdispersion parameter that maximizes the likelihood using numerical optimization. We assess statistical support for the Beta-Binomial model using Tarones z-score¹². If we find a z-score > 5, the Beta-Binomial model is used. Tarones z-score was > 5 for all our samples therefore the Beta-Binomial model was used in all our inferences. This is now discussed in more detail in the **Supplementary Note**.

- **L231-233: Did the tree construction process in Supp Fig 6 omit chr3? (If it didn't, then it would not be independent).**

We have removed this from our analysis.

- **There seem to be some mismatches between numbers reported in the manuscript and the supplementary table, e.g.: "We next investigated the landscape of copy number alterations in the tumor cohort from single cell whole genome sequencing (median 697 cells per sample, range 49-2,627), ..." Table S1 reports 4214 cells for SA609, so the range seems to be larger (and the median lower).**

We have ensured that the tables and manuscript text are consistent.

- **Figure 1d: I did not formally count them, but there seem to be way fewer rows of pixels than 1031. So is that image actually displaying some kind of average across adjacent cells? Please indicate the used bin size in the caption.**

It's correct that there are fewer pixels than bins X cells in the heatmaps. The ComplexHeatmap R package is used to plot the heatmaps, and we used the default rasterization. A blog post detailing the rasterization employed in ComplexHeatmap can be found here should it be of interest:

<https://jokergho.github.io/2020/06/30/rasterization-in-complexheatmap/>

We also now provide all data and copy number profiles at <https://zenodo.org/record/6362734> allowing readers to explore single cell copy number profiles at will. SIGNALS includes an extensive suite of functions to visualize single cell copy number profiles, see the following vignette for more details

https://shahcompbio.github.io/SIGNALS/articles/basic_scdnaseq.html.

4. Joint review of both papers

Referee #5

[Reviews to the manuscripts (Funnel et al.) and (Williams et al) are presented together below.]

The studies led by Shah and Aparicio (in the following we will refer to them by their respective first authors) build upon single-cell whole-genome sequencing (scWGS) analyses of human primary tumors, patient-derived xenografts, and engineered cell lines to explore features of chromosomal instability during tumor evolution. These studies certainly benefit from a compelling dataset (although already partly published in PMID: 34163070) and rigorous computational biology analyses, including the development of novel tools that we can see of interest and useful to the scientific community.

However, where these studies seem to suffer is in the biological novelty introduced by their finding. As we discuss in more detail below, the results presented are largely confirmatory and descriptive. In particular, observations from one manuscript could often benefit the other and the choice of separating the two studies seems rather counterproductive. Below, we separately discuss in more detail the two studies, highlighting where integration of the two would be most beneficial.

Funnel et al.)

In our opinion, this study is the weakest of the two in terms of biological findings and should be integrated into the second study. This work focused on mutational signatures associated with the emergence of structural variants rather than single nucleotide mutations and how these signatures determine chromosomal instability in different models of serous ovarian and triple-negative breast cancer. In particular, by scWGS, the authors explored these mutational processes within single cells.

The stratification of patient samples in Figure 2 sets the stage for future analyses but signatures and associations were previously shown, e.g. the differences between BRCA1 and BRCA2 mutations (e.g., see previous work from the authors PMID: 30794536 or PMID: 32118208) or the similarity between TNBC and HGSC (PMID: 23000897 or PMID: 23636398). This figure could either be supplemental or substantially reduced to show the stratification of patient subgroups. (Figure 1 should be substantially edited as it currently includes figure panels representing actual data - so it seems - but it is unclear what data.)

In our revised manuscript, the previous figure 2 has been moved to the supplement (Extended Data Figure 7). We have also substantially revised former Figure 1 to better highlight the novel contributions of our paper

including the description of new mutational processes and also moved this revised figure to the supplement (**Extended Data Figure 1**).

The analysis in **Figure 3** is rather intriguing as it delves into the differences between **BRCA1** and **BRCA2** chromosomal instability at the single-cell level. Here we have a few questions:

* **First of all, the choice of models is sometimes not sufficiently motivated. Why do the authors provide a heterozygous and a homozygous model for BRCA2, but not for BRCA1?**

We have now generated data from a **BRCA1** heterozygous cell line. As expected, genomic instability in this cell lines was more similar to the **TP53^{-/-}** cell lines than the **BRCA1^{-/-}** cell line, consistent with previous work showing that homozygous knockout of **BRCA1** is necessary for downstream phenotypic effects¹³. This data is included in our new **Figure 1**.

* **Similarly, for BRCA2^{-/-} the authors analyze two independent replicates at different time points, should the differences between the two samples be attributed to the different times they were kept in culture? If so, why BRCA2^{+/-} cells were analyzed at a much earlier time point? Would these cells at a later time point accumulate more chromosomal aberrations? Importantly, what is the replication time of these cells? The role of time-in-culture is not sufficiently discussed and analyzed in these results and the authors should consider the rate of accumulation of chromosomal aberration w.r.t. cell cycle divisions.**

We have now generated new data from the **BRCA2^{+/-}** line at a later time point that is more consistent with other libraries. As the reviewer correctly predicted this later time point has more chromosomal abnormalities than the earlier time point, see **Figure R17**. However, this data is more similar to the **TP53^{-/-}** line than the **TP53^{-/-} BRCA2^{-/-}** line. This is consistent with a heterozygous knockout being insufficient to increase genomic instability, and that the genotype has greater influence than time in culture on observed chromosomal instability.

All the cell-line data used for comparisons are now compared at similar time points, within 10 passages of each other.

Figure R17

Copy number heatmap for SA1188 from an early time point(left) and a later time point (right).

To further explore the role of time in culture we made use of our time-series data (some of which was first published in ²) to assess how the rate at which chromosomal alterations accumulate over time differs between genotypes. Lines for which we have time-series data include the WT (SA039), TP53^{-/-} (SA906a & SA906b) and TP53^{-/-} BRCA2^{+/-} (SA1188). We found that in all these data, the average number of chromosome arm aneuploidies per cell increased over time, **Figure R18**.

For each cell, we computed the number of chromosome arm abnormalities and then performed a regression as defined below

$$n_{chrarm} \sim \beta \times t + 0$$

Where t is the time in culture and we assume 0 aberrations at time 0 by enforcing the intercept to be 0. β is, therefore a coefficient that describes the number of alterations accumulated per passage. This analysis was consistent with our other analyses whereby genotypes with TP53^{-/-} accumulated more alterations per passage relative to WT (see **Figure R18** below). We also note that the rate of accumulation of chromosome arm aneuploidies in TP53^{-/-} BRCA2^{+/-} relative to TP53^{-/-} is comparable (0.139 vs 0.11 and 0.156).

Figure R18

Number of chromosomal arm aneuploidies per cell (y-axis) vs passage number (x-axis) for 4 cell lines. β is the coefficient from a regression of $y \sim x + 0$ and thus has units of number of chromosomal arm aberrations per passage. Red line shows best fit regression line.

Moreover, some results from this analysis are expected, e.g., the association between number altered segments and polyploidy, and could be moved to supplement, whereas it would be great to show here how BRCA1 and BRCA2 SV signature are implemented at the level of single alleles, which is part of the analyses in the second manuscript (Williams et al.).

We thank the reviewer for this suggestion, in our revised paper, Figures 1 and 2 present the data from our cell line experiments, these include:

- Example cells showing total copy number and allele specific copy number profiles
- Abundance of parallel CNAs as a function of genotype
- Rates of homozygosity as a function of genotype
- Distributions of cell-to-cell diversity computed using allele specific copy number distances as a function of genotype

The authors then move to a more detailed comparison between HRD-dup (i.e. the SV signature associated with BRCA1 mutations) and FBI-associated tumors (another SV signature independent of both BRCA1 and BRCA2) in human samples.

(We should note that although 5 additional samples not exhibiting either of these signatures are shown in the figure, these are never discussed in the manuscript. We suggest removing them from the main figure as they just add more complexity to an already complex figure.)

Here, it is difficult to exactly understand what is the point the authors want to make. Indeed, the features of FBI and HRD-dup signatures have already been characterized, in part by the authors themselves (see for example PMID: 28436987, PMID: 29754820, PMID: 30104763) and these results simply confirm in single cells what was already known from bulk tumor analyses.

We have simplified the patient stratification to include 4 groups: HRD-Dup, HRD-Del, TD, and FBI. As a result the single cell comparisons were updated to include the HRD-Dup, TD, and FBI groups. We have further reorganized the manuscript to focus on novel structural variation differences between the signature types, and moved parts of Figure 4 to the supplementary materials. The new supplementary figure has been updated to reflect the new patient stratification, and to be easier to read. See reply to reviewer #1 and **Extended Data Figure 7** for further details.

In addition, the figure is difficult to read as it misses color legends in individual panels, and some of the colors are used multiple times with different meanings. Results in Figure 4e are expected and results in Figure 4f and 4g are redundant. Additionally, the sample ordering in Figure 4d,e,f,g is not consistent, making it very difficult to match the results in the various panels. In Figure 4j there is a great overlap in the segment length distributions between HRD-Dup and FBI in interiors, while the only tangible difference at termini is between HRD-Dup gains and HRD-Dup losses and FBI gains/losses, so there is no clear distinction between the two signatures even in this case.

In order to focus on the more novel aspects of our analysis we have now moved parts of this figure to the supplement. Additionally we have reorganized the figure with new data, and maintained consistent sample ordering.

Given the previously reported association between FBI and HLAMP (see refs above), the authors search for known oncogenes in highly amplified regions from FBI samples. Here they found a few interesting targets, however, more details are needed.

First and foremost it is unclear how focal these events are. Indeed, instead of providing a genome-wide map of copy number statuses, the authors should zoom in on the affected loci and clearly show the size of these events. Indeed for some of the proposed targets (see BCL11A and MET), copy number gains seem to be arm-events.

We thank the reviewer for this suggestion, in our new Figure 4, we first focus on a focal KRAS amplification, showing that variability at the DNA copy number level results in phenotypic variability at the expression and protein level. In our assessment of the HLAMPs across the whole cohort, we have assessed the width of copy number region, finding that the majority of variable HLAMPs are indeed focal alterations (56% < 10Mb). We have included this data as an additional track in Fig4e (see below) and include a sentence as follows (line 241):

“The majority of events were < 10Mb in width (56%, Fig. 4e) and exhibited a distribution of maximum observed copy number with median 16.1 and IQR 8.7 (Fig. 4f)”

In these cases, it is difficult to say that these events are selected because they amplify a specific oncogene. Here the authors should have used matched RNA-seq data to show that these oncogenes or their downstream targets are up-regulated when amplified. In the absence of this data, I wouldn't make a big deal about these events.

We agree that demonstrating that this variability at the DNA level translates to variability at the RNA level is important. To do this, we generated single-cell RNA seq using the 10X platform from a subset of patients exhibiting these patterns. We clustered single cell transcriptomes using leiden clustering and then performed differential expression analysis between clusters. Oncogenes that we identified to be variable among clones in our scDNAseq data tended to have larger log fold change (logFC) between gene expression clusters relative to other genes, consistent with increased DNA copy number resulting in increased expression, **Figure 4h**.

Figure 4

e) Width of genomic segment containing the amplification, dashed red line indicates width = 10Mb **f)** Max cell copy number for genes outlined in **g)** Clone max/min copy number ratio of cancer genes overlapping HLAMP regions. Genes across all cancer datasets with ratio > 2 shown. Colors as per **f)**. For **e)-g)** distribution of values are shown with a violin plot on the right hand side

Furthermore, given the association of the FBI signature with bridge fusion break cycle (BFBC) events (which was shown in previous studies from the authors) this analysis would gain robustness and completeness if combined with the analysis of the same amplicons done in Williams et al. where BFBC events are analyzed in detail. Here it seems the authors did an overall analysis of these events to then split the results into two studies, but each half feels incomplete without the other.

We thank the reviewer for this suggestion and have now integrated these observations in our merged paper. In particular, we extracted and then clustered genomic features of the HLAMPs and found a cluster enriched for FBI rearrangements, consistent with BFBC.

Finally, the authors highlight what they call a serriform pattern for chromosomal breakpoints, where the same/matched structural variant exhibits slightly different breakpoints in different single cells. As discussed by the authors, this pattern was already shown in vitro (PMID: 32299917) but was here first shown in human tumors. Was it named differently before? or simply not named?

To our knowledge, this phenomenon was not previously given a name.

This analysis is rather interesting, here we would like however to have a better feeling of the incidence of this phenomenon across different samples. What is the fraction of breakpoints that exhibits this pattern?

Variable cell-to-cell breakpoints were common, with 6.6% of regions having serration scores ≥ 0.15 (i.e. $\geq 15\%$ of cells have a rare event breakpoint position) across all cases, with FBI cases having the highest (12.1%), HRD-Dup cases the lowest (1.2%), and TD cases intermediate (10.5%) rates. We have updated the text with these statistics.

The association with FBI, although significant, is rather weak and supported by a limited number of samples. Do the authors think it would be possible to develop an approach to estimate the presence of this pattern from bulk sequencing data (e.g. by detecting reads supporting a breakpoint in different positions)? Pseudo-bulk of their samples could be used for testing. This could be rather intriguing and possibly a way to validate their findings in a larger cohort of samples.

We agree this would be very interesting, however, identifying cell-specific rearrangement SVs remains an open problem and is fraught with confounding levels of likely false positive predictions. We have identified this as a key area for algorithm development in future work that will be the subject of a computational PhD student dissertation. Accordingly we leave the identification and study of cell-specific SSV breakpoints to future research.

Importantly, it would have been nice here to show this event at the level of single alleles using the approach proposed in the Williams et al. manuscript.

We thank the reviewer for motivating us to perform this analysis. We have now performed the SSV analysis using our haplotype specific copy number calls. Intriguingly in some cases we observe the phenomena happening on both alleles. Below we show an example of an SSV event impacting both alleles observed in one of our TP53^{-/-} cell lines, **Figure 2j** (right most heatmap). We did not observe convincing examples of such bi-allelic events in the tumor data however.

Figure 2 j Copy number heatmap showing variation in breakpoint location across cells. Top to bottom: dataset, breakpoint location, # cells; ideogram indicating chromosome region shown in heatmap; average copy number across cells in heatmap, with breakpoint-adjacent segment copy number states indicated with dotted black lines; copy number states inferred from HMMCopy; haplotype specific states inferred from SIGNALS. Heatmap x-axis: genomic bins; y-axis: cells with the indicated breakpoint. Greyscale passage number indicates time points (passage #) of each cell, cells are ordered by breakpoint position (left-to-right).

Overall, this study shows interesting features of tumor evolution under the effect of distinct mutational signatures. However, the findings are limited and descriptive and alone do not reach the level of novelty one expects from Nature. The authors could instead consider migrating the most innovative parts to the manuscript by Williams et al..

Williams et al.)

In this manuscript, the authors introduce a new approach to determine the copy number status of single cells at the level of single haplotypes. They then analyze haplotype-specific copy number alterations in the same datasets introduced above.

At first, the authors present their algorithm, called schnapps, and show its performance on a previously published dataset comprising multiple samples derived from the same patient and analyzed by scWGS. The analysis shows coherent B-allele frequencies (BAF) and variant allele frequencies (VAF) for SNPs, validating the approach. Here we have a few questions:

* In Figure 1B: given that the BAF is part of the inference procedure of schnapps, isn't it expected that values are distributed around the expected values for each allele combination?

This is indeed true that the BAF is used in the algorithm via the beta-binomial likelihood of allele read counts used in the HMM. We have removed this panel from the new version of this figure, included as **Extended Data Figure 3**. In addition, we have included a detailed **Supplementary Note**, providing further details on the algorithm, validation of results and comparison with other methods. We note that plotting the distribution of BAF can be a useful way to diagnose issues with haplotype specific copy number calls and we use this where appropriate in the supplementary document.

Extended Data Figure 3 a,b) haplotype-specific copy number from 2 individual cells from the OV2295 cell line. c) Total copy number heatmaps and haplotype-specific copy number heatmap for 1084 cells. Each row is an individual cell. Rows are ordered according to a umap + HDBSCAN clustering, with clusters annotated on the left hand side. d) Distribution of variant allele frequencies as a function of haplotype-specific state. e,f,g) Variant allele frequencies in clones where the dominant allele switches between A and B. Each point is the VAF of a mutation, with lines connecting the same mutation in different clones.

* the description of the algorithm in the method section could be improved, especially in the use of notation and terminology. Is there a difference between “block allele” and “haplotype block”? What is the difference between the (Ch, Dh) and (Ca, Cb) notation?

We apologize for a lack of clarity here, we have now included more detail of our algorithm in a supplementary methods document. In addition we have produced a diagram which gives a conceptual overview of the inference problem, included below as **Figure R19**. Our input data is haplotype blocks of variable length, which are defined based on regions where SNPs can be confidently phased together (using the SHAPEIT algorithm). Within each haplotype block, we have read counts for each allele, which is what we call a “block allele”. The challenge can be summarized as how do alleles across different haplotypes block phase together and allow us to reconstruct the homologous chromosomes. This can be solved by looking at cells with allelic imbalance and phasing together those alleles that shift in frequency in a coordinated way.

Figure R19 Schematic overview of the inference problem and how it can be resolved using allelic imbalance in single cells a) SHAPEIT partially phases genomes at a length scale of kb's to MB's into haplotype blocks, the inference challenge is then to go from these partially phased to fully phased genomes b). c) Allelic imbalance in single cell can provide unambiguous phasing. In the case of a cell with LOH, we will observe counts for the homozygous allele (purple in this case), thus in any given haplotype block, the "block allele" with non-zero counts are necessarily on the same homolog and can be phased together. In the case of a triploidy event, allele within each block will have different frequencies, the purple allele should have BAF $\frac{1}{3}$ and the green a BAF of $\frac{2}{3}$ allowing us to phase these events together.

* **The code description is great but at least the processed data should be made available for reproducibility (bin-level copy number status)**

All data needed to reproduce the findings of the paper have now been uploaded to <https://zenodo.org/record/6362734>.

* **What is the effect of the delta parameter in the transition matrix of the HMM? It is currently set at 0.95, but how was this value determined? How robust are the results of the method to changing this parameter?**

We have now assessed the influence of the transition probability using a range of values spanning from 0.1 to 0.999 and also including the case where there is no correlation structure between neighboring bins. We found that changing the transition probability has minimal effect on our results, even in the extreme case of comparing the results in the IID case versus $p_T = 0.999$ the similarity in assigned states is 95%. We found that when $p_T > 0.5$, the number of segments per cell and the distance between the raw BAF and expected BAF based on the assigned state converged to a stable value. Given these results, we reason that any value > 0.5 gives acceptable results and continue to use 0.95 as the default in schnapps. See **Figure R20**.

Figure R20

a Heatmap of similarity between haplotype-specific calls using different transition probabilities b Distribution of the distance between the expected BAF and the empirical BAF as a function of transition probability. Each data point is the BAF value from a bin in a single cell c Number of allele specific segment per cell as a function of the transition probability

* A comparison with previous approaches is warranted here. The authors have simply described the differences between schnapps and CHISEL and Alleloscope, but they should have actually tried to run these tools on the same dataset and show the differences in performance.

We have now performed a detailed comparison with other mentioned tools. In summary we found that the results were broadly similar, but SIGNALS provides increased resolution at both the genome level (5Mb vs 0.5Mb) and cellular level (ability to accurately infer haplotypes only present in a small number of cells) relative to other methods. See reply to reviewer #3 and the detailed **Supplementary Note** for more information.

* The authors do not provide access to the processed data. We understand that patient raw data should be accessed only under proper access control, but having access to the anonymized processed data (for example, bin level copy number estimates of each cell of the study together with their allelic status) could be very valuable. Additionally, we do not understand the inaccessibility of the engineered cell line raw data. Where can we find the processed data to reproduce the analysis presented in the Github repository?

It was an oversight not to provide the raw data in our original submission. We have now provided all processed data necessary to reproduce the results in our paper, they are available at <https://zenodo.org/record/6362734>.

Figure 1d is really nice but we have trouble understanding the color code of the second heatmap. If we read correctly the color legend, the allelic imbalance in the heatmap should range between 0 and 0.5 (only blue hues are shown), but this is in contradiction with Figure 1c, isn't it?

We apologize for the lack of clarity in our original manuscript. To ease the interpretation we have changed the color scheme from brown/blue to green/purple to better distinguish total copy number from haplotype specific copy number. We also introduce a new visualization which rather than plotting the total copy number and BAF separately, we plot the inferred copy number of each homolog, see panels a and b in **Extended Data Figure 3** above.

The analysis of clonal and subclonal events that follows (Figure 2) is an intriguing negative result, i.e. subclonal events do not show evidence of selection. A question that comes to mind is whether there is any association here with the genetic context.

Is there any difference between BRCA1/2 mutated tumors and others?

Most notably, given this information is already available to the authors, what about the association with mutational signatures?

This analysis feels a bit out of scope here, it does not focus on allele-specific changes and seems rather more connected to the other manuscript by Funnel et al.. Combining the two would make a much nicer chapter.

Due to the merging of the two papers and consequent limitations in space available for a full description of all results in the 2 separate papers we elected to remove this analysis. However, we appreciate the comments and will take these into account in ongoing and future work.

The analysis of parallel events is also quite interesting but very difficult to follow in Figure 3. Color legends should be more clearly shown in association with the figure panels and if possible I would refrain from using the same/similar colors for total copy number and allelic imbalance. A suggestion: would it be possible/simpler to show two separate heatmaps for the estimated copy number status of each allele?

We apologize for lack of clarity, as mentioned above we have now changed our color scheme from brown/blue to green/purple to better distinguish total copy number from haplotype specific copy number. This analysis constitutes Figure 4 in our revised manuscript.

Also, it is unclear from the methods what are the requirements to call a parallel gain or loss. It looks like several events are quite different in size between the two alleles, although intersecting in a region (e.g. Figure 3g,h,i). The authors should clarify how the overlapping of changes in maternal and paternal haplotypes are computed.

We apologize for the lack of clarity here. The reviewer is correct to point out that by this definition the breakpoints of the events do not need to match exactly. We have clarified this in the text (line 375 Methods):

“Parallel CNAs were defined as genomic regions greater than 4Mb where gain or loss of both the maternal and paternal haplotype was observed in more than 1% of cells. Copy number breakpoints of segments do not need to match to be included.”

The correlations in Figure 3k are rather weak with points clearly deviating from the line of fit. Anything to note in these samples? Is there an association with polyploidy or mutational signature?

We agree that the correlations here were rather weak and have now reassessed our original approach. Given the complexity of the genomes we study, simple missegregation rate estimates are not the most appropriate measure here. We have therefore now used 2 new metrics to assess how underlying instability influences rates of parallel CNA evolution: copy number distance distributions and phylogenetic distances. Here we found a much stronger relationship, see **Figure 5m**.

In addition we also explored how underlying rates of instability influence the abundance of parallel copy number evolution in our cell line data. We observe more parallel events as we progressively knockout key DNA damage response genes. The WT cell line has the lowest number of parallel events, which is increased by TP53^{-/-} and further increased by BRCA2 and BRCA1^{-/-}, **Figure 2g**. This analysis is now included in panel c of main Figure 2 and provides additional evidence that parallel copy number evolution increases as a function of underlying rates of instability.

Figure 2
g) Number of parallel copy number events per cell in 184-hTERT mammary epithelial cell lines. Cell line genotypes are shown at the bottom

Was the UMAP in Figure 3m generated using expression data for all genes or only for those in chr 2 (or 2q)? If the former, the authors cannot conclude that the clustering shows convergent phenotypes as it could be driven by a myriad of other SIGNALS independent of chr. 2q. Can the authors actually test for allelic expression imbalance for genes in the lost part of the chromosome? Bulk RNA-seq data could be used for that.

In our original analysis we used all genes. The rationale was that our single cell whole genome sequencing data did not find any other CNAs that were associated with these losses that would confound the analysis. To confirm this, we have now repeated the analysis, restricting to genes found only on chromosome 2q as the reviewer suggests and find similar results, **Figure R21**. We have also included a new analysis, leveraging the shared nearest neighbor graph of gene expression values to assess whether neighbors of cells with loss of allele A or allele B are more likely to be in the same state or not. We find that the nearest neighbors of cells with either type of loss were enriched for both cells with loss of A and loss of B, consistent with a convergent phenotypic effect.

In addition, we were able to look at allelic expression imbalance in highly expressed genes present on chr2q. We identified 15 genes that had sufficient coverage that we could determine allelic imbalance at the gene level and found BAF distributions in these genes consistent with mono allelic expression in cells that were called as homozygous with loss of either allele A or allele B. **Figure R21e**.

Figure R21

a UMAP plot coloured by gene expression clusters. Only genes found on chr 2 were included. b UMAP colored by kernel density estimation of cells with loss of B allele c UMAP colored by kernel density estimation of cells with loss of A allele d Nearest neighbor enrichment of cells in different chr2q states e Single cell RNAseq derived BAF distributions for 15 genes that had a minimum of 10 cumulative counts in each haplotype specific state. Distributions are stratified by whether chr2q is called in a balanced state or with loss of A or B. Large circles show the pseudobulk derived BAF's and violins show distributions over individual cells.

Lastly, the remaining figures of the manuscript are highly curated reconstructions of specific events. Whereas the effort to reconstruct some of these events is commendable, the value seems rather anecdotal. The analyses in Figures 4 and 5 could be combined and should be put in the context of the mutational signature that generated these events. Figure 6 is unreadable and, again, as much as we understand the difficulty of representing this data, it is unclear what we can learn from it.

These have been mostly removed as standalone figures, but are included to enhance the interpretation of the high level amplifications which are composed of complex chromosomal rearrangements. To facilitate their interpretation we have been careful to only include relevant chromosomes when these figures are included. In addition, these complex events are now validated with long-read nanopore technology as presented in Fig 4 and Extended Data Figure 12.

Furthermore, we have performed a more general analysis of the genomic features of HLAMPs by extracting various features such as the complexity of the copy number profiles and the type of structural variants identified close to the HLAMP locus. Here we found 3 notable groups: 1 group enriched in FBI rearrangements consistent with BFBC, a group enriched for simple duplications that drive HLAMPs in the HRD-Dup signature

type and a group enriched in inter-chromosomal rearrangements. We believe this improved presentation provides a clearer and more intuitive view of the data, see Extended Data Figure 11.

Extended Data Figure 11 Genomic feature of HLAMPS, each column is a HLAMP that amplifies an oncogene. Each row is a feature extracted from a region 15Mb either side of the amplification. Complexity = entropy of allele-specific states, #SV = total number of structural variants identified, proportion of SV's of each type: foldback, duplications, deletions and translocations. #chr = number of chromosomes involved in translocation. bin/chr ratio copy number of the bin containing the oncogene to the average copy number across the chromosome. Ratio is the copy number ratio between the clone with the maximum copy number state and the minimum copy number state.

In conclusion, we feel that this manuscript (Williams et al.) brings forward a new methodology of possibly broad applicability, and its potential is nicely shown through multiple analyses revealing high allelic imbalance in chromosomal instability and patterns of convergent evolution between events in the homologous chromosomes. These results could benefit from the mutational signature analysis performed in Funnel et al., which here are completely ignored.

Overall, we strongly encourage the authors to propose a substantially revised and harmonized combination of the two manuscripts into one, streamlined from the many confirmatory results and focused instead on what new we can learn from these scWGS datasets and computational approaches.

We have taken up the reviewer's suggestion to merge the concepts presented in the two distinct papers. In this revised, single manuscript, we indeed find additional novelty and value by integrating the signature types with allele-specific copy number analysis. This thread is pulled throughout the manuscript and allowed for focusing on how these three new 'foreground' patterns distribute in the cell lines and in the tumors with typed signature analysis.

References

1. Fischer, M. Census and evaluation of p53 target genes. *Oncogene* **36**, 3943–3956 (2017).
2. Salehi, S. *et al.* Clonal fitness inferred from time-series modelling of single-cell cancer genomes. *Nature* **595**, 585–590 (2021).
3. Funnell, T. *et al.* Integrated structural variation and point mutation signatures in cancer genomes using correlated topic models. *PLoS Comput. Biol.* **15**, e1006799 (2019).
4. Vázquez-García, I. *et al.* Immune and malignant cell phenotypes of ovarian cancer are determined by distinct mutational processes. *bioRxiv* 2021.08.24.454519 (2021) doi:10.1101/2021.08.24.454519.
5. Farmery, J. H. R., Smith, M. L., NIHR BioResource - Rare Diseases & Lynch, A. G. Telomerecat: A ploidy-agnostic method for estimating telomere length from whole genome sequencing data. *Sci. Rep.* **8**, 1300 (2018).
6. McPherson, A. W. *et al.* ReMixT: clone-specific genomic structure estimation in cancer. *Genome Biol.* **18**, 140 (2017).
7. Shafin, K. *et al.* Haplotype-aware variant calling with PEPPER-Margin-DeepVariant enables high accuracy in nanopore long-reads. *Nat. Methods* (2021) doi:10.1038/s41592-021-01299-w.
8. Wu, C.-Y. *et al.* Integrative single-cell analysis of allele-specific copy number alterations and chromatin accessibility in cancer. *Nat. Biotechnol.* (2021) doi:10.1038/s41587-021-00911-w.
9. Zaccaria, S. & Raphael, B. J. Characterizing allele- and haplotype-specific copy numbers in single cells with CHISEL. *Nat. Biotechnol.* **39**, 207–214 (2021).
10. Schwarz, R. F. *et al.* Phylogenetic quantification of intra-tumour heterogeneity. *PLoS Comput. Biol.* **10**, e1003535 (2014).
11. Zeira, R. & Raphael, B. J. Copy number evolution with weighted aberrations in cancer. *Bioinformatics* **36**, i344–i352 (2020).
12. Tarone, R. E. Testing the goodness of fit of the binomial distribution. *Biometrika* **66**, 585–590 (1979).
13. Jonsson, P. *et al.* Tumour lineage shapes BRCA-mediated phenotypes. *Nature* **571**, 576–579 (2019).

Reviewer Reports on the First Revision:

Referees' comments:

Referee #1 (Remarks to the Author):

The authors have combined their two previous manuscripts, and have done a considerable amount of work to revise the united manuscript according to the Reviewers' comments.

The focus of the manuscript has consequently changed, and I believe that -- compared to the previous version -- the current version of the manuscript is much clearer and easier to follow, is more interesting/novel, and does a much better job at integrating the data from the in vitro and in vivo data.

The authors have addressed my specific concerns (or, in some cases, shifted the focus of the paper and omitted the data to which those comments referred). However, I have a couple of additional comments:

(1) I am somewhat ambivalent about the definition of "foreground" vs. "background" mutational processes. While I appreciate the usefulness of this distinction, I am afraid it can be somewhat misleading, as it implies a temporal order for these processes (i.e. that "background" processes act earlier than "foreground" processes). However, the "foreground" and "background" processes act simultaneously, and "background" mutational processes are expected to keep contributing to ITH in parallel to the newly-identified "foreground" processes. I therefore believe that this requires adjustments, or at least further clarification in the text.

(2) Fig. 1d-h demonstrate that increased levels of CNAs are found in TP53-null/BRCA1-null and TP53-null/BRCA2-null cells in comparison to TP53-null cells. In the bulk WGS cohort, this marked difference across these genotypes is not observed at all, based on Fig. R2, which is not included in the manuscript. Therefore, the biological relevance of this finding from the engineered cell lines is unclear to me. Although not central to the current version of the manuscript, I think that this point should be mentioned explicitly and explained clearly in the manuscript.

I was asked by the Editor to also assess the authors' response to Reviewer #2's comments. The authors note that the revised manuscript de-emphasized the bulk data observations, and that this makes some of the comments by Reviewer #2 less relevant to the current manuscript. I generally agree with their contention. Reviewer #2 mentions two impact concerns: one is related to the novelty of the approach – in my opinion, this is less of a concern in the current version of the manuscript that doesn't emphasize the tool; the second is related to the therapeutic importance of the results – this concern has not been addressed, but PDX drug experiments are beyond the scope of this work, and the novelty of the paper comes from the identification and characterization of the cell-to-cell genomic variability in three intriguing mutational processes. I believe that the authors have adequately addressed the other specific comments of Reviewer #2.

Referee #3 (Remarks to the Author):

In this paper, the authors combined novel technology development with a systematic investigation

of copy number alterations in breast and ovarian cancers at single cell resolution. This is a merging of two previous drafts, one focused on biology and the other on methods development. Overall I think the revision took the strengths from the two contributing manuscripts and integrated them quite well.

Compared to existing studies, this paper goes to depth in the mutational mechanisms underlying genome instability. The contributions here add substantially to the picture we have of genome instability, and they are a strength of this paper.

The new more rigorous benchmarking, on wider range of data sets, substantially enhances the rigor of the methods component of the work.

The integration with scRNA-seq data is still not so convincing and offers limited insights. But that seems not to be the focus of this paper and so I am willing to let it go.

Some minor comments on the comparison between computational methods SIGNALS, CHISEL, and ALLELOSCOPE:

- Since CHISEL and SIGNALS is conceptually similar, both based on local smoothing (HMM vs bin-smoothing) it is not surprising that CHISEL and SIGNALS have similar performance metrics.

- Alleloscope requires the identification of a single diploid region in the tumor samples, after the estimation of cell-specific major haplotype proportion, for the ensuing computation of normalized cell-specific total coverage. In tricky cases, Alleloscope allows manual specification of this region, since incorrect assignment of baseline can drastically affect results. How is the diploid region chosen in the running of Alleloscope?

Referee #5 (Remarks to the Author):

We appreciate the authors' choice of combining the two previous studies, the new revised manuscript has gained in clarity and depth. The authors satisfactorily addressed our requests and the additional datasets introduced are noteworthy. We have no further requests.

Reviewer Reports on the First Revision:

Referee #1 (Remarks to the Author):

The authors have combined their two previous manuscripts, and have done a considerable amount of work to revise the united manuscript according to the Reviewers' comments.

The focus of the manuscript has consequently changed, and I believe that -- compared to the previous version -- the current version of the manuscript is much clearer and easier to follow, is more interesting/novel, and does a much better job at integrating the data from the in vitro and in vivo data.

We thank the reviewer for the positive assessment of our revised merged paper.

The authors have addressed my specific concerns (or, in some cases, shifted the focus of the paper and omitted the data to which those comments referred). However, I have a couple of additional comments:

(1) I am somewhat ambivalent about the definition of "foreground" vs. "background" mutational processes. While I appreciate the usefulness of this distinction, I am afraid it can be somewhat misleading, as it implies a temporal order for these processes (i.e. that "background" processes act earlier than "foreground" processes). However, the "foreground" and "background" processes act simultaneously, and "background" mutational processes are expected to keep contributing to ITH in parallel to the newly-identified "foreground" processes. I therefore believe that this requires adjustments, or at least further clarification in the text.

We agree the terminology does not cover all concepts. Accordingly, we now consistently refer to 'foreground' mutational cell-to-cell variation *patterns* (ie not vestigial) and 'background' mutational *processes* which are the endogenous processes resulting in the foreground patterns. We think this strikes the right balance for readers to clearly understand that we are comparing foreground cell-to-cell variation in the background of the different endogenous processes (e.g. HRD). We also reduced the number of instances of 'background' to reduce confusion.

(2) Fig. 1d-h demonstrate that increased levels of CNAs are found in TP53-null/BRCA1-null and TP53-null/BRCA2-null cells in comparison to TP53-null cells. In the bulk WGS cohort, this marked difference across these genotypes is not observed at all, based on Fig. R2, which is not included in the manuscript. Therefore, the biological relevance of this finding from the engineered cell lines is unclear to me. Although not central to the current version of the manuscript, I think that this point should be mentioned explicitly and explained clearly in the manuscript.

We appreciate this comment, however, we would note that differences between the in-vitro data and the human data make meaningful comparisons across genotypes challenging. The primary motivation for developing our in-vitro system was to produce a controlled system in which we would expect to see differences in the amount and extent of genomic instability. This then allowed us to interrogate common patterns and features of such instability. We chose to focus on the homologous recombination pathway as genes in this pathway are well characterized and thus amenable to genetic engineering. By contrast, within the human data, BRCAwt samples will include other phenotypes of genomic instability that we describe in our paper including the foldback inversion (FBI) type and tandem duplicator (TD) type. Thus, it is

not surprising that we see minimal differences in the extent of aneuploidy in BRCAmt vs BRCAwt in our bulk sequencing cohort and Figs 1d-h are therefore most relevant to HRD mediated genomic instability.

I was asked by the Editor to also assess the authors' response to Reviewer #2's comments. The authors note that the revised manuscript de-emphasized the bulk data observations, and that this makes some of the comments by Reviewer #2 less relevant to the current manuscript. I generally agree with their contention. Reviewer #2 mentions two impact concerns: one is related to the novelty of the approach – in my opinion, this is less of a concern in the current version of the manuscript that doesn't emphasize the tool; the second is related to the therapeutic importance of the results – this concern has not been addressed, but PDX drug experiments are beyond the scope of this work, and the novelty of the paper comes from the identification and characterization of the cell-to-cell genomic variability in three intriguing mutational processes. I believe that the authors have adequately addressed the other specific comments of Reviewer #2.

Referee #3 (Remarks to the Author):

In this paper, the authors combined novel technology development with a systematic investigation of copy number alterations in breast and ovarian cancers at single cell resolution. This is a merging of two previous drafts, one focused on biology and the other on methods development. Overall I think the revision took the strengths from the two contributing manuscripts and integrated them quite well.

Compared to existing studies, this paper goes to depth in the mutational mechanisms underlying genome instability. The contributions here add substantially to the picture we have of genome instability, and they are a strength of this paper.

The new more rigorous benchmarking, on wider range of data sets, substantially enhances the rigor of the methods component of the work.

We thank the reviewer for the positive assessment of our work.

The integration with scRNA-seq data is still not so convincing and offers limited insights. But that seems not to be the focus of this paper and so I am willing to let it go.

Some minor comments on the comparison between computational methods SIGNALS, CHISEL, and ALLELOSCOPE:

- Since CHISEL and SIGNALS is conceptually similar, both based on local smoothing (HMM vs bin-smoothing) it is not surprising that CHISEL and SIGNALS have similar performance metrics.

- Alleloscope requires the identification of a single diploid region in the tumor samples, after the estimation of cell-specific major haplotype proportion, for the ensuing computation of normalized cell-specific total coverage. In tricky cases, Alleloscope allows manual specification of this region, since incorrect assignment of baseline can drastically affect results. How is the diploid region chosen in the running of Alleloscope?

When running alleloscope we used the default approach described in the manual <https://github.com/seasoncloud/Alleloscope/tree/main/samples/SNU601/scDNA>. Many of our samples include highly complex genomes with considerable cell-to-cell variation which results in limited diploid control regions (sometimes no region of the genome remains in a heterozygous diploid state). Furthermore, the assumption made by Alleloscope that cells share an identical segmentation profile but with cell-to-cell differences in copy number is invalid in our data. These particularities of our data may explain the relatively poorer performance of Alleloscope relative to SIGNALS and CHISEL. We have clarified these points in the section “**Discussion of tools**” in the **Supplementary Methods** document.

Referee #5 (Remarks to the Author):

We appreciate the authors' choice of combining the two previous studies, the new revised manuscript has gained in clarity and depth. The authors satisfactorily addressed our requests and the additional datasets introduced are noteworthy. We have no further requests.

We appreciate the reviewer's positive assessment of our new combined manuscript.